# The Jurassic rise of squamates as supported by lepidosaur disparity and evolutionary rates

**Arnau Bolet[1,2]\*, Thomas L Stubbs[2], Jorge A Herrera-Flores[2], Michael J Benton[2]**

[1]Institut Català de Paleontologia Miquel Crusafont, Universitat Autònoma de Barcelona, Cerdanyola del Vallès, Spain; [2]School of Earth Sciences, University of Bristol, Bristol, United Kingdom

**Abstract** The squamates (lizards, snakes, and relatives) today comprise more than 10,000 species, and yet their sister group, the Rhynchocephalia, is represented by a single species today, the tuatara. The explosion in squamate diversity has been tracked back to the Cretaceous Terrestrial Revolution, 100 million years ago (Ma), the time when flowering plants began their takeover of terrestrial ecosystems, associated with diversification of coevolving insects and insect-eating predators such as lizards, birds, and mammals. Squamates arose much earlier, but their long pre-Cretaceous history of some 150 million years (Myr) is documented by sparse fossils. Here, we provide evidence for an initial radiation of squamate morphology in the Middle and Late Jurassic (174–145 Ma), and show that they established their key ecological roles much earlier than had been assumed, and they have not changed them much since.

## Editor's evaluation

This article presents an evaluation of the macroevolutionary history of squamates (lizards, snakes, and relatives) and is relevant to evolutionary biologists and paleontologists interested in this group. The 'early burst' of disparity in squamates demonstrates that squamates established their morphospace range much earlier than had been assumed, and the long-term stable morphospace occupation ever since.

**\*For correspondence:**
arnau.bolet@icp.cat

**Competing interest:** The authors declare that no competing interests exist.

## Introduction

Lepidosaurs, currently mainly represented by squamates, are one of the most species-rich tetrapod clades (*Uetz et al., 2019*), only rivaled by birds in terms of diversity. Evidence points to an explosion in squamate biodiversity 100 million years ago (Ma), in the Cretaceous (*Lloyd et al., 2008*; *Longrich et al., 2012*; *Mongiardino Koch and Gauthier, 2018*), corresponding to the time when flowering plants were diversifying and restructuring terrestrial ecosystems. However, the origin of squamates at least 250 Ma (*Jones et al., 2013*; *Simões et al., 2018*) poses two challenges: was that 150 million years (Myr) of poorly documented history real or evidence of a poor fossil record; and when did squamates acquire their current range of ecological adaptations? These uncertainties contrast somewhat with more confident knowledge of the early radiations of birds (*Lee et al., 2014*; *Wang and Lloyd, 2016*; *Benson et al., 2014*) and mammals (*Close et al., 2015*).

The squamate fossil record is patchy, especially through the Triassic to Early Cretaceous interval (252–100 Ma) when fossils are sparse and incomplete (*Cleary et al., 2018*). The earliest unambiguously identified squamate fossils date from the Middle and Late Jurassic (174–145 Ma), and among them are forms that can be assigned to major modern clades of squamates, including both lizards and

snakes (*Evans, 1998*; *Evans, 2003*; *Caldwell et al., 2015*), but many are isolated jaws and skull bones of difficult identification. Further, their rarity suggests diversity was not high.

The mid-Cretaceous shows an increase in abundance and diversity of squamates (*Gauthier et al., 2012*), linked to the Cretaceous Terrestrial Revolution (KTR), which triggered an outburst of terrestrial life, including major new clades, such as angiosperms, as well as ferns, hornworts, liverworts among plants, currently highly diverse insect groups, including cockroaches, termites, many groups of beetles, bugs, the wasp, ant, and bee lineage, and the butterfly and moth lineage. These rich new supplies of plants and insects provided food for expanding clades of insect-eaters, including spiders, birds, mammals, and lizards, and even perhaps some dinosaur groups (*Lloyd et al., 2008*; *Doyle, 2008*; *Benton, 2010*; *Meredith et al., 2011*; *Cardinal and Danforth, 2013*). It could be plausible to identify this as the time when squamate ecological adaptation expanded, but is this truly the case?

Here, we explore morphospace distribution, disparity (morphological richness), and evolutionary rates of lepidosaurs to understand these important stages in the first three-quarters of squamate history. Species richness is hard to document with confidence in the face of such a patchy fossil record (*Cleary et al., 2018*), although when combined with phylogenomic data, the relative timing of origins of major modern clades can be identified (*Jones et al., 2013*; *Simões et al., 2018*). Sparse fossil data can be used, on the other hand, to document disparity, even though extreme forms may be absent, and the morphospace may be poorly filled. Phylogenetic comparative methods are used here to explore whether the available fossil record before the Late Cretaceous suggests that the importance of the Jurassic in the rise of squamates has been underestimated, hinting at a cryptic diversity hidden behind an impoverished fossil record.

## Results

Our dated phylogeny of lepidosaurs (*Figure 1*, *Figure 1—figure supplement 1*), based on a morphological tree constrained by phylogenomic evidence (*Figure 1—figure supplements 2–4*, see *Figure 1—figure supplements 5–7* for results of an unconstrained analysis), shows that the clade originated around the Permian-Triassic boundary, and that by the Mid-Triassic it was represented by different extinct groups. The Rhynchocephalia diversified in the Mesozoic, but reverted to a single species subsequently, the living tuatara (*Sphenodon punctatus*) from New Zealand. The Squamata, on the other hand, show a step of diversification through the Jurassic, as the main modern clades emerged, and then a further diversification in the mid-Cretaceous, perhaps linked to the KTR. The ranges of ages for nodes differ depending on the method used (*Figure 1—figure supplement 1*), with the Hedman method (*Figure 1*) yielding the oldest divergence dates for nodes, the MBL (Minimum Branch Length) method yielding the youngest ones, and the equal method being intermediate. According to these results, it was regarded as necessary to consider all three methods of dating when performing the evolutionary rate analyses in order to discard the possibility that results were biased by the selected dating method.

The dated phylogenetic tree only describes the outline of the origins of squamate biodiversity, but does not map species numbers or, importantly, the range of morphology, and presumably the range of adaptation, reflecting ecological impact, of the group. Using a large morphological dataset (*Conrad, 2018*), covering 201 species of living and fossil lepidosaurs scored for 836 skeletal morphological characters, we analyzed disparity for lepidosaurs through time and tracked changes to morphospace occupation and major expansions. Stacked temporal morphospaces (*Figure 2*) show that rhynchocephalians and squamates occupy mutually exclusive morphospaces. Stepping up through time, from bottom to top of the stack, shows how lepidosaur morphospace expanded, not gradually, but marked by a single major step. At first, the total morphospace is small, formed by stem lepidosaurs and rhynchocephalians in the Triassic, and exclusively by rhynchocephalians in the Early Jurassic. Note that the lack of Early Jurassic stem lepidosaurs is artificial because they are found again (and possibly for the last time) in the Middle Jurassic as represented by *Marmoretta*. Then, with the addition of squamates in the Middle to Late Jurassic bin, morphospace expands to five or six times the area – the limits are established by rhynchocephalians, generalized lizards, anguimorph lizards, and snakes, each occupying a separate area of morphospace. We coin the term Jurassic Morphospace Expansion (JME) for the event related to this sudden increase in morphospace, which is interpreted as evidence of the initial radiation of the total group Squamata. Note that the coincident loss of stem lepidosaurs does not result in a modification of the morphospace hull because of the central position of these taxa. This

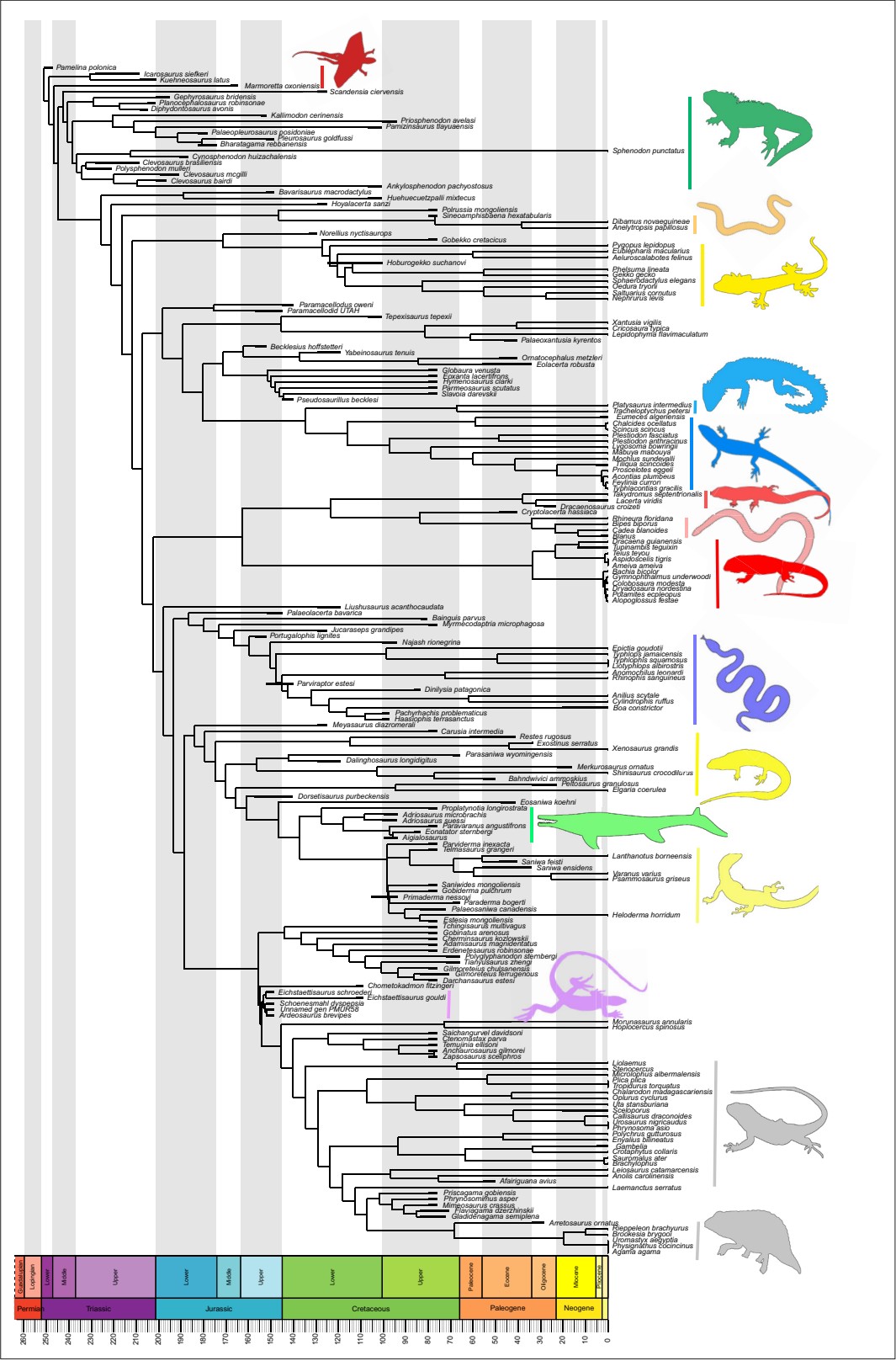

**Figure 1.** Lepidosaur— phylogeny, morphospace, disparity, and evolutionary rates. Phylogeny represented by a single randomly selected tree among those most parsimonious trees (MPTs) of the constrained analysis, and temporarily calibrated with the 'Hedman' method. Fossil ranges for each lineage are indicated according to the

*Figure 1 continued on next page*

*Figure 1 continued*

temporal distribution of the sampled taxa. For complete phylogenies and alternative datings, see *Figure 1—figure supplements 1–7*.

The online version of this article includes the following figure supplement(s) for figure 1:

**Figure supplement 1.** Single, randomly selected, most parsimonious tree among those resulting from the constrained analysis of lepidosaurs, dated using the equal method, and showing the position and ranges of nodes times using all three methods (Hedman in green, equal in blue, and MBL in red).

**Figure supplement 2.** Majority rule consensus tree for the constrained analysis of lepidosaurs.

**Figure supplement 3.** Adams consensus tree for the constrained analysis of lepidosaurs.

**Figure supplement 4.** Strict consensus tree for the constrained analysis of lepidosaurs.

**Figure supplement 5.** Majority rule consensus tree for the unconstrained analysis of lepidosaurs.

**Figure supplement 6.** Adams consensus tree for the unconstrained analysis of lepidosaurs.

**Figure supplement 7.** Strict consensus tree for the unconstrained analysis of lepidosaurs.

---

morphospace configuration remains remarkably stable from the Late Jurassic through to the present, with only subtle increases in morphospace occupation and in the density of points inside the envelope, notably in the mid Cretaceous coinciding with the KTR and the consequent recorded increase in diversity.

At this point, we should comment on the form-function relationship. It is well understood that form (skeletal morphology) does not always equate to function (*Wainwright et al., 2005*), with many functions sometimes performed by organisms of apparently similar morphology, or many different morphologies capable of performing a single function. However, here we calibrate the morphospace by mapping living taxa of known function and ecology onto the fossil time slices. This means we can mark (*Figures 2 and 3*) the rhynchocephalian (cluster 1, in green) pole as dietary generalists with robust jaws and tongue prey prehension like the modern *Sphenodon*; the generalized lizard (cluster 2, in blue) pole as diverse insect-eaters, like modern skinks; the anguimorph (cluster 3, in yellow) pole as active foragers with tendency to carnivory; and the snake (cluster 4, in violet) pole as limbless predators that feed mainly on other vertebrates and, to a lesser degree, invertebrates. Note that these clusters are loosely based on the ones recovered using the R 'pamk' function of the fpc package (see *Figure 3—figure supplement 6*, *Source code 3*). Admittedly, this is a simplification because, just as an example, rhynchocephalians contain dietary specialists (durophagous, piscivorous, etc.) and examples of forms adapted to swimming (like pleurosaurs), and the same occurs with specific clades of squamates (e.g., herbivore iguanians). Although a more precise ecomorphological classification would potentially provide more information on the distribution of ecologies through time, it has proven impractical for the current dataset, also in agreement with previous results (e.g., *Simões et al., 2020*). However, our classification serves the purpose of depicting that the extremes of morphospace had been achieved by the Middle to Late Jurassic (*Figures 2 and 3D*). Regarding squamates, it is worth noting that the first members of the group sampled in our analyses are Late Jurassic in age, but Middle Jurassic forms are known, just happen to be not included in the current morphological dataset because they are too incomplete.

The illustrated morphospaces (*Figures 2, 3A and B*, *Figure 2—figure supplement 2*, *Figure 3—figure supplement 1*) represent the first two major axes of variation, and there could be additional morphospace expansions along the other main axes: this is not the case (*Figure 2—figure supplements 3 and 5*; *Figure 3—figure supplements 2 and 3*). Further, the story does not change when the post-Cretaceous time bin is divided into Paleogene and Neogene time slices (*Figure 2—figure supplements 2; 4 and 6*). In a plot of total disparity (i.e., the sum of variances [SoVs] across all morphospace axes) of lepidosaurs through geological time (*Figure 4A*), the two peaks of elevated disparity (Middle–Late Jurassic and mid–Late Cretaceous) are clear. These summary data also confirm the much higher total disparity of squamates than rhynchocephalians (*Figure 4—figure supplement 1*) and among the former, a higher disparity of snakes and anguimorphs (and less clearly mosasaurs) than the remaining main groups of squamates among which dibamids, lacertids, and the extinct group of ardeosaurs (sensu lato) are the ones with the lowest disparity (*Figure 4—figure supplement 2*).

The 'two-peak' pattern is also identified through a study of evolutionary rates among lepidosaurs (*Figure 4B*). Maximum-likelihood analyses of rates of morphological character evolution clearly show

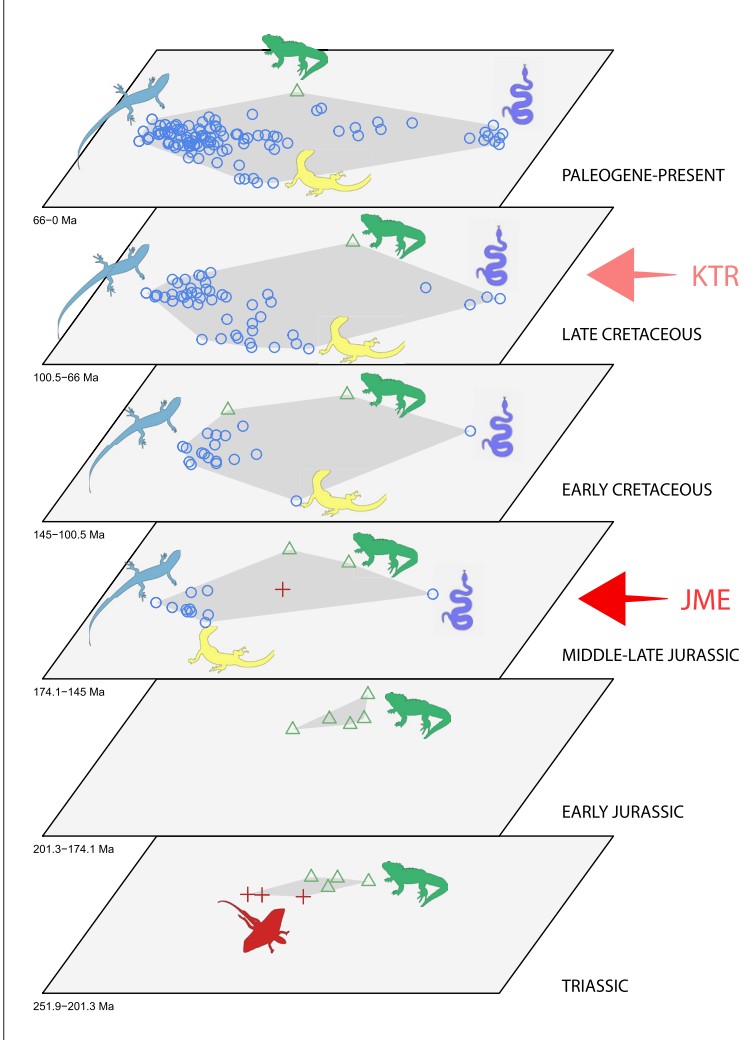

**Figure 2.** Morphospace occupation through time. Blue circles correspond to squamates, with the blue scincid silhouette indicating the position of generalized lizards, the yellow varanid indicating the position of anguimorphs, and the violet snake the position of snakes (and other limbless squamates). Green triangles correspond to rhynchocephalians (green *Sphenodon* silhouette). Red crosses correspond to stem lepidosaurs (red kuehneosaur silhouette). For additional plots of morphospace occupation through time, see *Figure 2—figure supplements 1–6*. JME: Jurassic Morphospace Expansion; KTR: Cretaceous Terrestrial Revolution.

The online version of this article includes the following figure supplement(s) for figure 2:

**Figure supplement 1.** Morphospace of lepidosaurs through time for PCO1 and PCO2.

**Figure supplement 2.** Morphospace of lepidosaurs through time for PCO1 and PCO2.

**Figure supplement 3.** Morphospace of lepidosaurs through time for PCO3 and PCO4.

**Figure supplement 4.** Morphospace of lepidosaurs through time for PCO3 and PCO4.

**Figure supplement 5.** Morphospace of lepidosaurs through time for PCO5 and PCO6.

**Figure supplement 6.** Morphospace of lepidosaurs through time for PCO5 and PCO6.

that the highest rates occur in the Late Jurassic (coinciding with the peak in disparity and roughly coinciding with the observed expansion of morphospace), and there is a lower peak in the mid-Cretaceous (see *Figure 4—figure supplements 5–7*). The high peak is for squamates (see *Figure 4—figure supplements 8–10* for squamates), not rhynchocephalians (see *Figure 4—figure supplements 11–13*), and in our opinion all three observations (morphospace expansion, increase of disparity, and fast evolutionary rates) are linked and support the existence of the JME event. Rhynchocephalians showed their highest rates of evolution in the Triassic, and those rates declined substantially through

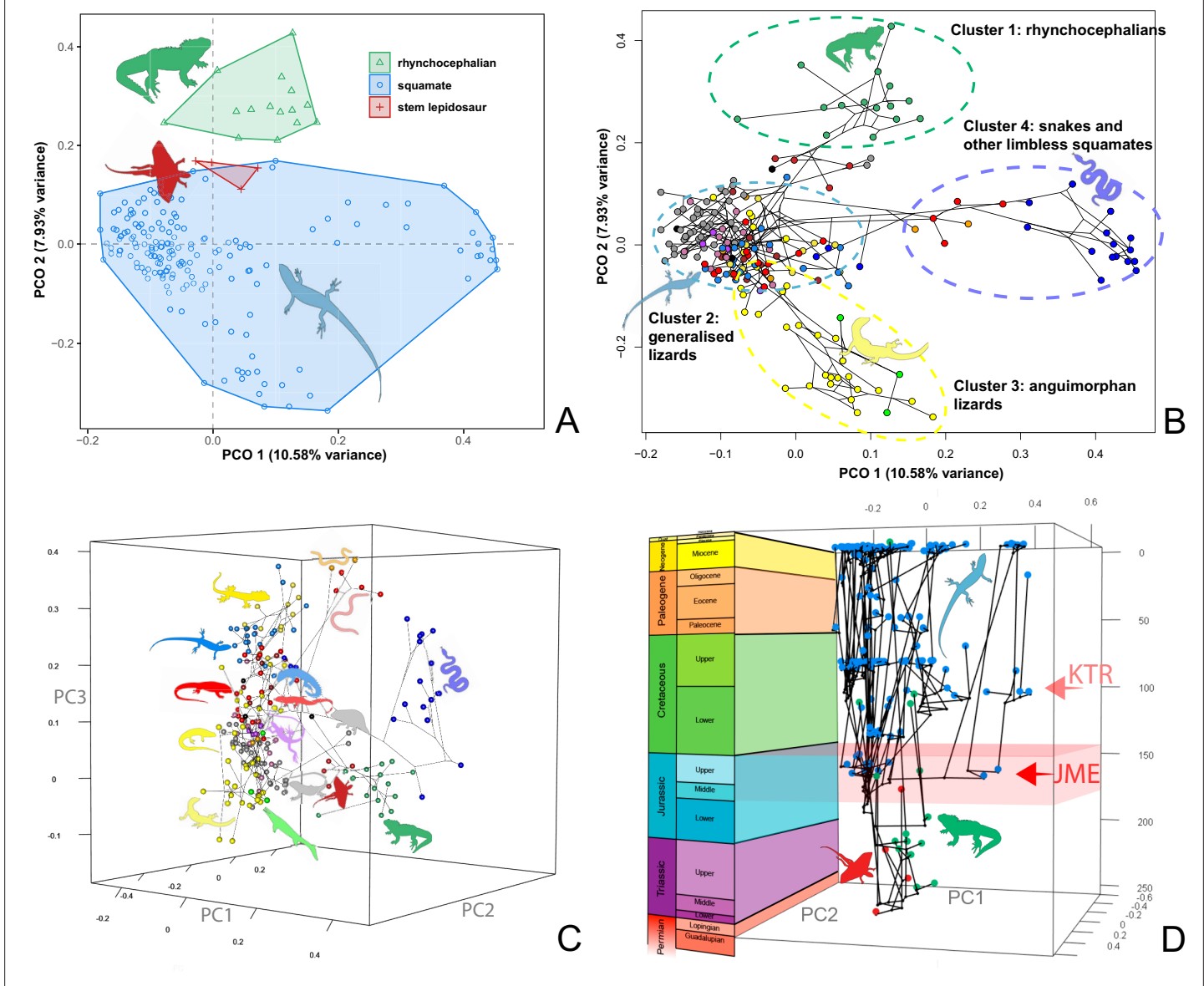

**Figure 3.** Lepidosaur morphospace. (**A**) Morphospace based on the two major axes of variation (PCO1 and PCO2), with colors and symbols according to the three main taxonomic groups. (**B**) Phylomorphospace distribution in PCO1 and PCO2, with lower taxonomic groups labeled. (**C**) 3D phylomorphospace illustrating the three major axes of variation (corresponding to PCO1, PCO2, and PCO3), with colors and symbols denoting to the lower taxonomic groups (see color legend in *Figure 1—figure supplement 2*). (**D**) Chronophylomorphospace of lepidosaurs showing the expansion of morphologies on the two major axes of variation (PCO1 and PCO2) through time. The phylogeny used corresponds to a randomly selected most parsimonious tree (MPT) of the constrained analysis. Silhouettes correspond to the same groups in *Figure 1*. JME: Jurassic Morphospace Expansion; KTR: Cretaceous Terrestrial Revolution. For additional plots of morphospace, see *Figure 3—figure supplements 1–5*, and *Supplementary files 1-5*.

The online version of this article includes the following figure supplement(s) for figure 3:

**Figure supplement 1.** Lepidosaur morphospace for PCO1 and PCO2 showing abbreviated taxa labels and colors according to clades (see color legend in *Figure 1—figure supplement 2*).

**Figure supplement 2.** Lepidosaur morphospace for PCO3 and PCO4 showing abbreviated taxa labels and colors according to clades (see color legend in *Figure 1—figure supplement 2* and abbreviations in *Figure 3—figure supplement 1*).

**Figure supplement 3.** Lepidosaur morphospace for PCO5 and PCO6 showing abbreviated taxa labels and colors according to clades (see color legend in *Figure 1—figure supplement 2* and abbreviations in *Figure 3—figure supplement 1*).

**Figure supplement 4.** Lepidosaur morphospace for PCO1 and PCO2.

**Figure supplement 5.** Lepidosaur phylomorphospace in 2D for PCO1 and PCO2.

**Figure supplement 6.** Plot of morphospace and clusters after applying the pamk function of the fpc package.

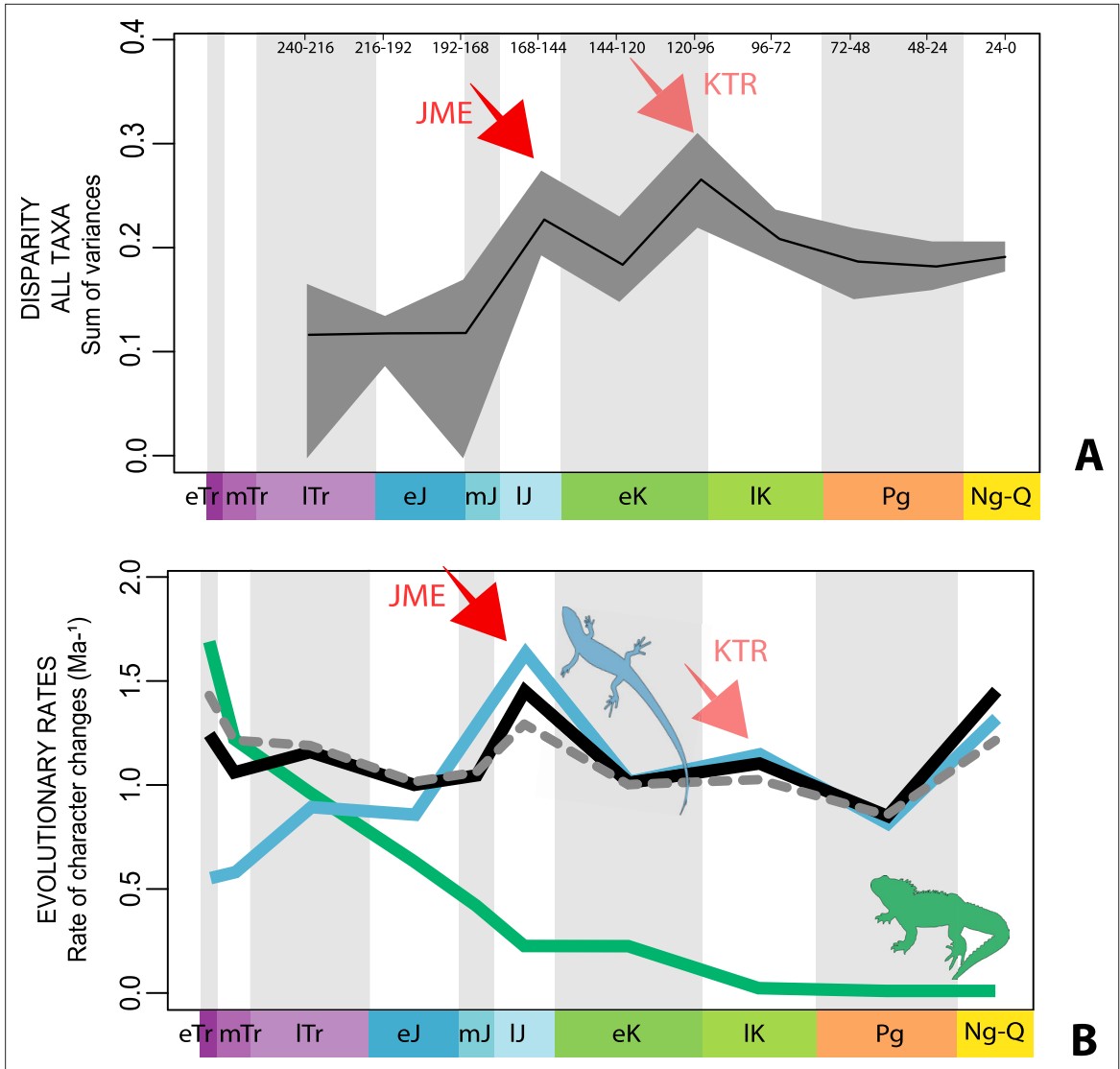

**Figure 4.** Disparity and evolutionary rates through time. (**A**) Temporal disparity patterns (bootstrapped and rarefied within bin sum of variances for all axes). For additional plots of disparity patterns, see *Figure 4—figure supplements 1–4*. (**B**) Evolutionary rates through time in epoch-scale bins. Black solid line corresponds to results for all taxa, blue solid line for lizards, and green solid line for rhynchocephalians plus stem lepidosaurs (all according to the constrained phylogeny). Dashed gray line corresponds to results for all taxa and unconstrained phylogeny. The curves represent averages from 25 iterations of each analysis using randomly selected trees dated with the Hedman method. For additional plots of evolutionary rates, see *Figure 4— figure supplements 5–22*. JME: Jurassic Morphospace Expansion; KTR: Cretaceous Terrestrial Revolution; ETr: Early Triassic; MTr: Middle Triassic; LTr: Late Triassic; EJ: Early Jurassic; MJ: Middle Jurassic; LJ: Late Jurassic; EK: Early Cretaceous; LK: Late Cretaceous; Pg: Paleogene; Ng-Q: Neogene-Quaternary.

The online version of this article includes the following figure supplement(s) for figure 4:

**Figure supplement 1.** Disparity (weighted mean pairwise distance) by high taxonomical group.

**Figure supplement 2.** Disparity (weighted mean pairwise distance) by low taxonomical group.

**Figure supplement 3.** Disparity (weighted mean pairwise distance) through time for lepidosaurs and time scheme 1.

**Figure supplement 4.** Disparity (weighted mean pairwise distance) through time for lepidosaurs and time scheme 2.

**Figure supplement 5.** Evolutionary rates for lepidosaurs according to the constrained phylogeny as represented by five most parsimonious trees (MPT) dated five times using the Hedman method.

**Figure supplement 6.** Evolutionary rates for lepidosaurs according to the constrained phylogeny as represented by five most parsimonious trees (MPTs) dated five times using the equal method.

*Figure 4 continued on next page*

*Figure 4 continued*

**Figure supplement 7.** Evolutionary rates for lepidosaurs according to the constrained phylogeny as represented by five most parsimonious trees (MPTs) dated five times using the MBL method.

**Figure supplement 8.** Evolutionary rates for squamates according to the constrained phylogeny as represented by five most parsimonious trees (MPTs) dated five times using the Hedman method.

**Figure supplement 9.** Evolutionary rates for squamates according to the constrained phylogeny as represented by five most parsimonious trees (MPTs) dated five times using the equal method.

**Figure supplement 10.** Evolutionary rates for squamates according to the constrained phylogeny as represented by five most parsimonious trees (MPTs) dated five times using the MBL method.

**Figure supplement 11.** Evolutionary rates for stem lepidosaurs plus rhynchocephalians according to the constrained phylogeny as represented by five most parsimonious trees (MPTs) dated five times using the Hedman method.

**Figure supplement 12.** Evolutionary rates for stem lepidosaurs plus rhynchocephalians according to the constrained phylogeny as represented by five most parsimonious trees (MPTs) dated five times using the equal method.

**Figure supplement 13.** Evolutionary rates for stem lepidosaurs plus rhynchocephalians according to the constrained phylogeny as represented by five most parsimonious trees (MPTs) dated five times using the MBL method.

**Figure supplement 14.** Evolutionary rates for lepidosaurs according to the unconstrained phylogeny as represented by five most parsimonious trees (MPTs) dated five times using the Hedman method.

**Figure supplement 15.** Evolutionary rates for lepidosaurs according to the unconstrained phylogeny as represented by five most parsimonious trees (MPTs) dated five times using the equal method.

**Figure supplement 16.** Evolutionary rates for lepidosaurs according to the unconstrained phylogeny as represented by five most parsimonious trees (MPTs) dated five times using the MBL method.

**Figure supplement 17.** Evolutionary rates for squamates according to the unconstrained phylogeny as represented by five most parsimonious trees (MPTs) dated five times using the Hedman method.

**Figure supplement 18.** Evolutionary rates for squamates according to the unconstrained phylogeny as represented by five most parsimonious trees (MPTs) dated five times using the equal method.

**Figure supplement 19.** Evolutionary rates for squamates according to the unconstrained phylogeny as represented by five most parsimonious trees (MPTs) dated five times using the MBL method.

**Figure supplement 20.** Evolutionary rates for stem lepidosaurs and rhynchocephalians according to the unconstrained phylogeny as represented by five most parsimonious trees (MPTs) dated five times using the Hedman method.

**Figure supplement 21.** Evolutionary rates for stem lepidosaurs and rhynchocephalians according to the unconstrained phylogeny as represented by five most parsimonious trees (MPTs) dated five times using the equal method.

**Figure supplement 22.** Evolutionary rates for stem lepidosaurs and rhynchocephalians according to the unconstrained phylogeny as represented by five most parsimonious trees (MPTs) dated five times using the MBL method.

time. On the other hand, squamates show a further step-up in rates during the Neogene, the past 23 Myr. The peak in evolutionary rates towards the present is particularly acute when using the MBL method, in what we regard as an artifact due to the condensing of divergence dates towards the present. These evolutionary rates are not the result of choosing a specific phylogenetic context – the black line represents summed rates for all lepidosaurs using the phylogenomic constrained trees, and the dashed gray line the rates from unconstrained morphological trees, with iguanians and fosso-rial species in traditional positions. This alternative version yields a similar general result, where the Late Jurassic peak is again clearly recovered (see *Figure 4—figure supplements 14–22*). In both cases, most parsimonious trees (MPTs) were randomly selected and dated five times according to each of the three dating methods, and evolutionary rates were calculated for each of the resulting dated trees. The curve shown represents average evolutionary rates among those calculated using the randomly selected MPT's and the equal method (see *Figure 4—figure supplements 5–22* for results according to all methods). The Late Jurassic peak in evolutionary rates is thus also robust to changes in the particular (randomly selected) point inside the stratigraphical range of a given fossil, and to different methods of dating the trees, including 'equal' and 'Hedman' methodologies, but not the 'MBL' method (but see the discussion for a possible explanation).

## Discussion

In exploring the nature of the 'early burst' in squamate disparity, we wanted to understand how the different clades occupied morphospace. Such an early establishment of squamate morphospace has never been documented in the few studies dealing with early radiations of squamates (e.g., *Simões et al., 2020*). Our finding that morphospace dimensions were established as early as the Middle or Late Jurassic is a counterintuitive result because it substantially predates the apparent increase in species richness that is usually tied to the KTR. Also, the high morphological rates recovered by a similar approach in *Simões et al., 2020* applied to a different dataset of lepidosaurs plus a wide array of other diapsids (*Simões et al., 2018*) are in their case not correlated to a high disparity. In the latter study, morphospace was not plotted through time, so our time series (*Figure 2*) or chronophylomorphospace (*Figure 3D*) have no counterpart in their results.

It is also interesting to observe so little addition to squamate morphospace after its establishment; the great expansion in species numbers up to 10,000 today has happened partly by minor expansions of the total morphospace envelope, but mainly by packing ever more species inside the existing morphospace area. Although we would expect that a more fine-tuned grouping of ecomorphotypes would reveal minor changes in the specific portions of occupied morphospace, it is unlikely that such improvement in resolution would change the recovered outer limits of morphospace.

The total morphospace (*Figures 2 and 3A*) confirms the central location of stem lepidosaurs, and that rhynchocephalians and squamates explored distinct morphospace throughout their evolution (see their spread along PCO2). *Martínez et al., 2021* reported that rhynchocephalians seem to present a lower proportion of morphological traits not shared with other diapsids than squamates do, which they interpreted as supporting the traditional view of rhynchocephalians as retaining a more plesiomorphic morphotype than squamates. Although our dataset is not comparable in the sense that it does not contain such a wide array of non-lepidosaurian diapsids, our morphospace, showing stem lepidosaurs in an intermediate position between squamates and rhynchocephalians, does not seem to support a plesiomorphic morphology for the latter. Our results are in line with current reinterpretations of many supposedly plesiomorphic traits of rhynchocephalians as actually derived in the context of Lepidosauromorpha. Regarding the distribution of squamates groups in particular, in our 2D phylomorphospace (*Figure 3B*) the squamate clades that lie furthest from the centroid are anguimorphs and marine mosasaurs (forming cluster 3) as well as amphisbaenians (worm lizards), dibamids (blind skinks), and snakes (forming cluster 4, associated with the limbless morphotype). A more detailed distribution of clades through the morphospace is represented in *Figure 3—figure supplement 4*.

*Simões et al., 2020* reported a comparable morphospace for lepidosaurs, where rhynchocephalians and squamates do not overlap in morphospace, and squamates are mainly divided into limbed (lizards) and limbless (snakes and amphisbaenians) taxa. However, their large sample of non-lepidosaur diapsids forced a relatively low sample of lepidosaurs (in comparison to our study) that resulted in a loss of resolution for lepidosaur and squamate morphospace. It is worth noting that our morphospace is also similar to the one recovered by *Watanabe et al., 2019* as based on skull geometric morphometrics, in identifying a cluster of snakes (and other limbless taxa like amphisbaenians and dibamids), anguimorphs, and mosasaurs at another pole, and a poor differentiation between other lizard groups (what we regard as 'generalized lizards').

The 3D phylomorphospace (*Figure 3C*, *Supplementary files 4 and 5*) shows how additional variation along PCO3 reveals the separation of several lizard morphotypes. This morphospace is easier to interpret in the 3D interactive plot (*Supplementary file 5*), which more clearly separates amphisbaenians and dibamids from snakes, the latter occupying an intermediate position between the two former clades and anguimorphs along PCO3. The inclusion of PCO3 slightly improves separation among 'generalized lizards,' although some clades persist as mixed groups in morphospace (e.g., scincids and gekkotans). However, PCO3 separates limbed (e.g., *Eumeces* and *Acontias*) from limbless (*Typhlacontias* and *Feylinia*) scincids. There is also a superposition between iguanians and the portion of anguimorphs that is closer to the middle part PCO3, which seem to represent less specialized anguimorphs (e.g., *Elgaria*) than those at the edge (varanid-like forms). In our opinion, all available studies (including ours) fail in fine-tuning ecomorphological groups of squamates, hinting at a problem that seems to be shared to both types of source data (discrete morphological characters and morphometric data). Ours is, however, the first study to track lepidosauromorph morphospace changes through time.

The chronophylomorphospace (*Figure 3D*) highlights the Mid–Late Jurassic expansion of morphospace, linked with the first radiation of the novel 'snake-like' morphology (see also *Appendix 3—figure 1*), but also that of clearly predatory forms as represented by *Dorsetisaurus*. In our interpretation, the first event (JME) is tracking the initial radiation of total group Squamata, when the clade radiated into its main components, as revealed by the branching timing of the dated trees, the primary expansions seen in morphospace plots, the Late Jurassic peak in disparity, and the Late Jurassic peak in evolutionary rates. This is particularly true for the trees dated using the Hedman and equal methods, but in the case of the MBL method the signal is overprinted by the high concentration of short branches close to the present time, which result in artificially higher rates for that period, and relatively lower Mesozoic peaks. Besides the MJE, a second event, which fits well with the timing of the KTR, would be coincident with the radiation of the constituent crown groups of Squamata as revealed by a limited expansion and infilling of morphospace plus the record of coincident peaks of disparity and evolutionary rates. Whether this second event represents an actual event of diversification among squamates or is the result of a greatly improved fossil record remains unclear.

Our study offers a new perspective on the early evolution of the major clade Squamata and the other groups of lepidosauromorphs. It benefits from current phylogenomic evidence on phylogeny, as well as fossil data on the timings of events and the expansion of skeletal morphologies and disparity. Our results show that, although the first assemblages of lizards (and possibly snakes) in the Middle–Late Jurassic are not particularly diverse or abundant, the basic structure of the present morphospace distribution had already been achieved (*Figures 2 and 3D*, *Appendix 3—figure 1*). This is independent of the interpretation of the affinities of a given taxon because the points in morphospace do not change with changes in topology (only the branches uniting them in phylomorphospaces). Finding support for this early burst of disparity and associated rapid evolutionary rates was rather unexpected, especially so long before the KTR – a reported key driver of squamate evolution (*Gauthier et al., 2012*) – and before a good fossil record is documented.

Further, we confirm that these distributions in morphospace, marking broad ecological and functional groupings, were remarkably stable for the subsequent 150 Myr, through to the present day (*Figure 2*, *Figure 2—figure supplements 1–6*). In other words, the range of adaptations in the current huge diversity of squamate species tracks back very deep in Earth history, some 60 Myr before the KTR. The only observable changes from then on correspond to a slight expansion of the edges of the occupied morphospace, and a notable increase in the density of points filling this morphospace. We acknowledge, however, that the recovered structure represents a simplification that only corresponds to groups according to general bauplans (e.g., limbed vs. limbless morphotypes) and, to a lesser degree, adaptations achieved by specific clades like, for example, anguimorphs. It is thus not possible to track finer ecomorphologies like, for example, adaptations of snakes to different environments (marine, fossorial, or ground-dwelling), which would likely add some variability in the form of shifts in the occupied morphospace through time. Although this is possibly related, in part, to the use of a phylogenetic morphological matrix that was constructed to capture the deep phylogenetic relationships of the constituent groups inside Squamata, it is worth noting that our morphospace is not too different from one recovered from a geometric morphometrics approach (*Watanabe et al., 2019*), suggesting that this poor resolution is not entirely explained by this procedural choice.

As a final note on squamate morphospace distribution and evolutionary rates, the results presented here also differ from recently published studies dealing with dentition shape, jaw size disparity (as informed by geometric morphometrics *Herrera-Flores et al., 2021a*), and body size as a continuous character (*Herrera-Flores et al., 2021b*). In the latter, even though the divergence times for most clades were in line with our results (they applied the Hedman method to their dataset), their results on evolutionary rates greatly differ from the ones presented here. Our study here yields consistently higher rates for squamates than rhynchocephalians, when the opposite trend was recovered in reference (*Herrera-Flores et al., 2021b*). Moreover, our results show a trend of decreasing evolutionary rates for rhynchocephalians through time, whereas the opposite was recovered by *Herrera-Flores et al., 2021b* through the Mesozoic, with a marked increase across the Jurassic–Cretaceous boundary. We regard these striking differences as related to the radically different sources of information used in both studies. Although differences in body size can be used to track shifts in evolution and, accordingly, to hint at macroevolutionary patterns, they do not need to be necessarily related to the same processes explored here. Both mentioned studies (*Herrera-Flores et al., 2021b*; *Herrera-Flores*

*et al., 2021a*) dealt with particular aspects of lepidosaur evolution (body size and dentition/diet), whereas the results presented here are derived from an approach that considered many different ecomorphological aspects (as many as can be reflected in a morphological matrix that includes osteological characters for the entire skeleton, as well as soft-tissue characters). In addition, the focus on Mesozoic taxa in the aforementioned studies make the results obtained here, which include a good representation of extant taxa, difficult to compare.

It is important to consider whether results could represent bias in the fossil record. It is well understood that the Mesozoic fossil record of squamates is patchy, including some very poorly sampled time intervals (*Cleary et al., 2018*; *Evans, 2003*). As a counter to this concern, we note that the occupied total squamate morphospace in the Middle–Late Jurassic is just slightly smaller than that for the Late Cretaceous or the Paleogene to extant time bin (*Figure 2*), even though the two latter have yielded much higher sample sizes of specimens (*Appendix 1—figure 5*) that are also anatomically more complete (*Appendix 1—figures 1–4*). In particular, note that the morphospaces through geological time, from the Middle Jurassic onwards, are not much smaller than the morphospaces occupied by the represented sample among 10,000-strong extant squamates. Therefore, we have either identified more or less the correct extent of morphospace for the Middle to Late Jurassic, despite the poor fossil record at that time, or that with much richer finds from that time interval occupied morphospace was even larger than we identify here. This would then enhance our interpretation of an early burst in squamate morphology and function. We posit that our conclusions regarding morphospace expansion are not affected by sampling; we predict that new fossil finds in the future will mostly fit inside the demarcated area of occupied morphospace.

Our interpretation here of the long-term stable morphospace occupation by squamates is compelling because the apparent increase in species richness through time, even if partially influenced by bias in the fossil record (*Cleary et al., 2018*), is not linked to a great increase in occupied morphospace. The observed expansion in occupied morphospace and rapid evolutionary rates coincides not only with the first presence of squamates in the fossil record, but also with the time when many crown groups are first recorded (e.g., scincoids, anguimorphs, and likely gekkotans and snakes), in the Middle–Late Jurassic (*Evans, 2003*; *Caldwell et al., 2015*; *Estes, 1983*; *Evans, 2008*; *Figure 1*). The Jurassic expansion of squamates is further supported by (1) the fact that all three main squamate morphological groups in morphospace (clusters 2–4) are already present in the Middle–Late Jurassic bin (*Figure 2*); (2) bootstrapped and rarefied measures of disparity through time (*Figure 4A*) present a peak roughly corresponding to the Late Jurassic for all lepidosaurs and for squamates alone; and (3) the evolutionary rates calculations also show peak in the Late Jurassic (*Figure 4B*).

This explosive adaptive radiation of squamates in the Middle–Late Jurassic situates the dates of origin of major clades in line with current phylogenomic analyses (*Mulcahy et al., 2012*; *Zheng and Wiens, 2016*; *Pyron, 2017*; *Burbrink et al., 2020*), mostly into the Jurassic, except for some groups in the Hedman trees, the divergence ages of which are even older, placed in the Triassic. It is worth noting that other key tetrapod groups also radiated ecologically in the Jurassic, namely, illustrated by the rapid diversification of paravians, the clade including birds and related small, feathered theropods with elongate wing-like arms (*Lee et al., 2014*; *Puttick et al., 2014*; *Brusatte et al., 2015*) and the expansion of early mammalian clades (*Close et al., 2015*). This predates the second ecological expansion of these three major clades, accounting for more than 95% of the modern biodiversity of tetrapods, which happened in the mid-Cretaceous in association with the KTR (*Lloyd et al., 2008*; *Doyle, 2008*; *Benton, 2010*; *Meredith et al., 2011*; *Cardinal and Danforth, 2013*), when diversity, and probably also abundance, exploded in line with the new food resources on land. Squamates remained at low diversity through the Triassic (where unambiguous fossils are yet to be recovered, but supposed to be present), Jurassic, and Early Cretaceous (*Jones et al., 2013*; *Caldwell et al., 2015*; *Evans, 2008*), and species richness seems to have risen massively during the KTR some 100 Ma, but the morphological expansion had already happened some 60 Myr earlier, in the Middle to Late Jurassic. This seems to fit an already identified pattern where the main diapsid groups present a long chronological lag between the initial phenotypic radiation of the group and its subsequent taxonomic diversification (*Simões et al., 2020*; *Close et al., 2019*) or, alternatively, it is related to a failure of an impoverished fossil record to reveal the true diversity achieved in pre-Late Cretaceous times.

What was happening in the Middle Jurassic that could have triggered squamate morphological diversity? (1) The supercontinent Pangaea began to split into precursors of the modern continents;

(2) temperatures rose sharply for a short time; (3) gymnosperm plants diversified; and (4) various insect groups (e.g., mayflies, crickets, cockroaches, bugs, cicadas) diversified. All these factors may have had a role in driving some aspects of the early burst of squamate disparity, and they all require further investigation. This early radiation of squamates had been previously inferred (*Evans, 1998*; *Evans, 2003*) on the basis of a crude interpretation of the fossil record and tree topologies. However, it is the first time this issue has been approached with quantitative methods involving such an array of diverse points of view (phylogeny, dating, fossil record, morphospace, disparity, and evolutionary rates). Moreover, most of the Jurassic forms are very difficult to classify, and many of them have been reinterpreted since this was proposed (e.g., *Marmoretta*, see *Griffiths et al., 2021*). Our methods, however, do not necessarily rely on the achieved identification of each fossil because they feed on the morphological information stored in the character matrix, and not the specific topology derived from its analysis. Thus, the method used accounts for possible shifts in the phylogenetic position recovered for each form.

In their later evolution in the Mesozoic, all living clades of squamates diversified rapidly through the KTR. In addition, new and short-lived squamate groups arose in the Late Cretaceous, such as the terrestrial borioteiioids and the marine mosasaurs and relatives, but they disappeared, together with non-avian dinosaurs and other groups of diapsids at the end of the Cretaceous. The non-survival of such groups emphasizes the importance of the origin of the key modern clades in the Middle Jurassic and the establishment of their key ecomorphological adaptations – these then proved robust to various crises, including the end-Cretaceous mass extinction. Our integrative study here, incorporating current phylogenomic analyses of relationships of squamate clades with current fossil data, and novel computational methods in disparity and evolutionary rates, provides a synthetic narrative of the origin of one-third of modern tetrapod biodiversity, the Squamata.

Although morphological matrices might not be ideal for macroevolutionary inferences because they were specifically built for inferring phylogenetic relationships, they are handy in that they represent readily available sources of information, and they allow the mixture of taxa for which ecology is known (extant taxa) and taxa for which it can only be inferred (fossils). Moreover, results actually show that some ecomorphological signal is present in such datasets. Although the recovery of a limbless cluster of taxa might seem trivial, in fact it shows that the ecomorphological signal is overprinting the phylogenetic signal in that case because otherwise snakes and amphisbaenians would cluster with their respective closer clades (anguimorphs and iguanians for the former, lacertids for the latter). There are many other examples of this convergence in morphospace that can be interpreted as related to ecological niche convergence, for example, xantusiids and gekkotans, two groups that are not closely related phylogenetically but greatly overlap in our morphospace. The tight clustering of multiple groups close to the centroid does not help in interpretation, but the overall morphospace distribution shares many similarities with the niche plots reported by reference (*Pianka et al., 2017*) according to extant taxa scored for five niche dimensions. Their *Figure 4* perfectly shows that a mixture of niche conservatism (phylogenetically close taxa tend to occupy similar niches) and niche convergence (distantly related species with similar ecomorphology tend to cluster together) occurs. Although represented taxa are not completely comparable (Pianka et al.'s dataset includes only lizards, lacking snakes and amphisbaenians among squamates, and also rhynchocephalians), and even the lizards sampled are different at the genus or species levels, similarities between our morphospace plot and their niche plot for extant groups include (1) anguimorphans are in both cases the most differentiated group, far from a much more populated cluster of taxa around the centroid that includes most of the rest of lizards; (2) this centered cluster includes scincoids, lacertoids, gekkotans, and some iguanians in our plot, whereas in Pianka's plot gekkotans and most iguanians overlap outside this cluster, in the opposite direction of anguimorphans along PC1; and (3) small teiids overlap lacertids in both cases, but large teiids (*Dracaena* and *Tupinambis* in our case, *Tupinambis* in the Pianka et al.'s plot) are closer to anguimorphs.

Macroevolutionary studies can be strongly influenced by an array of potential biases that sometimes compromise results to variable degrees. Several potential issues have been identified through the design, development, and review of this study, ranging from sampling to methodological and interpretative factors. Moreover, methods are quickly evolving and can be quickly displaced by more refined approaches or criticized in their use or misuse. We have made an effort to consider as many variables as possible by assessing multiple potential resolutions for the phylogenies (constrained

vs. unconstrained), specific changes in topology (by randomly selecting multiple MPTs), dating (by using three different methods, and randomly dating each fossil tip multiple times, accounting for geological range uncertainty), and by using multiple metrics and time bins when necessary. Other factors, like the possibility that our results are biased by the nature of the fossil record and how it conditions effective sampling across different time bins, are difficult to circumvent. We think, however, that if the poor fossil record is affecting results, it is most probably undermining the effect of the JME because (1) we have not been able to include any of the known Middle Jurassic squamate fossils and (2) the Late Jurassic contains a low number of samples compared to the Early and Late Cretaceous.

Among other studies that have emphasized the importance of the KTR in squamate evolution is *Lafuma et al., 2021*. In a study of origins and losses of tooth complexity across the clade, they found that tooth complexity first increased in the Late Jurassic, although it is regarded as marginal until the KTR. This increase in tooth complexity is apparent when its distribution through the Jurassic and Cretaceous is analyzed, but the possibility that the change in the quality of the fossil record might be enhancing the much more complete sample occurring in the Late Cretaceous is not discussed. Further, lumping the diverse morphologies of unicuspid teeth into a single category is potentially problematic if carnivores and insectivores are to be considered as distinct styles of predators. Another interesting but ignored result of that study is that, besides presenting a Cretaceous turnover (speciation/extinction) peak coinciding with the KTR, there is a previous peak mostly coinciding with the Jurassic–Cretaceous boundary. In any case, it seems that there is a shared pattern to our results, where innovations are initially explored in the Jurassic and then fully exploited in the Cretaceous, coinciding with the KTR.

To the uncertainty generated by the incomplete fossils that can only be scored for a minor portion of the morphological characters, we face the added problem of unscorable characters. In a morphological matrix of characters for lepidosaurs, this is not a minor issue because there is a long list of characters that cannot be scored for multiple groups, for example, characters related to limbs in limbless taxa, or characters related to structures only found in the snake skull. Even though the use of inapplicable characters has been discouraged (*Gerber and Ruta, 2019*), we think that simply removing them from the analyses is not the best solution, just as it would not be for a phylogenetic analysis.

Finally, results that directly depend on the estimation of time-calibrated branch lengths, such as the calculation of evolutionary rates, should be treated as preliminary because they must be validated under the use of more robust methods of time calibration, such as those that incorporate molecular data alongside the fossil record, as well as employ more realistic models of diversification such as the fossilized birth–death prior. In this sense, an ongoing study (work in progress) aims to analyze the present matrix and other datasets by using Bayesian tip dating under relaxed morphological clocks as described in *Zhang and Wang, 2019*. This allows us to calculate phylogeny and estimate divergence times and evolutionary rates while accounting for their uncertainties, and allow the use of both morphological and combined (morphological plus molecular data) matrices. This additional study should help clarify if the signal recovered in this work is reliable, or on the contrary it is biased by the chosen methodology. Meanwhile, the results presented here question the alternative view that regards the great diversification of squamates as occurring in the mid-Cretaceous, coinciding with the KTR. The first half of the Mesozoic has a great potential for unveiling the key milestones in the evolutionary history of lepidosauromorphs in general, but also of squamates in particular. Current reanalyses of classic material and the description of new specimens and taxa are already displacing the focus from the Late Cretaceous to the first half of the Mesozoic and are expected to provide insights on the issue presented here.

## Materials and methods
### Taxa and character data
The data source for all morphological character and taxon data analyses is the morphological data matrix of *Conrad, 2018*, reduced in our study to 201 species of living and fossil lepidosaurs scored for 836 skeletal morphological characters. We used this data matrix because it is by far the most extensive in terms of taxa and characters.

## Phylogeny and timescaling

Phylogenetic analyses were performed in TNT 1.5 (*Goloboff and Catalano, 2016*). The settings for the unconstrained analysis are the same as in the original publication (*Conrad, 2018*) (ratchet and drift options activated, except that we set analyses to 100 replicates instead of 200). An alternative version of the phylogeny was obtained after constraining the general relationships recovered in molecular studies for those groups that present discrepancies in their position in morphological analyses, among others the sister group relationship of Iguania to the rest of crown squamates and the grouping of limb-reduced and limbless forms in the called 'fossorial' group (including dibamids, snakes, amphisbaenians, and limbless skinks), which is the result of convergences and clearly do not form a monophyletic group. For this, we randomly chose one of the MPTs recovered in the first analysis and forced the topology of phylogenomic studies for extant clades by defining the monophyly of the main extant groups according to *Pyron et al., 2013*. We set up fossils, which account for more than half of the taxa comprising the matrix, as floaters, so they could freely move around the tree. In both cases (constrained and unconstrained analysis), the resulting MPTs were exported to PAUP (*Howard et al., 2002*), where consensus trees were calculated. We produced the time trees for illustration (*Figure 1*, *Figure 1—figure supplement 1*) and rates calculations using fossil data to date origins of clades and time calibrated the trees in Paleotree v. 3.3.0 (*Bapst, 2012*) and using the Hedman method (see below).

## Morphological disparity

All disparity and macroevolutionary analyses were performed in R (*R Development Core Team, 2013*). For disparity analyses, the pipeline started with the calculation of a pairwise morphological distance from the original character data using the package Claddis and maximum observable rescaled distances (MORD; *Lloyd, 2016*). The pairwise distances data was then subject to principal coordinates analysis (PCO) to identify the major axes of morphological variation. The resulting ordination matrix was used to plot morphospace based on PCOs 1–3. This morphospace was combined with a single topology (dated using the same method) to illustrate phylomorphospace and a chronophylomorphospace. We also plotted morphospace occupation in temporal bins. Finally, we used both pre-ordination (weighted mean pairwise distance, WMPD) and post-ordination (SoV, calculated in DispRity, *Guillerme and Poisot, 2018*) metrics to calculate global disparity, disparity in specific groups, as well as disparity through time. We also calculated completeness and sampling across the different time bins for comparisons with disparity results. We used various packages in R for plotting, namely, Plotly (*Sievert, 2018*), ggplot (*Wickham, 2016*), Geomorph (*Adams et al., 2019*), Claddis (*Lloyd, 2016*), and Phytools (*Revell, 2012*).

## Morphological evolutionary rates

Rates of morphological evolution were analyzed using maximum-likelihood methods applied to the discrete skeletal character dataset and a range of phylogenetic trees. We used the DiscreteCharacterRate function from the R package Claddis and ran calculations for five of the unconstrained MPTs and five of the constrained MPTs, separately. We used a modified version of the code from *Moon and Stubbs, 2020*. The methodology first seeks to identify rate heterogeneity across the whole tree and then highlights branches or temporal bins with significant rate deviations (notably fast or slow) using likelihood ratio tests (*Lloyd, 2016*). To ensure rate results are consistent, the different topologies were dated multiple times (in our case, five dating replicates for each of the five randomly selected trees, for both unconstrained and constrained trees). We also repeated this for three dating methodologies, using the 'equal' method (*Brusatte et al., 2008*), 'minimum branch length' approach (*Laurin, 2004*), using the R functions from *Lloyd et al., 2016* and a whole-tree extension of the Bayesian Hedman algorithm (*Hedman, 2010*). The Hedman node-dating approach uses Bayesian statistics, incorporating probability distribution constraints based on successive outgroup taxa ages (*Hedman, 2010*). We calculated per-bin evolutionary rates in two sets of time bins, one corresponding to geological stages and one corresponding to equal 10 Myr bins. To illustrate the rates results, we use 'spaghetti plots' showing individual lines for each combination of tree and dating (25 individual lines), as well as an average line, and also highlighting iterations and bins with significantly fast and slow evolutionary rates (*Figure 4—figure supplements 5–22*). In the main *Figure 4B*, we present summaries of these analyses.

## Plots of Cramér coefficients

We used Cramér coefficients (*Appendix 1—figure 7*, *Appendix 2—figure 2*) to show correspondence between characters and PCO axes (*Kotrc and Knoll, 2015*; *Nordén et al., 2018*). See Appendix 4 for more details.

R scripts are available as *Source code 1* (for morphospace and disparity), *Source code 2* (evolutionary rates), *Source code 3* (morphospace clusters), and *Source code 4* for alternative analysis without integument and Cramér values, and *Source code 5* for plotting clade-colored consensus trees.

## Acknowledgements

We thank Graeme Lloyd, Benjamin Moon, Armin Elsler, and Guillermo Navalón for help with the scripts and discussion. Funding for AB comes from a Newton International Fellowship (NF170464) and a Juan de la Cierva Incorporación fellowship (IJC2018-037685-I) of the Spanish Government, and by the CERCA Programme of the Generalitat de Catalunya. This is part of the ERC Innovation Advanced Grant to MJB. (ERC 788203). We thank Min Zhu, George Perry, and two anonymous reviewers for comments on an early version of this manuscript.

## Additional information

### Funding

| Funder | Grant reference number | Author |
|---|---|---|
| Royal Society | NF170464 | Arnau Bolet |
| Ministerio de Ciencia, Innovación y Universidades | IJC2018-037685-I | Arnau Bolet |
| European Commission | ERC 788203 | Michael Benton |
| Natural Environment Research Council | NE/I027630/1 | Michael J Benton |

The funders had no role in study design, data collection and interpretation, or the decision to submit the work for publication.

### Author contributions

Arnau Bolet, Conceptualization, Data curation, Formal analysis, Funding acquisition, Investigation, Methodology, Project administration, Visualization, Writing - original draft, Writing – review and editing; Thomas L Stubbs, Methodology, Validation, Writing – review and editing; Jorge A Herrera-Flores, Michael J Benton, Conceptualization, Validation, Writing – review and editing

### Author ORCIDs

Arnau Bolet ⓘ http://orcid.org/0000-0003-4416-4560
Thomas L Stubbs ⓘ http://orcid.org/0000-0001-7358-1051
Jorge A Herrera-Flores ⓘ http://orcid.org/0000-0002-9660-4161
Michael J Benton ⓘ http://orcid.org/0000-0002-4323-1824

### Decision letter and Author response

Decision letter https://doi.org/10.7554/eLife.66511.sa1
Author response https://doi.org/10.7554/eLife.66511.sa2

## Additional files

### Supplementary files

• Supplementary file 1. Interactive plot of 3D morphospace (PCO1, PCO2, and PCO3), colors according to high taxonomical groups.

• Supplementary file 2. Interactive plot of 3D morphospace (PCO1, PCO2, and PCO3), colors

according to low taxonomical groups.

• Supplementary file 3. Interactive plot of 3D morphospace (PCO1, PCO2, and PCO3), colors according to time bins.

• Supplementary file 4. Interactive plot of 3D phylomorphospace (PCO1, PCO2, and PCO3), colors according to high taxonomical groups.

• Supplementary file 5. Interactive plot of 3D phylomorphospace (PCO1, PCO2, and PCO3), colors according to low taxonomical groups.

• Supplementary file 6. Interactive plot of squamate 3D morphospace (PCO1, PCO2, and PCO3). Black spheres correspond to toxicoferans, white spheres correspond to non-toxicoferans.

• Transparent reporting form

• Source code 1. R script and source files for morphospace and disparity analyses.

• Source code 2. R script and source files for evolutionary rate analyses.

• Source code 3. R script and source files for plotting morphospace clusters.

• Source code 4. R script and source files for plotting Crámer values.

• Source code 5. R script and source files for plotting consensus trees.

### Data availability

All data generated or analysed during this study are included in the manuscript and supporting files. More specifically, all necessary files (Nexus files and source tables) are included in the Source code 1–4.

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

# Appendix 1

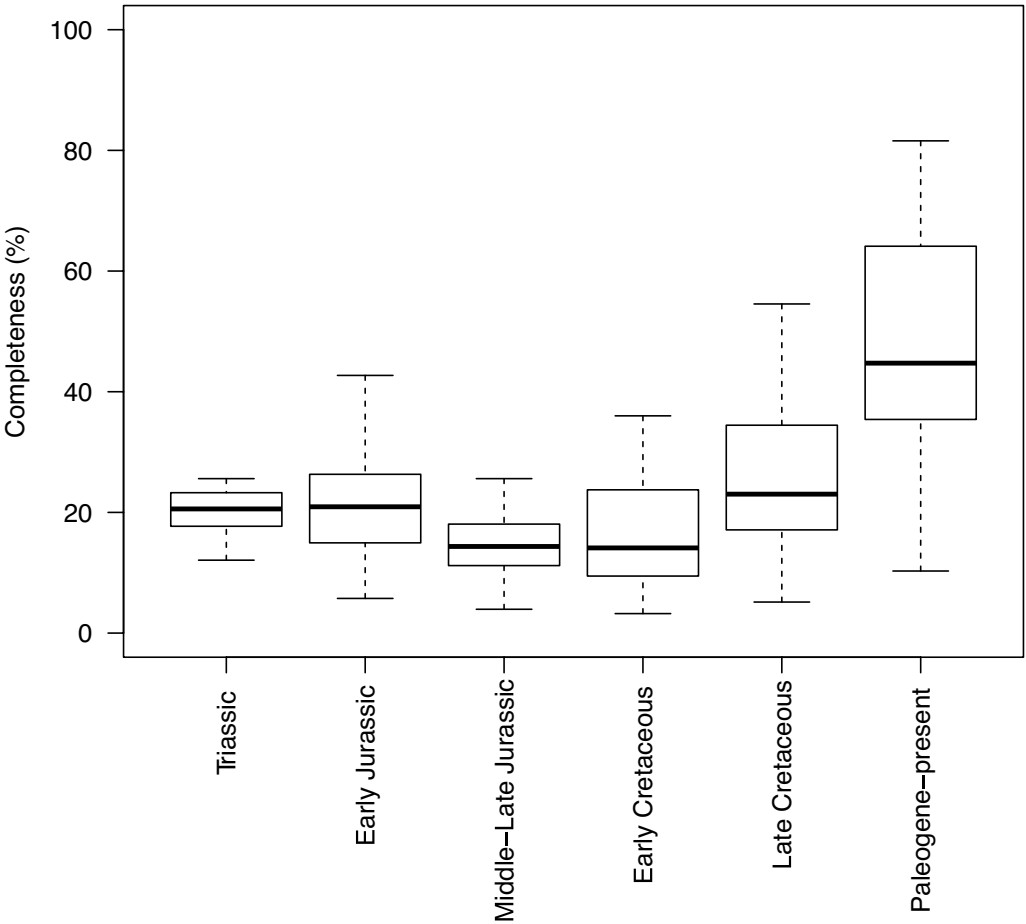

**Appendix 1—figure 1.** Completeness percentage by time bin for the complete dataset (all lepidosaurs and stem), for time scheme 1.

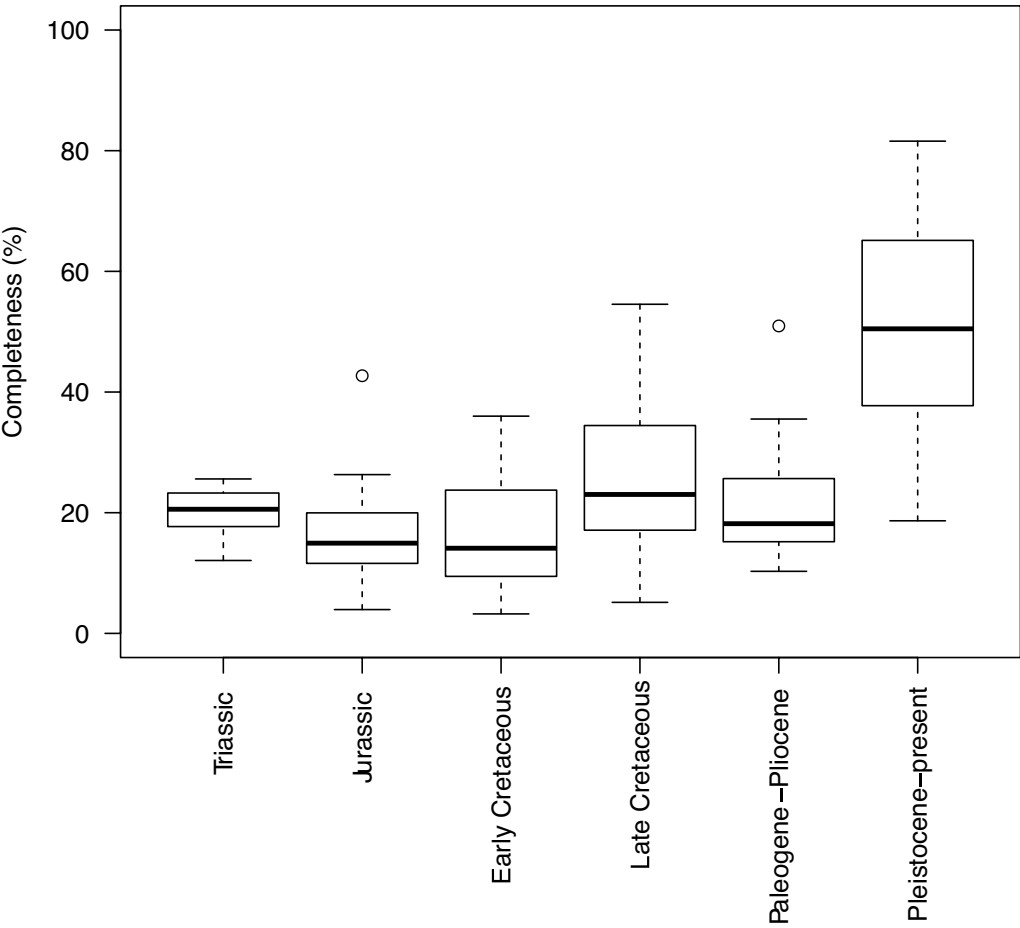

**Appendix 1—figure 2.** Completeness percentage by time bin for the complete dataset (all lepidosaurs and stem) for time scheme 2.

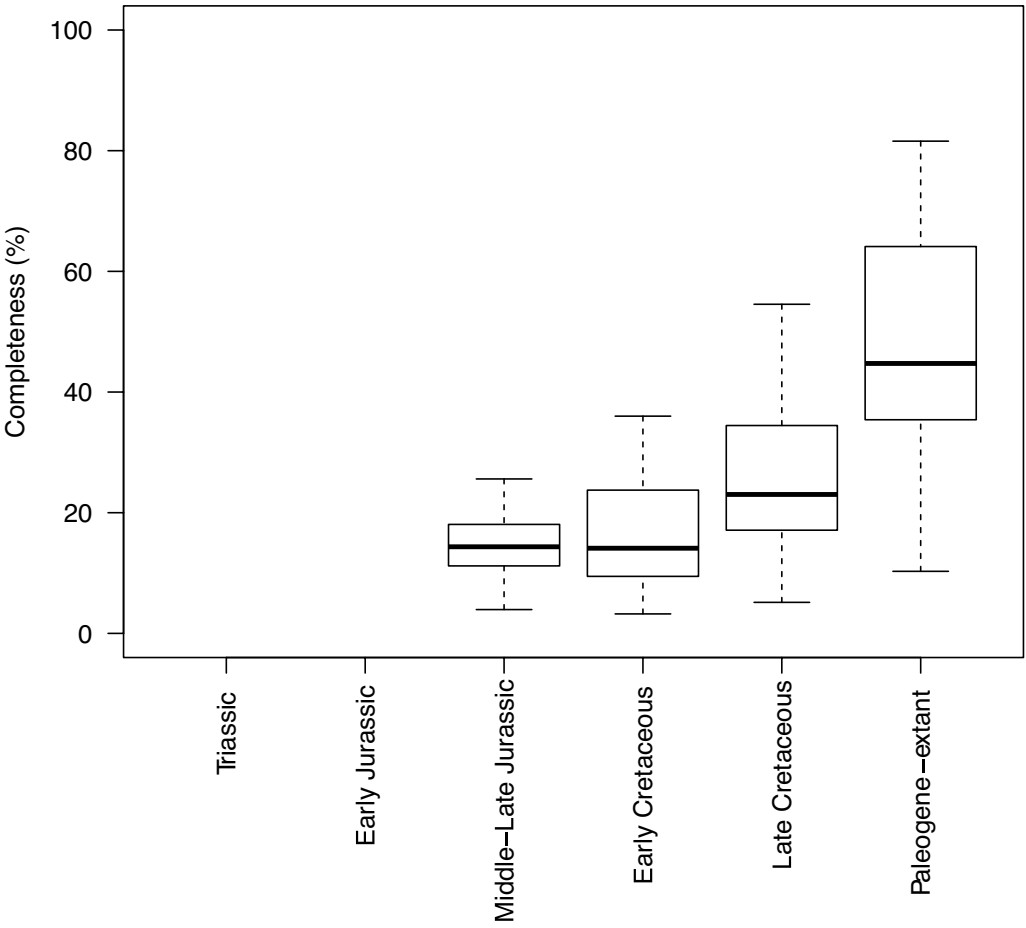

**Appendix 1—figure 3.** Completeness percentage by time bin for squamates and time scheme 1.

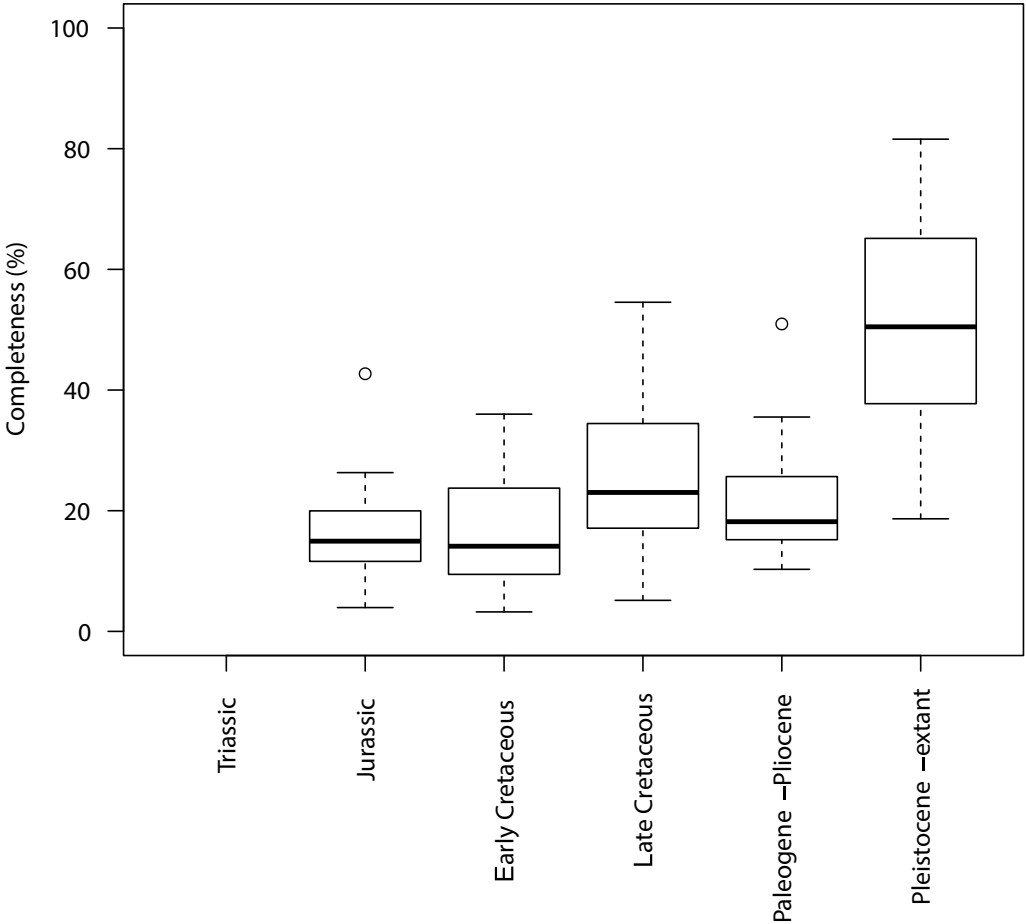

**Appendix 1—figure 4.** Completeness percentage by time bin for squamates and time scheme 2. N/A

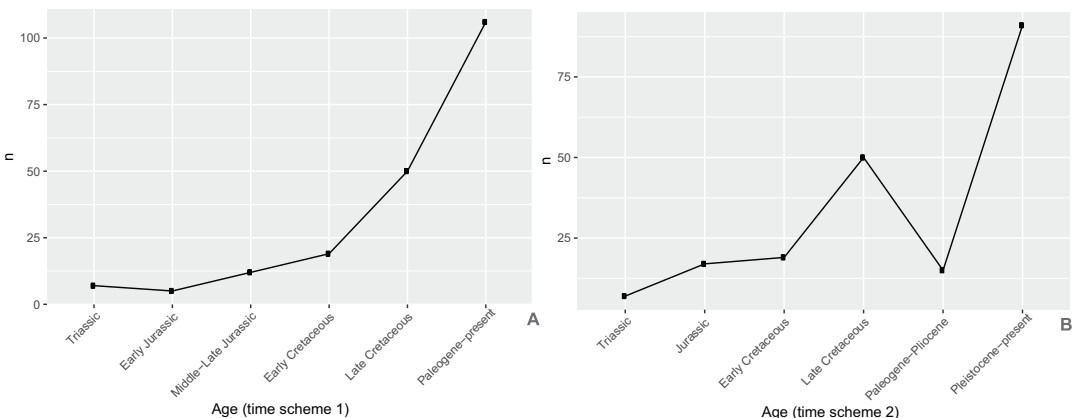

**Appendix 1—figure 5.** Plot of the number of taxa sampled for each time bin in (**A**) time scheme 1 and (**B**) time scheme 2.

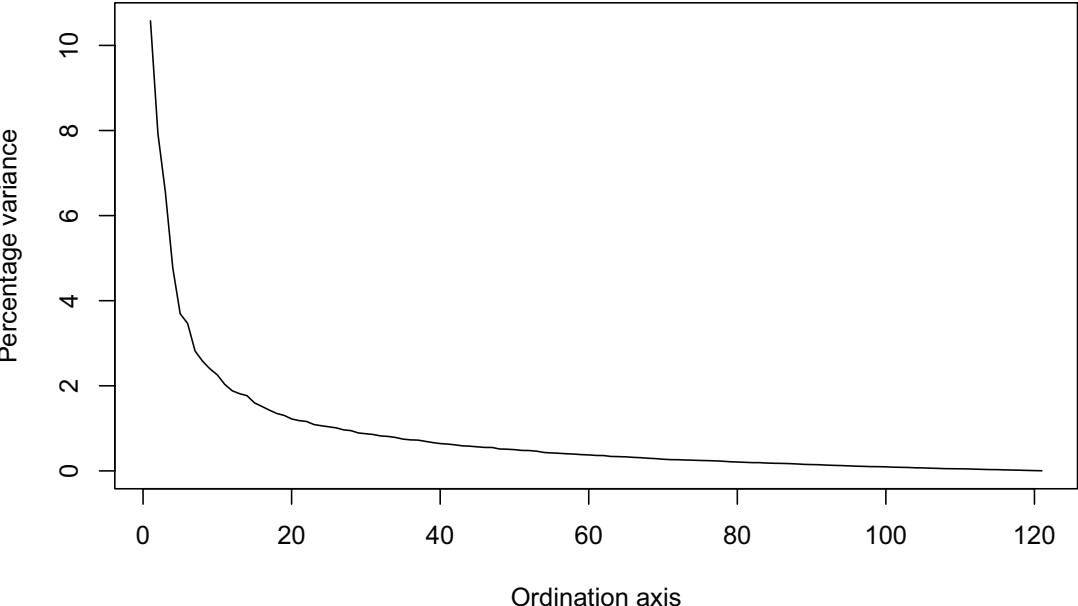

**Appendix 1—figure 6.** Percentage of variance explained by each axis.

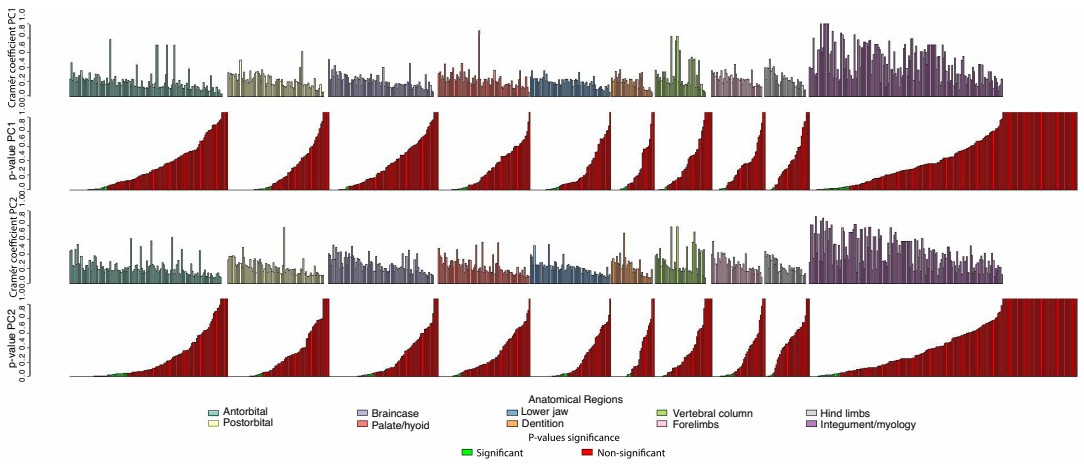

**Appendix 1—figure 7.** Plot of Cramér coefficients for PCO1 and PCO2 for the full dataset. Characters do not appear in the original order because they have been grouped by anatomical regions. Inside each region, characters on the left are significant (according to their p-values, in green when significant, in red when not significant).

## Appendix 2

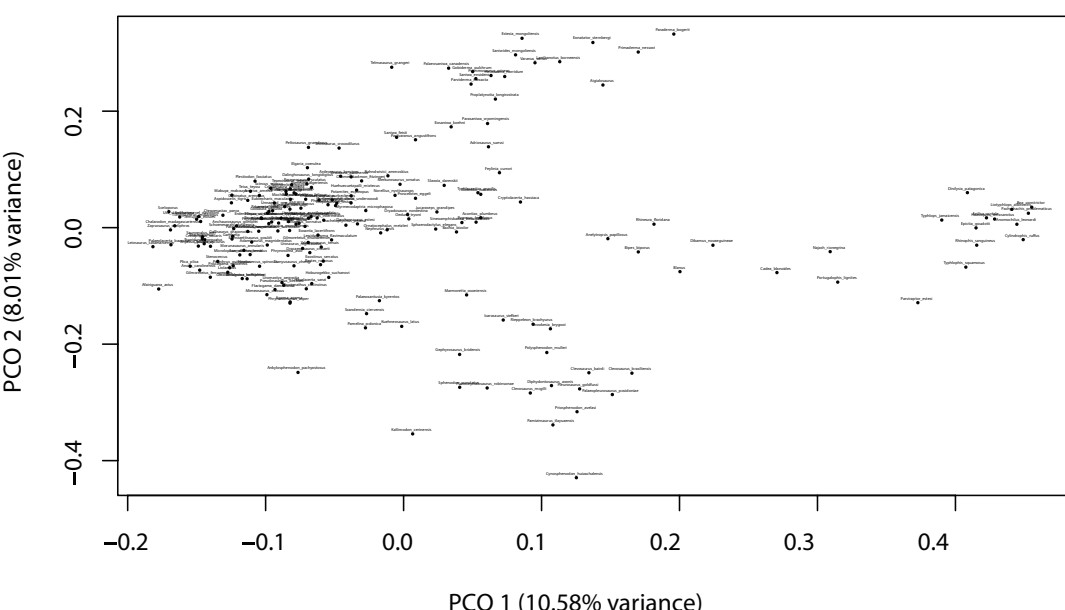

**Appendix 2—figure 1.** Morphospace (PCO1 and PCO2) when myological/integument (except for osteoderm characters) are deactivated.

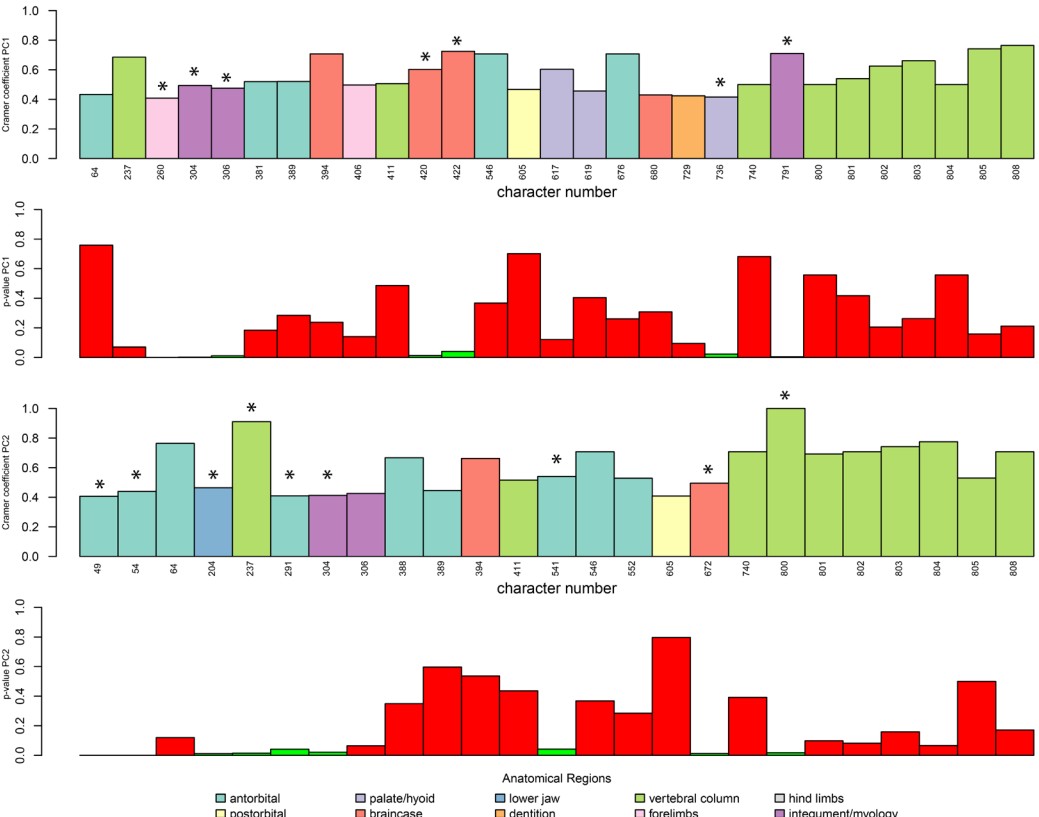

**Appendix 2—figure 2.** Plot of Cramér coefficients for PCO1 and PCO2 for the dataset without myological/ integument (except for osteoderm characters), limited to characters with a coefficient <0.4. Character numbers correspond to those of the original morphological matrix. Significant characters are marked with an asterisk.

## Appendix 3

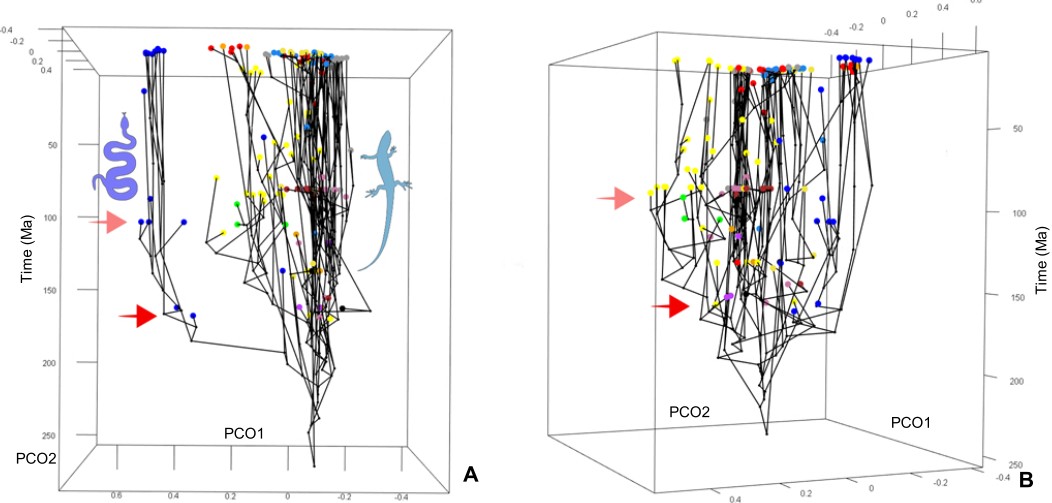

**Appendix 3—figure 1.** Chronophylomorphospace for squamates. Screenshot of the interactive plot in two (**A**, **B**) views. Colors correspond to low-level taxonomical groups. The solid red arrow points to the morphospace expansion occurring by the Middle–Late Jurassic, and the light red arrow to a minor event occurring around the middle–Late Cretaceous boundary.

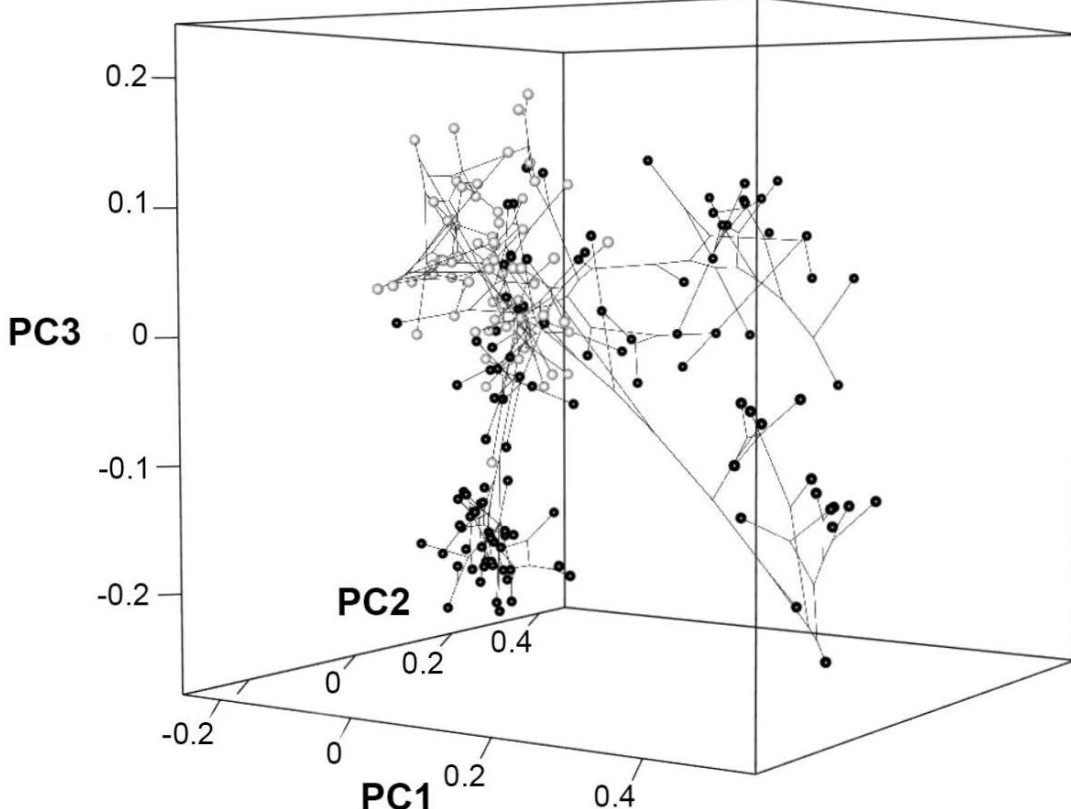

**Appendix 3—figure 2.** Phylomorphospace in 3D for PCO1, PCO2, and PCO3 for squamates only. Screenshot of the interactive plot, where black spheres correspond to toxicoferan squamates and white spheres correspond to non-toxicoferan squamates.

## Appendix 4

### Extended methods

The morphological matrix (**Conrad, 2018**) used for phylogenetic analyses included in this work feeds upon previously published matrices for osteological characters (**Conrad, 2008**; **Conrad et al., 2012**; **Smith, 2009**; **Gauthier et al., 2012**), soft tissue (**Schwenk, 1988**), and salivary compounds (**Fry et al., 2006**). A few preliminary runs of disparity and evolutionary rates in alternative matrices (e.g. **Simões et al., 2018**) revealed that the results (not shown) were compromised by the poor sampling of squamates at key time bins (e.g., Late Jurassic), and we abandoned their use. Moreover, during the process of review of our article, **Simões et al., 2020** published their own study on evolutionary rates and disparity in reptiles based on their own matrix (2018), and an additional work (**Martínez et al., 2021**) deals with the description a new stem lepidosaur and its position in morphospace. Results of those studies are compared to our own in the results section below. Regarding the final version of the matrix (**Conrad, 2018**) used here, it should be noted that running the original file as it was published does not yield the results reported in the corresponding paper. One problem relates to the comment in the text (p. 618, line 6) regarding deactivating characters 236, 242, and 364. This should not be done because these characters were evidently removed from the final matrix, and this is the reason why it has only 836 characters instead of the 839 stated in the text. We also decided to delete a few taxa for various reasons. Some terminal taxa correspond to additional specimens of a same taxon (e.g., *Slavoia*, *Globaura*, or *Eoxanta*). Others, like the fragmentary *Tikiguania* (**Datta and Ray, 2006**), are no longer relevant since it has been reinterpreted as a non-Triassic (probably Quaternary) agamid lizard (**Hutchinson et al., 2012**), and in any case it is not particularly complete or informative. Regarding *Ardeosaurus*, the matrix contains more forms than those reported in the original study (**Conrad, 2018**), so we deleted the taxon named ArdeosauruscfBRE. Finally, we deleted the taxon labeled ChineseScincoid because it is not clear which specimen is intended. A second problem concerns the list of ordered characters. Apparently, this list remained unchanged in the published matrix after the deletion of characters 236, 242, and 364. Accordingly, ordering of characters from 236 onwards needs to be changed, and all these modifications have been applied to our published version of the matrix, so this file (Nexus file in **Source code 1**) is ready to be used. Finally, we suggest that some aspects of the matrix may benefit from a detailed revision (e.g., the scorings of some characters, character ordering, addition of taxa belonging to poorly sampled taxonomical groups) if phylogenetic results are the aim of future works.

Several different time bin schemes are used throughout the analyses. Two of them use stratigraphic bins corresponding to the Triassic, Early Jurassic, Middle–Late Jurassic, Early Cretaceous, Late Cretaceous, and Paleogene-present (time scheme 1) or to the Triassic, Jurassic, Early Cretaceous, Late Cretaceous, Paleogene–Pliocene and Pleistocene-present (time scheme 2). Other time schemes correspond to bins with a specified constant length (e.g., 10 Myr or 24 Myr).

Analyses of disparity and morphospace occupation (see **Source code 1** for the code used and source files) are based on the same matrix as the phylogeny. However, we removed *Bharatagama rebbanensis* and *Adriosaurus microbrachis* from those analyses because their inclusion resulted in incalculable distances. We used the MORD metric because of reported problems (**Conrad, 2008**) with the Geometric Euclidean Distance (GED) in matrices with a high percentage of missing information. We used the uncorrected option for the cmdscale function in Claddis, but we also checked that results were comparable using the corrected version (results not shown). We report the stacked temporal plots for PCO1–PCO6 and both stratigraphical schemes but note that the rest of the axes can be easily plotted upon minimal editing of the code. A single randomly selected topology (among the MPTs resulting from the constrained phylogenetic analysis) dated using all three methods ('Hedman,' 'equal,' and 'MBL') was selected for plotting phylomorphospaces using the function plotGMPhyloMorphoSpace of the package Geomorph (**Adams et al., 2019**). We used colors for different groups in separate plots: (1) major lepidosaur groups (stem lepidosaurs, rhynchocephalians, and squamates); (2) more restricted groups (e.g., iguanians, anguimorphs, etc.); and (3) toxicoferan vs. non-toxicoferan squamates. In order to be able to plot the chronophylomorphospace, we re-calculated principal coordinates using the function MorphMatrix2PCoA in Claddis applying the following options: distance.method = "MORD", transform.proportional.distances = "arcsine_sqrt", correction = "none", estimate.allchars = FALSE, and estimate.tips = FALSE. Then we edited the Claddis package function ChronoPhylomMorphospace so that plot parameters can be easily changed. The chrono.subsets function of DispRity (**Guillerme and Poisot, 2018**) was used to split morphospace along two different time bin schemes. The first scheme contains equally long (24 Myr)

time bins, whereas the second one corresponds to the stratigraphic time scheme 1 described above. The function boot.matrix was used to bootstrap the data 100 times, and rarefy to n = 6. The SoV was calculated for the two different time bin schemes with the function DispRity. For both time schemes, we plotted the disparity of all taxa and of squamates alone. WMPD was calculated in Claddis (*Lloyd, 2016*) and bootstrapped 100 times. Analyses were performed for all taxa and for squamates alone. The clusters used in *Figure 3D* are based on the result of applying the pamk R function of the package fpc, as plotted in *Figure 3—figure supplement 6* (data for the first two axes used).

In order to assess changes in sampling across different time bins, we plotted both the sample number (number of taxa in each time bin) and completeness (percentage of characters recorded for each taxon, grouped by time bin). This was applied to the two stratigraphic time schemes (time scheme 1 and 2). Note that for those time bins with one or no taxa sampled completeness and disparity measures are not calculated.

Evolutionary rate analyses were run for three subsets of the morphological matrix: (1) all taxa, (2) all squamates, and (3) all rhynchocephalians and stem lepidosaurs (see *Source code 2* for the code used and source files). We ran analyses twice, one for five randomly selected MPTs of the constrained analysis, and the other for five randomly selected MPTs with the unconstrained analysis as the source. We dated each of these five MPTs five times with three different methods: 'equal' method (*Brusatte et al., 2008*), 'minimum branch length' approach (*Laurin, 2004*) using the R functions from *Lloyd, 2016*, and a whole-tree extension of the Bayesian Hedman algorithm (*Hedman, 2010*). The Hedman node-dating approach uses Bayesian statistics, incorporating probability distribution constraints based on successive outgroup taxa ages (*Hedman, 2010*). Successive outgroup taxa that are both more 'basal' and predate the first lepidosaurs were required to date the nodes close to, and including, the root. Occurrence dates of the following outgroup taxa were used: *Weigeltisaurus jaekeli, Eunotosaurus africanus, Lanthanolania ivakhnenkoi, Orovenator mayorum, Petrolacosaurus kansensis, Anthracodromeus longipes, Hylonomus lyelli, Palaeomolgophis scoticus, Casineria kiddi, Ossirarus kierani, Tulerpeton curtum, Ymeria denticulata*, and *Ichthyostega stensioi* (ages are provided in the supplementary code). For illustrating the results of the per-bin evolutionary rates (according to the two time sets, one corresponding to geological stages and one corresponding to equal 10 Myr bins), we used 'spaghetti plots' showing individual lines for each combination of tree and dating (25 individual lines), as well as an average line, and also highlighting iterations and bins with significantly fast (red triangle) and slow (blue rhomb symbol) evolutionary rates.

R scripts are available as *Source code 1* (for morphospace and disparity), *Source code 2* (evolutionary rates), *Source code 3* (morphospace clusters), *Source code 4* for alternative analysis without integument and Cramér values and *Source code 5* for plotting clade-colored consensus trees.

## Appendix 5

### Extended results

### Extended phylogenetic analyses results

Results of phylogenetic analyses are explained here in detail for two main reasons: (1) the constrained analysis is completely new, as such constraints have not been applied to analyses of this matrix before; (2) a detailed explanation of the results of the unconstrained analysis was not provided by *Conrad, 2018*, who did not figure complete consensus trees because the subject of interest for the article was the position and interrelationships of ardeosaurs and related forms; (3) for both analyses the composition of our sampled taxa is different from the original because we removed a few taxa from the matrix; and (4) we changed the ordering of a few characters (see comment above).

### Constrained analysis

Note that the phylogenetic relationships between extant clades were forced according to phylogenomic results, prior to this analysis. In contrast, the interrelationships between taxa forming these clades and, more importantly, the position of all fossil taxa, which were treated as floaters and could thus move to any possible position in the tree (*Figure 1—figure supplements 1–4*), are results of the present analysis. The strict consensus tree (*Figure 1—figure supplement 4*) has poor resolution because of the poorly defined position of a few rogue taxa, so we selected the majority rule tree (*Figure 1—figure supplement 2*) to discuss the results of the phylogeny. Note, however, that five randomly selected MPTs were used for evolutionary rate analyses, being one of them randomly selected for constructing phylomorphospace and chronophylomorphospace.

#### Outgroup and non-lepidosaur lepidosauromorphs

We followed *Conrad, 2018* in considering *Pamelina polonica* as the outgroup. The list of stem lepidosaurs is completed with *Icarosaurus siefkeri* and *Kuehneosaurus latus* (sister taxa, considered kuehneosaurs in most works) and *Marmoretta oxoniensis*, which is sister to crown lepidosaurs.

#### Rhynchocephalians

Rhynchocephalians form a well-supported monophyletic group containing all forms typically regarded as such but, unexpectedly, with *Scandensia ciervensis* as their sister taxon. The latter had been previously regarded as a stem squamate (e.g., *Evans and Barbadillo, 1998*), as a stem 'scleroglossan' (e.g., *Conrad, 2008*; *Bolet and Evans, 2011*) or in a more derived position (*Bolet and Evans, 2011*), but never as closely related to rhynchocephalians. Note, however, that we have noticed that *Conrad, 2018* scored the dentition of *Scandensia* as triangular (character 212, state 1), when it would be best coded as straight and pointed (state 0). This, together with the persistent notochord (character 230, state 0), a character state that is most probably convergent with rhynchocephalians, and a few other characters, is possibly behind this unexpected (and presumably unreliable) result. Accordingly, we regard the previously reported squamate affinities of *Scandensia* as unquestioned. The few MPTs that do not recover *Scandensia* as sister to rhynchocephalians place it in a more typical position among stem squamates. Our results also differ from specific studies on rhynchocephalians (e.g., *Herrera-Flores et al., 2017*) in that *Gephyrosaurus*, *Planocephalosaurus*, and *Diphydontosaurus* usually (most MPTs) form a monophyletic group, instead of forming a paraphyletic assemblage on the stem of 'derived rhynchocephalians.' Among the latter, two main monophyletic groups are recovered: the first contains *Kallimodon* (a sapheosaur), *Priosphenodon* (an eilenodontine), *Pamizinsaurus*, *Palaeopleurosaurus*, *Pleurosaurus* (pleurosaurids), and *Bharatagama*; the second one contains *Sphenodon* and *Cynosphenodon* (sphenodontids) and *Ankylosphenodon*, *Polysphenodon*, and a paraphyletic *Clevosaurus* (clevosaurs).

#### Stem squamates

*Bavarisaurus macrodactylus*, *Huehuecuetzpalli mixtecus* (forming a monophyletic group), and *Hoyalacerta sanzi* are recovered as stem squamates in a high percentage (97%) of the MPTs. *B. macrodactylus* had previously been recovered as a stem scincogekkonomorph in the original analysis of this matrix (*Conrad, 2018*), but all three taxa have been previously suggested to be stem squamates (*Reynoso, 1998*; *Evans and Barbadillo, 1999*; *Evans et al., 2006*).

## Dibamidae

We forced the position of dibamids as the sister to the rest of crown squamates, but it is worth mentioning that none of the fossils included in the analysis is recovered as related to dibamids in the majority rule consensus tree, rendering the branch leading to them as one of the longest in the tree (together with that of the sole extant rhynchocephalian *Sphenodon*). A minor portion of the MPTs, however, recovered the Cretaceous taxa *Sineoamphisbaena* and *Polrussia* on the stem of Dibamidae.

## Gekkota

*Norellius nyctisaurops* and *Gobekko cretacicus* are recovered as stem gekkotans, but *Hoburogekko suchanovi* forms part of the crown in our analysis because the extant *Pygopus lepidopus* and eublepharids are recovered as consecutive sister taxa of the remaining crown gekkotans (including *Hoburogekko*).

## Scincoidea

*Paramacellodus oweni* and the paramacellodid from Utah (but not other paramacellodids in the classic view, see below) are recovered as stem scincoids. *Tepexisaurus tepexii* is recovered as a stem xantusiid, coinciding with other studies (**Gauthier et al., 2012**). This genus was originally considered a stem scincoid (**Reynoso and Callison, 2000**), but note that in the corresponding phylogenetic analysis xantusiids were recovered as lacertoids rather than scincoids. Our results are coincident with most previous studies (e.g., **Gauthier et al., 2012**) in that *Palaeoxantusia* is a crown xantusiid. *Pseudosaurillus becklesi* is nested within globaurids, a Campanian radiation on the stem of Scincidae plus Cordyliformes and containing *Globaura*, *Bainguis*, *Eoxanta*, *Parmeosaurus*, *Hymenosaurus*, *Myrmecodaptria*, and *Slavoia* in our results. *Palaeolacerta* is recovered (in just 68% of the trees) as sister to cordyliformes, but its position is rather unstable as it is recovered in several distant positions.

## Lacertoidea

None of the fossil taxa included in our analysis is recovered as a stem lacertoid. *Dracaenosaurus* is confirmed as a crown lacertid (e.g., **Čerňanský et al., 2017**). Besides the position of *Liushusaurus* as a stem teioid (an unusual position for this taxon), the fossil record of this group in our analysis is limited to fossils of species reaching the present (e.g., *Tupinambis teguixin*).

## Serpentes

An unexpected result of the present analysis is the position of *Jucaraseps*, *Sineoamphisbaena*, and *Cryptolacerta* in the stem of Serpentes in most trees. None of these has ever been claimed to be related to snakes. The phylogenetic relationships of *Jucaraseps* are not clear, but it has been regarded as a 'scincogekkonomorphan' (**Bolet and Evans, 2012**). The position of *Sineoamphisbaena* has been problematic as it was described as related to amphisbaenians and later reinterpreted as related to macrocephalosaurs (e.g., **Kearney, 2003**). *Cryptolacerta* presents a similar case: it was described as a stem amphisbaenian (**Müller et al., 2011**), but it has been regarded as more closely related to lacertids (e.g., **Longrich et al., 2015**). *Portugalophis lignites* and *Parviraptor estesi* are recovered as Jurassic crown snakes (**Caldwell et al., 2015**) in all trees. *Dinilysia patagonica* and *Najash rionegrina*, on one side, and *Haasiophis terrasanctus* and *Pachyrhachis problematicus*, on the other, are two separate lineages in our results.

## Anguimorpha

Different versions of Conrad's matrix (**Conrad, 2008**; **Conrad, 2018**; **Conrad et al., 2011**) are consistent in recovering as anguimorphs a few taxa that had been rarely referred to this group of lizards. In our majority rule tree, these include the clade formed by *Meyasaurus diazromerali*, *Eolacerta robusta*, *Ornatocephalus metzleri*, *Yabeinosaurus tenuis*, and *Becklesius hoffstetteri*. The latter has been usually regarded as a paramacellodid (e.g., **Estes, 1983**). Regarding the crown, Xenosauridae includes the extant *Xenosaurus grandis* as well as the fossil *Exostinus serratus* and *Restes rugosus*. The next clade to diverge is Shinisauridae, including the extant *Shinisaurus crocodilurus* and the fossil taxa *Bahndwivici ammoskius*, *Merkurosaurus ornatus*, and, less expectedly, *Dalinghosaurus longidigitus* and *Parasaniwa wyomingensis*. Anguidae are represented by the extant *Elgaria coerulea* and the fossil glyptosaur *Peltosaurus granulosus*. The rest of the anguimorphs form a monophyletic group consisting of *Dorsetisaurus purbeckensis*, *Eosaniwa koehni*, and a clade formed by mosasaurs and more advanced 'varanoids,' replicating the typical

morphological structure of the tree. This is because although a sister group relationship between iguanians and anguimorphs was forced, the molecular topology of the constituent clades of anguimorphs was not. Mosasaurs include *Eonatator sternbergi* and *Aigialosaurus* with *Paravaranus angustifrons* as sister taxon of both, on one side, and *Adriosaurus* plus *Proplatynotia longirostrata,* on the other. 'Varanoids' consist of *Parviderma inexacta*, helodermatids (the extant *Heloderma* plus *Saniwides mongoliensis*, *Palaeosaniwa canadensis*, *Estesia mongoliensis*, *Gobiderma pulchrum*, *Paraderma bogerti,* and *Primaderma nessovi*), and varanids that, in our analysis, include the extant *Lanthanotus borneensis*, *Varanus varius,* and *Psammosaurus griseus*, plus the fossil *Telmasaurus grangeri* and a paraphyletic *Saniwa*.

### Iguania

One of the most interesting results of the constrained analysis is the position of both ardeosaurs (sensu lato) and borioteiioids, on the stem of Iguania. This is most likely the result of forcing iguanians into a more crownward position (sister to anguimorphs), which somehow seems to drag both groups that were stem scleroglossans in the original analysis of *Conrad, 2018* and our unconstrained analysis here with them. If this is correct, then ardeosaurs would be filling a gap between the earliest known anguimorphs and the earliest known iguanians. Ardeosaurs in our analysis consist of *Ardeosaurus brevipes*, *Schoenesmahl dyspepsia,* and *Chometokadmon fitzingeri*. A monophyletic *Eichstaettisaurus* is the next to diverge and is thus very closely related to the group above, but rendering Ardeosauridae paraphyletic if *Eichstaettisaurus* is included. Note that *Conrad, 2018* considered Ardeosauridae, Eichstaettisauridae, and Bavarisauridae (as well as borioteiioids) as separate monophyletic clades on the stem of 'Scleroglossa.

Borioteiioids form in our results a monophyletic group consisting of *Polyglyphanodon sternbergi*, *Erdenetesaurus robinsonae*, *Tianyusaurus zhengi*, *Darchansaurus estesi*, *Gilmoreteius ferrugenous*, *Gilmoreteius chulsanensis*, *Adamisaurus magnidentatus*, *Cherminsaurus kozlowskii*, *Gobinatus arenosus,* and *Tchingisaurus multivagus*. Among crown iguanians, the two main groups (the paraphyletic pleurodont iguanians and monophyletic Acrodonta) have radiations in the Campanian. On the one hand, Gobiguania includes the following pleurodont iguanians: *Zapsosaurus sceliphros*, *Anchaurosaurus gilmorei*, *Ctenomastax parva*, *Temujinia ellisoni,* and *Saichangurvel davidsoni*; on the other hand, the stem of Acrodonta is formed by priscagamids (*Priscagama gobiensis*, *Phrynosomimus asper*, *Mimeosaurus crassus*, *Gladidenagama semiplena,* and *Flaviagama dzerzhinskii*) and *Arretosaurus ornatus*.

## Unconstrained analysis

Trees resulting from the unconstrained phylogenetic analysis (*Figure 1—figure supplements 5–7*) roughly match the results of the original analysis of this matrix (*Conrad, 2018*), mainly if we take into account that we removed a few taxa, and that a detailed comparison is not possible because this author did not publish a complete tree, but a strict consensus tree with most major clades collapsed instead. *K. latus* and *I. siefkeri* (kuehneosaurs) are sister taxa. However, in contrast to the results of the constrained analysis described above, *M. oxoniensis* is sister to rhynchocephalians instead of being sister to crown lepidosaurs. The phylogenetic relationships among rhynchocephalians are less well resolved and, besides this, the group formed by *S. punctatus* and *Cynosphenodon huizachalensis* is nested within non-clevosaur advanced rhynchocephalians, instead of grouping with clevosaurs.

### Stem Squamata

*Hoyalacerta sanzi* and *Huehuecuetzpalli mixtecus*, but not *Bavarisaurus macrodactylus* (see constrained analysis above), are recovered on the stem of Squamata.

### Iguania

The unconstrained version of the analysis recovers the typical position in morphological analyses of iguanians as sister to the rest of the squamates (e.g., *Gauthier et al., 2012*; contra *Simões et al., 2018*). Neither ardeosaurs nor borioteiioids are recovered on the stem of iguanians. Gobiguanians, with the same composition as in the constrained analysis, are here recovered on the stem of Iguania, instead of nested within crown iguanians. Crown iguanians are recovered again as containing a paraphyletic group of pleurodont iguanians and a monophyletic Acrodonta, the latter with *Arretosaurus* in its stem, and priscagamids as sister clade.

### Stem 'Scleroglossa'

The stem of 'Scleroglossa' is formed by borioteiioids and ardeosaurs (sensu lato), coinciding with the original results of **Conrad, 2018**. The latter are paraphyletic, formed by a first monophyletic group formed by *S. dyspepsia*, *A. brevipes*, and the unnamed genus corresponding to PMUR58; a second monophyletic group formed by *B. macrodactylus*, *S. ciervensis*, *Eichstaettisaurus gouldi*; and, finally, *Eichstaettisaurus schroederi*.

### Gekkotans

*Eoxanta lacertifrons* and *N. nyctisaurops* are recovered as stem gekkotans. In contrast to the constrained analysis, *Gobekko* joins *Hoburogekko* as a Cretaceous crown gekkotan.

### 'Autarchoglossa'

The group containing non-gekkotan 'scleroglossans' differs slightly from previous results in that teiioids are sister to a group formed by scincoids and anguimorphs. Scincoids include, besides the typical members of the group (scincids, cordyliforms, and xantusiids), globaurids (on the stem of scincidae + cordyliforms), dibamids, amphisbaenians, and snakes. The three latter form a convergent group inside Scincidae, which is why **Conrad, 2008** coined the term Scincophidia.

*Note*: The unexpected and rather unlikely position of *Jucaraseps*, *Sineoamphisbaena,* and *Cryptolacerta* on the stem of Serpentes (**Figure 1—figure supplement 2**) in the constrained phylogenetic analysis seems to be incongruent with morphospace (see **Figure 3—figure supplement 1**) because these taxa cluster with lizards instead of snakes. This seems to hint at some problem with these taxa in particular that for some reason (possibly convergence in some cranial characters related to fossoriality or semi-fossoriality) are attracted to snakes in the phylogeny, but are plotted closer to lizards in morphospace.

## Disparity and morphospace occupation

Plots of morphospaces with all taxa labels for PCO1–PCO6 are reported here (**Figure 3—figure supplements 1–3**), but plots for additional axes can be easily obtained by modifying the code to get the desired PCOs. **Figure 3—figure supplement 4** shows hulls by taxonomic groups. A simplification of these groups into rhynchocephalians, generalized lizards, anguimorphs, and snakes and other limbless squamates in **Figures 2 and 3b** is based on the results of applying the pamk function (see resulting plot in **Figure 3—figure supplement 6**). **Figure 3B** and **Figure 3—figure supplement 4** show that iguanians, and especially chameleons, are the squamate group that is closest to rhynchocephalians. Stem lepidosaurs are situated between rhynchocephalians and squamates, but closer to the latter. Besides iguanians, the cluster containing generalized lizards is formed by scincoids (scincids, cordyliforms, and xantusiids), gekkotans, lacertids, teiioids, and four fossil clades: ardeosaurs (and the possibly related eichstaettisaurs), borioteiioids, globaurids, and paramacellodids. A third cluster contains anguimorphs (including mosasaurs, which are contained in this clade in our phylogenetic results), and a fourth cluster consists of all clades of limbless taxa (snakes, amphisbaenians, and dibamids). Of these, snakes are situated furthest from the centroid.

We also provide the 2D phylomorphospace plot (**Figure 3—figure supplement 5**) that corresponds to the morphospace colored by squamates, rhynchocephalians, and lepidosaurs (**Figure 3A**), and a 3D phylomorphospace plot (**Appendix 3—figure 2**, **Supplementary file 6**) where toxicoferan vs. non-toxicoferan squamates are represented. This plot shows that overlap between both groups is present, but rather limited. In order to complement the temporal morphospace stack of **Figure 2**, we provide stacks for additional PCOs and time scheme 2 (**Figure 2—figure supplements 4–6**). The sudden increase of morphospace in the Middle–Late Jurassic and stability through time until the present day is confirmed in all these additional plots.

Regarding measures of disparity, the SoV (**Figure 4A**) presents two highs in the Late Jurassic and mid Cretaceous, and marked drops in the earliest Cretaceous and around the K-Pg boundary, a point from which disparity remains stable and intermediate between the low disparity of the Triassic and the high disparity of the Late Jurassic and mid Cretaceous peaks. The WMPD plotted by taxonomic group shows that squamates have a much higher disparity than either rhynchocephalians or stem lepidosaurs (**Figure 4—figure supplement 1**). For the WMPD by less inclusive taxonomical groups (**Figure 4—figure supplement 2**), if we arbitrarily set a high disparity for values above 0.4, low disparity for values below 0.35, and intermediate disparity for values between these values, we see that scincids, cordyliforms, snakes, mosasaurs, and anguimorphs show high disparity; stem

lepidosaurs, xantusiids, lacertids, dibamids, borioteiioids, and ardeosaurs (including the possibly related eichstaettisaurs) show low disparity; and for rhynchocephalians, gekkotans, amphisbaenians, and iguanians disparity is intermediate. This same measure plotted through time (*Figure 4—figure supplements 3 and 4*) fails to recover the Middle–Late Jurassic peak on disparity, although disparity for this time bin is higher than those of the Triassic and Early Jurassic, similar to that of the Early Cretaceous, and only slightly lower than that of the Late Cretaceous. Note, however, that this measure has been bootstrapped but not rarefied, and is thus more sensitive to uneven sampling.

## Evolutionary rates

Results for the stratigraphical time scheme (top plot in each figure of *Figure 4—figure supplements 5–22*, with bins corresponding to the Early, Middle, and Late Triassic; Early, Middle, and Late Jurassic; Early and Late Cretaceous; and Paleogene and Neogene) recover similar results as the 10 Myr long time bins (bottom plot of each figure), although, as expected, some resolution is lost. Results for the constrained phylogeny including all taxa and the 'Hedman' method (*Figure 4—figure supplement 5*) recover the highest peaks of evolutionary rates by the Late Jurassic and the Neogene-present time bins, with a moderate peak corresponding to the Late Cretaceous. The same analysis using the equal method (*Figure 4—figure supplement 6*) recovers the same two highest peaks in the Late Jurassic and Neogene-present, and two additional peaks, in the Late Triassic and Late Cretaceous, are only slightly lower. Using the 'MBL' method (*Figure 4—figure supplement 7*) instead produces an extremely high peak in the Neogene-present, possibly as the result of the placement of many nodes towards modern times according to the specific procedure of the method for establishing the age of a node. The other three peaks are present, but only in the Late Triassic and Late Cretaceous of the 10 Myr time bin scheme some trees show significantly high evolutionary rates (the rest being nonsignificant). Results for squamates alone using the 'Hedman' method (*Figure 4—figure supplement 8*) are very similar to those of all taxa, again with the highest peak in the Late Jurassic, a slightly lower peak in the Neogene-present, and an even lower (although still significant) peak in the Late Cretaceous. Roughly the same results are recovered when using the 'equal' method (*Figure 4—figure supplement 9*), but not when using the 'MBL' method (*Figure 4—figure supplement 10*), which again boosts the Neogene-present evolutionary rates to such a high level that the Late Jurassic and Late Cretaceous peaks are no longer significant. Rhynchocephalians (plus stem lepidosaurs) present high evolutionary rates mostly in the Late Triassic in all three dating methods (*Figure 4—figure supplements 11–13*), and in all cases there is a rather constant trend of decreasing evolutionary rates towards the present.

Results for all analyses of the unconstrained topology (*Figure 4—figure supplements 14–22*) are very similar to those for the constrained topology. The presence and prevalence (except for the MBL method) of the Late Jurassic peak in all iterations of the complete dataset and of squamates alone are confirmed. Slight differences include a less clear peak in the Late Cretaceous either because a lower number of trees present significantly high rates or because the peak moves to the Early Cretaceous (e.g., *Figure 4—figure supplement 15*, top and bottom, respectively). Regarding rhynchocephalians (plus stem lepidosaurs), results of the unconstrained analyses (*Figure 4—figure supplements 19–22*) are almost identical to those of the analyses of the constrained phylogeny, although this was expected because most changes in topology affecting divergence times (and thus branch lengths and related evolutionary rates) are concentrated in squamates.

## Sampling, selection of axes, and Cramér values

Because the number of taxa and their relative completeness (number of scored characters against the total number of characters) may have an influence on disparity and evolutionary rates results, we plotted these statistics for the entire dataset in time schemes 1 and 2 (*Appendix 1—figures 1–5*). Time scheme 1 shows that the Middle–Late Jurassic has one of the lowest taxon counts (only slightly surpassing that of the Triassic and Early Jurassic) and the lowest completeness of all time bins (*Appendix 1—figures 1, 3 and 5A*). The Late Cretaceous is the time bin with greatest completeness if the one including the extant taxa (the Paleogene-present) time bin is excluded and presents also a high taxon count. For time scheme 2 (*Appendix 1—figures 2, 4 and 5B*), the Jurassic is retained as one of the two time bins with the lowest completeness. The Paleogene–Pliocene bin (used only in this second time scheme) records a decrease of completeness after the Late Cretaceous bin. In order to discard an important influence of characters with a high degree of missing scores (either because they are rarely preserved in fossils or because they represent unscorable characters in a good portion of the taxa sampled in the matrix), we performed an additional analysis where only those

characters scored for more than 40% of taxa were included. This procedure removed an important number of characters, but resulted in a matrix containing less than 25% of total missing data, a threshold considered safe in terms of completeness. Interestingly, resulting plots of morphospace (not shown) are completely comparable to those reported for the complete matrix, suggesting that characters showing a high degree of incompleteness are barely contributing to the distribution of taxa in morphospace of the first axes.

It is worth noting that although results for all PCOs are not figured, they can easily be obtained applying small edits to the code. In any case, we show in *Appendix 1—figure 6* that the variance explained by additional axes is very low in relation to the first ones.

Finally, we report Cramér coefficients in order to show the correlation between characters and the first two PCOs. Because the chosen methodology (calculation of PCoA) transforms the distribution of character states into a distance matrix, it is not possible to calculate PC loadings as is usually done with continuous data and PCA. Cramér coefficients have been used as an alternative to PC loadings (e.g., *Kotrc and Knoll, 2015*; *Nordén et al., 2018*), although most papers using PCoA skip this step entirely (e.g., *Simões et al., 2020*; *Martínez et al., 2021*). We generated a first plot (*Appendix 1—figure 7*) where we show all characters first grouped by anatomical regions (e.g., preorbital region, postorbital region, palate, braincase, etc.) and then, inside every region, characters that are significant are grouped on the left (green columns of the second and fourth plots, sometimes barely visible), whereas nonsignificant characters are situated to the right (red columns of the same plots). This is a good way to see all data, clearly showing that characters with a high Cramér value are concentrated in the integument/myology anatomical region. However, because most of these characters cannot be scored for fossils, we decided to recalculate the distance matrix, and then plot the new morphospace without including all those characters that were likely missing in all fossils (e.g., those related to skin or myology), and leaving only those that had some potential to be preserved in fossils (e.g., those related to osteoderms), besides those of osteology. The resulting morphospace (*Appendix 2—figure 1*) is very similar to that of the full dataset. This seems to be indicating that despite presenting the highest Cramér coefficients (meaning that they correlate well with the distribution in the corresponding axis), they are not essential to recover the morphospace we are discussing. This probably has to do with the high number of sampled characters, which dilute the weight of any given character. In order to focus discussion to a smaller number of characters, we plotted Cramér values for this second (reduced) dataset, which does not include these soft-tissue characters, only for those characters that presented a Cramér value >0.4 (*Appendix 2—figure 2*). Note, however, that characters with lower Cramér values are still contributing to the distribution, mainly for those taxa that cannot be scored for characters with higher Cramér values. Characters with a Cramér value >0.4 are predominantly concentrated in the vertebral column anatomical region (nine characters for both PCO1 and PCO2) corresponding to characters related to vertebral morphology. The rest of the regions are much less widely represented in PCO1, ranging from four characters related to the braincase to one related to the dentition or postorbital region, or even without representation like the hindlimbs. For PCO2, characters related to the vertebral column are equaled by characters related to the antorbital region (nine in both cases), being the rest of regions represented by a much smaller number of characters, from 0 (e.g., palate, forelimbs, dentition, or hindlimbs), 1 (postorbital, lower jaw), to 2 (e.g. integument – osteoderms or braincase). Note that, despite the fact that all these characters are contributing to the recovered distribution of taxa in morphospace, only those that present a p-value<0.05 can be regarded as significant (marked with an asterisk in the plot). These characters, which are more reliable for interpretation, include in the case of PCO1 the following list of characters: 260 (coracoid anterior emargination), 304 (dorsal compound osteoderms), 306 (ventral compound osteoderms), 420 (facial notch in the crista prootica), 422 (ventrolateral margin of the paroccipital process of the otoccipital), 736 (Hyoid cornu), and 791 (compound supraorbital scale osteoderms in the orbit). In the case of PCO2, the list is formed by the following characters: 49 (jugal, postorbital branch), 54 (quadratojugal), 204 (prearticular, crest with imbedded angular process), 237 (presacral vertebrae, length of transverse processes), 291 (egg teeth), 304 (presence/absence of dorsal compound osteoderms), 541 (frontal(s), subolfactory process fusion, and obliteration of midline suture), 672 (opisthotic/otoccipital, contribution to the posterior auditory foramen), and 800 (dorsal vertebra, midline inter-zygosphenoidal spur). Characters like, for instance, 49 or 54 are clearly contributing to the separation of rhynchocephalians and squamates along PCO2. Interpreting which characters are contributing to the separation of groups along PCO1

appears much more complicated and is not attempted further here. The conclusion would be that, although the characters related to the integument–myology are the ones with highest Cramér coefficients, they are not essential to produce a stable morphospace. Among the rest, it seems that the vertebral column first, and then regions like the braincase, dentition, and postorbital region would contain character states distributions that fit best with PCO1, whereas PCO2 would be more tightly related to the vertebral column again, and the antorbital region. However, as said above, the number of characters is so high that it is unlikely that a given character or set of characters is responsible to a great degree of the recovered morphospace distribution. Instead, the morphospace represents a culmination on multiple changing character scores, some with complex distributions. This is partially expected given that the dataset is composed of characters from across the whole skeleton, all of which change across the major morphotypes.

# Appendix 6

## Extended discussion

### Phylogeny

Phylogenetic results for all main groups are reported above, but there are a few specific results that are worth discussing here. One is that forcing the molecular constraints results in a crownward movement of borioteiioids and ardeosaurs, which become stem iguanians. This is important in filling a gap between the earliest anguimorphs and the earliest iguanians. This placement also provides added evidence of the presence in the Jurassic of Toxicofera, a clade that is usually considered as highly derived, and suggesting that an important part of the evolutionary history of squamates occurred before the end of that period. The three main toxicoferan clades would be represented by Jurassic forms, namely, *P. estesi* and *P. lignites* as snakes (according to *Caldwell et al., 2015*, and results herein), *Dorsetisaurus* as an anguimorph, and ardeosaurs (sensu lato) as iguanians. Difficulties in the placement of borioteiioids, as reflected in their unstable position among different phylogenies in previous works, are potentially related to a possible convergent nature of similarities between teiioids and toxicoferans. Note, however, that the exact phylogenetic position of problematic taxa (e.g., the identification of *Parviraptor* and *Portugalophis* as snakes) is irrelevant to morphospace and disparity discussions because topology is not considered in the construction of the morphospace, just used to illustrate phylomorphospaces.

## Morphospace

Results of our disparity, morphospace occupation, and evolutionary rate analyses are discussed here in the context of all available evidence, including patterns of diversification as informed from the fossil record and current phylogenies. Phylogenetic results are not discussed in greater detail because they were only meant to provide a phylogenetic framework for evolutionary rate analyses (and a specific topology to be used in plots of phylomorphospace). Morphospace distribution of points and associated measures of disparity do not rely on phylogeny and, as expressed in the Materials and methods section, we have incorporated phylogenetic uncertainty into our analyses by conducting separate analyses for constrained and unconstrained phylogenies, and by using multiple randomly selected most parsimonious trees. Although the constrained phylogeny has been chosen to illustrate the results of our analyses, it is worth noting that our results are robust to changes in topology. A conflicting point regarding the evolutionary history of squamates is that the acceptance of the molecular topology requires that the numerous morphological similarities between iguanians and *Sphenodon* (e.g., *Estes et al., 1988*) are the product of reversals and convergences (*Losos et al., 2012*) and should expectedly change the timing of the events in the evolutionary history of the group. This similarity between iguanians and rhynchocephalians is evident in the 2D and 3D plots of morphospace (e.g., *Figure 3B*, *Figure 3—figure supplement 4*, *Supplementary file 2*), where they are plotted close to each other in morphospace. The 3D plot (*Figure 3C*) shows iguanians (especially chameleons) among all squamates as the closest group to rhynchocephalians. Note that striking similarities between rhynchocephalians and iguanians in soft tissue and osteology, such as an apparently conserved morphology of the tongue, the vomeronasal organ, and closely placed cranial bones (*Conrad, 2008*; *Conrad et al., 2011*), must be convergences if molecular studies are correct in placing iguanians with anguimorphs (*Mongiardino Koch and Gauthier, 2018*). Of note, removing rhynchocephalians and coloring toxicoferan vs. non-toxicoferan squamates (*Appendix 3—figure 2*, *Supplementary file 6*) results in a good separation of both groups, except in the contact region between them, where some overlap occurs. This suggests that some morphological support for Toxicofera is present in the dataset, even if unconstrained phylogenetic analyses fail to recover monophyly for the group.

## The JME event

The results obtained here, implying the existence of a previously unidentified event triggering an early increase in disparity of squamates linked to high evolutionary rates by the Late Jurassic at the latest, have profound implications for interpreting evolutionary dynamics in lepidosaurs. Moreover, it not only changes the focus for understanding the main radiation of lizards (and lepidosaurs) from the Cretaceous to the Jurassic, but also adds to current discussions on the importance of Jurassic events in shaping Mesozoic tetrapod assemblages as a whole. Understanding the processes behind extremely successful clades is key not only to acquiring a more complete picture of past and present biodiversity, but also to help in the prediction of future trends for vulnerable portions of the tree of life. The study of squamates (lizards, amphisbaenians, and snakes), as one of the largest tetrapod clades

that dominate modern landscapes yet one of the least understood, is not trivial in this regard. This limited knowledge on the early evolutionary history of the group is, in part, linked to a fossil record that is poor for a great part of its early history (*Evans, 2003*), as well as the fact that other clades of vertebrates have received greater attention for various reasons. This admittedly poor and uneven fossil record limits our understanding of the timing, mode, and reasons behind the diversification of the clade. As an example, a recent study (*Cleary et al., 2018*) on the diversity of lepidosaurs, based on generic occurrences along the Mesozoic and Paleogene fossil record, showed an apparent low diversity for the group for the greatest part of the Mesozoic, and then a sudden peak in the Late Cretaceous. However, the same study highlights that the available data for the greatest part of the Mesozoic is too poor to provide confident conclusions. This is in line with *Evans, 2008*, who stated that mid-Cretaceous squamates showed increased diversity, although they claimed that it was not possible to determine if this was a real Cretaceous trend or the result of a more complete record in the Cretaceous. At the same time, this impoverished early fossil record could be partly behind the delay in the first observed record of many clades and, according to this, in the calculated divergence ages and tied evolutionary rates of many key lineages.

From the Early Jurassic onwards, mesic Laurasian deposits are characterized by the presence of fish, lissamphibians, crocodiles, turtles, and choristoderes, in what *Evans, 2008* describes as representing lowland, freshwater lagoonal, or wetland deposits. According to these authors, no Triassic/Early Jurassic deposit has yielded an equivalent assemblage, which instead usually contain rhynchocephalians. The fact that rhynchocephalians and squamates are recorded together in some Jurassic and Cretaceous deposits but always in unequal proportions (*Evans, 1995*) led some authors (e.g., *Evans, 2008*) to the conclusion that these differences in composition may be related to differences in ecology, rather than being purely taphonomic. We argue that the previously detected (but possibly artificial) increase in diversity related to the improved fossil record of the Late Cretaceous has overemphasized the importance of the events occurring in this period regarding the radiation of squamates. Although the KTR might be a real event that triggered diversification among different clades, it is not the first time that the fit of a clade radiation in this Late Cretaceous event is questioned. Just as an example, *Lloyd et al., 2008* suggested that a sampling bias might be behind the apparent increase in diversification of dinosaurs through the Late Cretaceous, in a case similar to the one presented herein.

Another example showing that oversampling in a particular time bin might result in an exaggerated signal is the high rates recovered for the last 10 Myr time bin, an issue that sees its most exaggerated version in the evolutionary rates calculated using the MBL method. The latter method tends to place divergence times towards the present, mainly when the number of extant taxa is high, as a result of an effect equivalent to the pull of the recent described for diversity analyses (*Raup, 1979*; *Sepkoski, 2016*). This concentration of nodes in this time bin results in short branches that are necessarily correlated to generally higher evolutionary rates. This is because, if we assume a constant rate of changes, such a high concentration of short branches is expected to yield higher evolutionary rates. The opposite might be true for extremely long branches, which could be behind the low evolutionary rates recovered for extant taxa lacking closely related forms (e.g., the taxon *Sphenodon* in *Herrera-Flores et al., 2017*). The fact that the Late Jurassic peak in evolutionary rates appears in all the versions of the analysis performed, and in most of them being the highest peak, is interpreted here as proof of a true signal for high evolutionary rates because this is not a particularly highly sampled period (in terms of number of taxa, or of their completeness, see *Appendix 1—figures 1, 3 and 5A*).

Other considerations regarding our results concern groups that have a record that is much younger than their expected origin, or that have fragmentary and/or early records not included in the data matrix used (which is biased towards more complete specimens informative for phylogenetic analyses). For example, some Jurassic lizards have been interpreted as iguanians (e.g., *Bharatagama*, but see results in the original analysis of the matrix and herein), but the first clear record of the clade might be as young as the Late Cretaceous. If anguimorphs were present in the Late Jurassic (*Dorsetisaurus*), then iguanians, as their sister taxon, should be equally old. As expressed above, the recovery of ardeosaurs and borioteiioids as stem iguanians is interesting in this regard, as, if the position of the former is confirmed, it would fill a great portion of the gap between the first recorded anguimorphs and the first recorded iguanians. Other groups, like dibamids, have a much younger fossil record (if any) than would be expected from their sister group position to the remaining squamates. Other examples exist, but the point here is that a future discovery of stem members of clades with a poor fossil record is expected to result in even higher evolutionary rates

(and possibly increased disparity) especially in early time bins. Current patterns of diversification and the apparently poorly sampled Middle and Late Jurassic, point to these intervals as the ones with a greatest potential for providing new evidence of an early radiation of squamates as new forms are discovered.

Another point that needs to be considered is that our analysis is biased towards more complete specimens. This is the result of the initial selection of taxa (*Conrad, 2018*), based on those scored for a maximum amount of codifiable characters in order to be useful in phylogenetic analysis. According to this selection procedure, the first sampled member of a group is not always the earliest taxon (or specimen) of that group to appear in the fossil record highlighting, again, that the morphospace occupation (and disparity) in Jurassic bins is probably underestimated. The possible influence of the 'Lagerstätten effect' in the current results is not investigated here, but deserves attention and will be assessed in future studies.

A second time bin scheme (scheme 2) was used to explore the possible effect of the K-Pg boundary. The results show a very slightly decreased morphospace occupation for the Paleogene–Pliocene time bin. However, because sampling and completeness for this time bin is smaller than for the previous time bin (*Appendix 1—figures 2, 4 and 5B*), we interpret this decrease as possibly an artifact of an impoverished fossil record and/or sample selection. This is related, in our particular case, to the limited number of well-preserved fossils of this age (Messel and the Green River formation being among the few exceptions), rather than a comprehensively poor fossil record. Even if we accepted the results as related to the true signal, occupied morphospace after the K-Pg boundary is not severely reduced as would have been expected if the K-Pg extinction had strongly affected squamates, as previously suggested (e.g., *Longrich et al., 2012*), but exploring the effect of this event would require shorter bins in order to track the post K-Pg event recovery, and much more complete Paleogene sampling. Even in terms of diversity alone, statements that lizards underwent a mass extinction at the K-Pg boundary are hard to sustain, considering that the number of clades that completely disappeared during this extinction is extremely low. Moreover, the number of extant clades that were already present in the Late Cretaceous (and thus, survived the extinction) is notable. Regarding the measures of disparity, the SoV and WMPD through time both show a decrease in disparity for the Paleogene–Pliocene time bin, although again we argue that this lower relative disparity might be related to a low sample size for this time bin, which hampers the recording of representatives of forms that would be necessary to get the true extent of morphospace and associated disparity values.

Evolutionary rates seem to provide stronger evidence for a decrease in evolutionary rates through the entire Paleogene. There is little diversification through this time period, which results in a higher proportion of long branches that, at the same time, can result in lower evolutionary rates. It is not clear, thus, if the Paleogene represented a period of poor diversification and low evolutionary rates or if, on the contrary, we are looking at an additional case of decrease in this measure related to poor sampling. Even if the Paleogene was interpreted as a time of slightly reduced morphospace and lower evolutionary rates, all seem to recover in the last time bin. This shows long-term stability for the morphospace occupied by lepidosaurs in general (and that of squamates in particular). Several specific features of the distribution of the different points in post K-Pg morphospace occupation in our results explain the apparent stability recorded: (1) the presence of the extant *Sphenodon* is important in maintaining the vertex corresponding to rhynchocephalians. It is not difficult to imagine an alternative scenario where this taxon (and thus all rhynchocephalians) had gone extinct, causing the removal of an important pole of lepidosaur morphospace. It has to be noted, however, that the large empty space between rhynchocephalians and squamates is gradually filled by iguanians (specially by chameleons) from the Cretaceous onwards, and thus the lack of rhynchocephalians would not result in such a large loss of occupied morphospace as might have been the case in earlier periods. (2) The clades that appear in post K-Pg assemblages do not expand the occupied morphospace. Instead, they fill empty spaces inside the envelope of already occupied morphospace or they directly overlap with clades already present.

## Extended conclusions

According to our results, the age for crown Squamata would be Middle Jurassic ('MBL' method), or even earlier, in the Late Triassic ('equal' method), or the Middle Triassic ('Hedman' method). The age provided by other studies (e.g., *Jones et al., 2013*) is around the Triassic–Jurassic boundary, thus

suggesting an intermediate age. In summary, the results provided by our disparity and evolutionary rate analyses strongly suggest that squamates underwent a great adaptive radiation by the Late Jurassic at the latest. Disparity analyses show a great change in occupied lepidosaur morphospace by the Middle–Late Jurassic, linked to the appearance of squamates in the fossil record. The Late Jurassic peak in lepidosaur evolutionary rates is present in all analyses, showing that these results are robust to changes in composition of the dataset and topology of the phylogeny. Importantly, recognized uncertainties regarding the phylogeny of squamates were considered by randomly sampling among different MPTs (uncertainties at a small scale, concerning a small number of labile taxa), but also confronting the pure morphological topology against the topology from constraining extant taxa to the molecular position. The Late Jurassic peak is recovered in all cases, suggesting that these results are robust to these changes in topology, and thus more reliable than if only supported by a few of the trees. Both the temporal range of each fossil and alternative methods in dating nodes have been considered. Because disparity results do not depend on a phylogeny (except when plotting a phylomorphospace), our results reveal a strong signal for this Middle–Late Jurassic squamate adaptive radiation, independent of which topology or dating for the trees is preferred. There has been a recent shift towards more integrative methods such as Bayesian clock estimates of divergence times and morphological evolutionary rates that are not used in this work because it was devised before these were widely used. It is worth noting that (1) our divergence times cover the ranges of divergence times recovered by other works that have applied Bayesian methods (e.g., *Simões et al., 2020*) and (2) our own preliminary results after applying the same Bayesian methods to the present matrix (work in progress) are largely congruent with the conclusions presented here.

It has been said (*Gauthier et al., 2012*) that rhynchocephalians dominated the fossil record of Lepidosauria during the Triassic and Jurassic, 'with the squamate branch becoming abundant, in a classic pattern evolutionary relay, only much later during the Cretaceous.' However, these authors refer to abundance, presumably in the fossil record, which in the case of squamates coincides with an increase of diversity at the genus level (see how *Appendix 1—figure 5* reflects this increase in sampled taxa in the morphological matrix used for this study). This Cretaceous diversification, coinciding with the KTR, can be interpreted as a secondary radiation of the crown into modern clades, and is thus not strictly related to the primary radiation we describe in the Middle–Late Jurassic. Another point that favors our interpretations is that dated phylogenies, and no matter what method is used to date them, suggest that many of the main squamate clades had already diverged by the Late Jurassic (*Figure 1—figure supplement 1*). Because we know that the fossil record is fragmentary and uneven, it is even possible that the addition of new taxa/fossils could move this radiation backwards, but it is unlikely that an improved fossil record would result in a movement of this radiation towards the Cretaceous. The discoveries of the last decades from the Jurassic of China (e.g., *Sullivan et al., 2014*) regarding dinosaurs (especially feathered avian dinosaurs), but also mammals and lizards show that our knowledge of the timing of events can radically change with the reporting of new deposits containing key fossils, and have a tendency to move backwards in time. The Early Cretaceous of China has shown that squamates from this age already present extreme adaptations (an example would be *Xianglong*, the earliest known gliding squamate described by *Li et al., 2007*), supporting the view that important events in the radiation of squamates occurred before the Late Cretaceous. Accordingly, we suggest that more attention should be paid to events occurring in the Jurassic that could have triggered the diversification and disparity expansion of squamates.

The Jurassic sees a first major diversification in terms of taxa, but also of morphotypes. As explained above, all four major morphotypes of lepidosaurs (rhynchocephalians, generalized lizards, anguimorphs, and the limbless forms as represented by snakes) were already present by the Middle Jurassic. Finally, it seems that the KTR might have had a moderate influence on the evolutionary history of squamates, likely overinterpreted by an enhanced fossil record from the middle–Late Cretaceous. Most of the extant clades of squamates were already present in the Mesozoic (many of them as early as the Middle–Late Jurassic), and differences between Mesozoic and post-Mesozoic assemblages are the result of the mixture of the extinction of a low number of clades at the K-Pg boundary, and the rise of a few new clades (e.g., lacertids, amphisbaenians) along the Paleogene. Considering that the groups of lepidosaurs containing the taxa with a largest body size (mosasaurs, borioteiioids, and most rhynchocephalians) seem to have been most affected by the K-Pg extinction, small size might be one of the reasons for a greater survival of squamates, although providing statistical evidence for this is beyond the scope of our work.

