## [Editor Report]

This article presents an evaluation of the macroevolutionary history of squamates (lizards, snakes, and relatives) and is relevant to evolutionary biologists and paleontologists interested in this group. The ‘early burst’ of disparity in squamates demonstrates that squamates established their morphospace range much earlier than had been assumed, and the long-term stable morphospace occupation ever since.

---

## [Decision Letter]

**Decision letter after peer review:**

Thank you for submitting your article "The Jurassic rise of squamates as revealed by lepidosaur disparity and evolutionary rates" for consideration by *eLife*. Your article has been reviewed by 3 peer reviewers, including Min Zhu as Reviewing Editor and Reviewer #1, and the evaluation has been overseen by George Perry as the Senior Editor.

Essential revisions:

The present work should be of interest to evolutionary biologists concerned with macroevolution as well as taxonomic specialists. The main conclusions are supported by the data with caution, and they complement and refine some other recent results. The authors go to considerable lengths to rule out alternative explanations. Overall, while the conclusion of this manuscript is novel, many concerns addressed by the reviewers (below) should be considered for revisions and improvement. The manuscript could be published without the need for new analyses, although we would like to see more, especially how rates are affected by enforcing divergence times more in line with molecular trees. If new analyses are not performed, then the manuscript should be toned down to account for the many sources of uncertainty that permeate the present analyses, and acknowledge the ways in which the results depend on methodological decisions that could be improved upon.

(1) The discussion of ecology (lines 95 and following) is oversimplified (as the authors recognise), perhaps inevitably so.

The authors recognise that the "ecology" of the clusters discovered by the authors are difficult to generalise (what is the ecology of limbless forms that unites scolecophidians and boas?). Given that, we would consider striking "ecology" from the title and focus just on disparity.

(2) While we believe that the authors have done sufficient due diligence to rule out potential confounding factors, it would still be worthwhile to be more explicit about one limitation, namely the issue of missing data and inapplicable characters. For instance, this paper could be brought up:

Gerber, S. (2019). Use and misuse of discrete character data for morphospace and disparity analyses. Palaeontology, 62(2), 305-319.

We are not convinced that a broad-scale expunging of "inapplicable" characters (say) is the best way to solve the issue (one thinks of limb character in limbless forms), but a more explicit acknowledgment of this limitation would make the paper more compelling.

(3) One weakness is the issue of completeness, which has been raised prominently in recent years. The authors approach this by way of examining Cramér coefficients and throwing out characters that are generally missing for fossils (like those of integument). While this is not unreasonable, and results in disparity plots that are broadly comparable to analyses in which these characters are present, there remain other characters (especially those related to the limbs), this problem was less thoroughly addressed than others.

(4) One of the striking aspects of the results is that the interpreted expansion of morphospace in both the Jurassic and Cretaceous occurs *before* the interpreted spike in evolutionary rates (not exactly "roughly coincident" as stated). This seemingly counterintuitive result is not addressed.

(5) As someone who has performed principal coordinate analyses on lepidosaur morphological datasets, one reviewer can confirm that the first few dimensions invariably separate limb-bearing from limbless forms, as well as rhynchocephalians from squamates (in the case of this analysis, anguimorphs seem to also contribute to this axis). This is exactly what is found in Figure 1b, but also previously in Watanabe et al., (2019) and Simoes et al., (2020). As such, separating these four groups (rhynchocephalians, snakes, anguimorphs and "generalized lizards", as defined in lines 95-104 by the authors) explains a large proportion of total variance, which loads prominently on the first axes but is likely to contribute to some extent to others. Given that the matrix being used was "designed to identify the relationships among the major squamate clades" (Conrad 2018), it is no surprise that the same pattern emerges in this study. The "Jurassic morphospace expansion" (JMA) proposed here is nothing more than a period of time when a morphospace occupied by only one of these groups (rhynchocepalinas) gives way to one occupied by all four. This result is not novel (i.e., does not depend on any discovery made in this study), it is just presented here in a novel and quantitative way. In fact it is already stated in some of the first lines of the introduction that these morpho-groups are known to already be present in the Middle and Late Jurassic (lines 47-49), and that a similar pattern had already been mentioned in the literature (line 238). Furthermore, whether this does represent a true event of expansion or an artifact of the fossil record is also unclear given the poor fossil record of the clade during this period, as noted by the authors. Furthermore, it ultimately depends on many methodological decisions not entirely validated (such as temporal binning, choice of distance metric, etc), as well as character and taxon sampling decisions that were not optimized for the purpose of macroevolutionary inference, and might therefore supported biased patterns, not just during the Jurassic but also later on.

(6) The discovery of this peak in disparity if further linked to high rates of morphological evolution. Note however that disparity and rates are not necessarily linked, and high rates are not necessary to support a JME. However, this result is entirely dependent on the topology and divergence times supported by this study. The methods of inference and calibration employed here are much more simplistic than those of previous studies, and allow for a much poorer integration of uncertainty into subsequent analyses. Results differ crucially from those of all previous studies, including a much faster early diversification of squamates, and much older ages for some crown clades such as snakes, anguimorphs and iguanians. This leads to a concentration of branches with high levels of morphological change in the mid to late Jurassic, ultimately responsible for the peak in rates found. The methods employed and the way in which they are presented once again do not rule out the possibility that this is an artifact of the way the study was set up, and a major reason why no such increase in rates was discovered by previous studies.

(7) There is a lack of consistency throughout the manuscript. For example, the JME label applied to each of the four panels of Figure 1 seems to represent four different periods of time. The topology in Figure 1A was calibrated using a different method that the rates shown in Figure 1D, making it difficult to draw conclusions. While a staggering 51 supplementary figures are presented, many of these are unedited and thus highly cryptic, others such as the many ordination plots are unlabeled and thus do not contribute much.

Some aspects of this manuscript are very interesting, such as the position of extinct lineages under topologies constrained to show different relationships among major extant clades. However, its main claim is somewhere between trivial (a change from 1 to 4 morpho-groups somewhere in the second half of the Jurassic) and potentially unsubstantiated (a coincident peak in rates).

The current narrative, emphasizing a completely new and unexpected view on squamate diversification might be true, but it rests on many assumptions. A more realistic interpretation should focus on the existence of four morpho-groups within squamates (which could even be tested with this data, see below), and their timing of origin. The manuscript starts by posing the question "why are some groups highly species-rich and others are not?" (line 16). This question is never answered, and in fact the manuscript does not seek to do so or use data that could. But that kind of exaggeration permeates the rest of the manuscript, and the reviewer thinks it will be benefit from a more realistic approach to the data at hand.

(8) One reviewer thinks the paper should validate some of their methodological choices better, or even better improve upon them. For example, two disparity metrics are employed (one of which does not even show a JME), but no reason is given why these were chosen. Results from other post-ordination metrics should be presented as well, along with some validation of why certain metrics where favored. This is implemented by Guillerme et al., 2020 Eco Evo and employs some of the same programs already used. Furthermore, a large part of the findings relies on time-calibrated analyses that are poorly explained. For example, although 3 methods of post hoc scaling of topologies are explored, only 1 topology is presented as a supplementary figure. Details on the methods themselves are missing, as it is unclear what time information went into the calibration and how it was gathered and treated. Ultimately however, branch lengths scaling seems to be resulting in unlikely ages for many clades. The reviwer believes results need to be repeated using a topology calibrated using molecular data, or at the very least including secondary node constraints based on other published molecular studies.

One aspect that could also help improve this paper is an exploration of morpho-functional groups within morphospace. Currently, traditional views are simply repeated (see above regarding the four groups of lepidosaurs) and forced onto the results rather than extracted from them. What the data at hand could help establish (possibly along other datasets that are said to have been dropped due to poor sampling of Jurassic fossils) is how many morphological clusters of taxa can be found (using k-means or expectation maximization algorithms for example), and when in time do those originate based on the topologies at hand (not just the one used here but those previously published as well).

(9) The earliest fossil records of the crown squamates are the Middle Jurassic. All these records (for some constituent major clades) in the Middle Jurassic should be clearly stated and/or cited in the text. As one reviewer thinks, this is why the authors separated Early Jurassic from Middle-Late Jurassic in their analyses. The criteria for this separation should be discussed. Otherwise, the readers would doubt why you did not separate Early-Middle Jurassic from Late Jurassic. Figure 1a, which shows the earliest record of the crown squamates later than the Middle Jurassic, is misleading in this regard.

(10) The manuscript should be edited carefully following the author guidelines of *eLife*. Some of the discussions in the supplementary information text can be moved to the main text to gain a better readership. The Discussion and especially Extended Discussion would benefit from some further subdivision. Some of the supplementary figures (Extended Data Figures 1-10), which are not well demonstrated, can be deleted or replaced with a batch file. Overall, the supplementary figures should be re-organized to have a uniform format. The authors should put more effort into the supplementary figures, many of which are unintelligible.

Other claims that need to be revisited:

(1) Line 132 – "These evolutionary rates are not dependent on phylogeny": They are very dependent on phylogeny. This analysis shows they are not dependent on topology (molecular vs. morphological) but both of those are likely to share very similar branch lengths for all branches in common, making the result still strongly dependent on phylogeny.

(2) Lines 172-173 – "The second event, which fits well with the timing of the KTR, would be coincident with the radiation of the constituent crown groups of Squamata": Figure S1, which is the only time-calibrated topology shown in its entirety, shows crown snakes, anguimorphs, iguanians, lacertoids and skinks originating in the Jurassic, and radiating well before the KTR.

3) Lines 189-191 – "The only observable changes from then on correspond to a slight expansion of the edges of the occupied morphospace, and a notable increase in the density of points filling this morphospace": This is entirely contingent on analyses using a matrix that was tuned to capture the deep phylogeny of the clade, and is something that should be acknowledged.

4) Line 271 – "basal position of Iguania": This and other instances of the use of the word basal should be corrected, living groups cannot be basal to other living groups.

(5) The reviewer has not been able to find which dimensions were considered to estimate the disparity curve shown in Figure 1C. Please add this information to the caption.

6) The reviewer would consider striking some comments from the body of the paper that seem more like personal annotations ("We expected that….").

7) L.24: "the Middle and Late Jurassic (174-145 Ma)" is clearer than "the second half of the Jurassic (170-145 Ma)" to the readers, and also consistent with that in Line 48.

(8) L.82: Figure 1b-d as a separate figure?

9) L.90: For "pan-Squamata", do you mean the total group Squamata?

10) L.108, 171, 212: "mid-late Jurassic" Ambiguous here、Middle and Late Jurassic?

(11) L.145：'stem lepidosaurs' for 'ancestral lepidosaurs'

(12) L. 151: Extended Data Figure 11 as a main-text figure?

(13) L.172, 175, 'late', upper case.

(14) Extended Data Figures 23, 24, 43-47, 'early', 'middle', 'late', upper cases.

*Reviewer #1:*

This study reveals the explosive adaptive radiation of squamates in the Middle and Late Jurassic, situating the dates of origins of their constituent major clades. The "early burst" of disparity in squamates demonstrates that squamates established their morphospace range much earlier than had been assumed, and the long-term stable morphospace occupation ever since. Overall, the results are innovative and significant to evolutionary biology of lepidosaurs.

*Reviewer #2:*

The evolution of disparity, as distinct from species diversity, is a central question in macroevolution. The authors apply recently developed methods keyed to widely available, increasingly large-scale data sets (phylogenetic character matrices) to examine when major changes in body form occurred in the evolution of lizards and snakes, one of the most species-rich living vertebrate groups. They conclude that major changes in body form -- and an expansion of the parameter space occupied by lizards and snakes -- occurred tens of millions of years earlier than anticipated by most previous studies.

Strengths:

One of the major strengths of the disparity method (as is well recognised) is the fact that it is based on widely available data sets (phylogenetic character matrices), rather than the bespoke morphometric data sets traditionally employed in the study of disparity. This makes it much more widely applicable.

One of the major strengths of the paper is the lengths to which the authors go to rule out alternative explanations. For instance, the analyses of evolutionary rates were conducted under different assumptions of tree topology and divergence times to rule out a dependency on that factor (a potentially serious problem in squamates).

Weaknesses:

The discussion of ecology (lines 95 and following) is oversimplified (as the authors recognise), perhaps inevitably so.

One weakness is the issue of completeness, which has been raised prominently in recent years. The authors approach this by way of examining Cramér coefficients and throwing out characters that are generally missing for fossils (like those of integument). While this is not unreasonable, and results in disparity plots that are broadly comparable to analyses in which these characters are present, there remain other characters (especially those related to the limbs), this problem was less thoroughly addressed than others.

One of the striking aspects of the results is that the interpreted expansion of morphospace in both the Jurassic and Cretaceous occurs *before* the interpreted spike in evolutionary rates (not exactly "roughly coincident" as stated). This seemingly counterintuitive result is not addressed.*Reviewer #3:*

Bolet et al., explore the evolutionary history of lepidosaur morphology. Using a previously published dataset sampling a wide range of living and extinct lineages, they analyze both morphospace occupation and rates of evolution through time. By combining a series of phylogenetic and macroevolutionary methods, they reveal an event of morphospace expansion (coupled with increased evolutionary rates) during the second half of the Jurassic. This contrasts with previous studies, which had recovered a predominant signal of diversification and ecological innovation during the mid-Cretaceous, coinciding with the radiation of other major animal and plant lineages and a large-scale restructuring of terrestrial ecosystems. While their conclusion is novel, the manuscript is not based on any new morphological or paleontological data, and employs relatively simple methodologies that can potentially determine the outcome. My main concerns are the following:

(1) As someone who has performed principal coordinate analyses on lepidosaur morphological datasets, I can confirm that the first few dimensions invariably separate limb-bearing from limbless forms, as well as rhynchocephalians from squamates (in the case of this analysis, anguimorphs seem to also contribute to this axis). This is exactly what is found in Figure 1b, but also previously in Watanabe et al., (2019) and Simoes et al., (2020). As such, separating these four groups (rhynchocephalians, snakes, anguimorphs and "generalized lizards", as defined in lines 95-104 by the authors) explains a large proportion of total variance, which loads prominently on the first axes but is likely to contribute to some extent to others. Given that the matrix being used was "designed to identify the relationships among the major squamate clades" (Conrad 2018), it is no surprise that the same pattern emerges in this study. The "Jurassic morphospace expansion" (JMA) proposed here is nothing more than a period of time when a morphospace occupied by only one of these groups (rhynchocepalinas) gives way to one occupied by all four. This result is not novel (i.e., does not depend on any discovery made in this study), it is just presented here in a novel and quantitative way. In fact it is already stated in some of the first lines of the introduction that these morpho-groups are known to already be present in the Middle and Late Jurassic (lines 47-49), and that a similar pattern had already been mentioned in the literature (line 238). Furthermore, whether this does represent a true event of expansion or an artifact of the fossil record is also unclear given the poor fossil record of the clade during this period, as noted by the authors. Furthermore, it ultimately depends on many methodological decisions not entirely validated (such as temporal binning, choice of distance metric, etc), as well as character and taxon sampling decisions that were not optimized for the purpose of macroevolutionary inference, and might therefore supported biased patterns, not just during the Jurassic but also later on.

(2) The discovery of this peak in disparity if further linked to high rates of morphological evolution. Note however that disparity and rates are not necessarily linked, and high rates are not necessary to support a JME. However, this result is entirely dependent on the topology and divergence times supported by this study. The methods of inference and calibration employed here are much more simplistic than those of previous studies, and allow for a much poorer integration of uncertainty into subsequent analyses. Results differ crucially from those of all previous studies, including a much faster early diversification of squamates, and much older ages for some crown clades such as snakes, anguimorphs and iguanians. This leads to a concentration of branches with high levels of morphological change in the mid to late Jurassic, ultimately responsible for the peak in rates found. The methods employed and the way in which they are presented once again do not rule out the possibility that this is an artifact of the way the study was set up, and a major reason why no such increase in rates was discovered by previous studies.

(3) There is a lack of consistency throughout the manuscript. For example, the JME label applied to each of the four panels of Figure 1 seems to represent four different periods of time. The topology in Figure 1A was calibrated using a different method that the rates shown in Figure 1D, making it difficult to draw conclusions. While a staggering 51 supplementary figures are presented, many of these are unedited and thus highly cryptic, others such as the many ordination plots are unlabeled and thus do not contribute much.

Some aspects of this manuscript are very interesting, such as the position of extinct lineages under topologies constrained to show different relationships among major extant clades. However, its main claim is somewhere between trivial (a change from 1 to 4 morpho-groups somewhere in the second half of the Jurassic) and potentially unsubstantiated (a coincident peak in rates).

[Editors’ note: further revisions were suggested prior to acceptance, as described below.]

Thank you for resubmitting your work entitled "The Jurassic rise of squamates as supported by lepidosaur disparity and evolutionary rates" for further consideration by *eLife*. Your revised article has been evaluated by George Perry (Senior Editor) and a Reviewing Editor.

The manuscript has been improved but there are some remaining issues that need to be addressed, as outlined below:

See comments from Reviewer #3.

*Reviewer #1:*

The manuscript is well revised to address the comments from the reviewers.

*Reviewer #3:*

I congratulate the authors on the effort put towards revising their manuscript, and after carefully reading their response and new version of the manuscript I have just a handful of minor suggestions that I would like to see implemented, or more strongly stated, in the manuscript before the final version is published.

(1) Editing some of the supplementary files shown would not take that much time at all, and it would make them much more readable than they are now. Reducing the size of the label text in the trees now shown as supplementary figures to figure 1, and color coding them by clade, does not represent too much work. Same holds true for color coding by clade the dots in the many supplementary PCO plots.

(2) The authors state in their response that "Methods in some other recent studies include extensive genomic data as well, but such data is only available for modern taxa and can only change the overall tree topology, but not the dating of deep divergences, as here." Genomic data can most definitely change a lot more than just topology, and in fact do modify the dating of deep divergences. The methods implemented currently for time-scaling make a number of simplified decisions regarding the calculation of branch lengths that come from the fact that they do not incorporate any quantifiable (morphological or molecular) sense of the degree of divergence occurred between successive nodes. Methods directly employing genomic data have access to this amount of divergence, and can therefore not only incorporate it into the analysis but also eschew some of the simplistic assumptions of methods such as equal or mbl (which remains true regardless of the latter being very common in the literature). It is wrong of the authors to assume that time calibration relies exclusively on the use of the right fossils (although this is certainly crucial), and there are good reasons why approaches incorporating genomic data to divergence time estimation are unambiguously better than those that don't, even when molecular data is available only for extant terminals. I strongly believe that this caveat needs to be made explicit in the main text of the manuscript, and that it is not a problem that disappears just because several different methods explored agree with each other.

3) In a similar vein, I would suggest the authors to explicitly state that some results that directly depend on the estimation of time-calibrated branch lengths (such as the calculation of evolutionary rates) necessitate a confirmation with more robust methods of time calibration, such as those that incorporate molecular data alongside the fossil record, as well as employ more realistic models of diversification such as the fossilized birth-death prior.

---

## [Author Response]

Essential revisions:The present work should be of interest to evolutionary biologists concerned with macroevolution as well as taxonomic specialists. The main conclusions are supported by the data with caution, and they complement and refine some other recent results. The authors go to considerable lengths to rule out alternative explanations. Overall, while the conclusion of this manuscript is novel, many concerns addressed by the reviewers (below) should be considered for revisions and improvement. The manuscript could be published without the need for new analyses, although we would like to see more, especially how rates are affected by enforcing divergence times more in line with molecular trees.

We agree with the reviewers in that the divergence ages provided by some methods (mainly the Hedman method) are older than those reported in other studies. However, the use of the Hedman method upon a different dataset (e.g. Herrera-Flores et al., 2021a) yielded similar ages. In that case, the Hedman method was the only method considered, when we have instead performed the analyses according to two additional dating methods that yield more recent divergence ages estimates. According to this, it cannot be considered that results are tied to the old divergence times resulting from applying the Hedman method, because they are recovered when using the equal method too. Our results using the MBL method are admittedly younger, but we think that the MBL suffers from some sort of pull of the recent effect. We also performed a preliminary Bayesian analysis to see if divergence times were in line with those from the parsimony methods just noted. Surprisingly, the ages recovered using MrBayes (preliminary results provided for evaluation, but not implemented in the manuscript) are more in line with those of the Hedman method (see the attached MCCT tree) than with the equal or MBL methods, which both provide younger ages. We are planning to repeat analyses forcing specific time divergences according to molecular studies, but this approach will require a careful assessment of the ages of fossils in our dataset that could represent older records than the ages of given clades in molecular analyses.

If new analyses are not performed, then the manuscript should be toned down to account for the many sources of uncertainty that permeate the present analyses, and acknowledge the ways in which the results depend on methodological decisions that could be improved upon.(1) The discussion of ecology (lines 95 and following) is oversimplified (as the authors recognise), perhaps inevitably so.The authors recognise that the "ecology" of the clusters discovered by the authors are difficult to generalise (what is the ecology of limbless forms that unites scolecophidians and boas?). Given that, we would consider striking "ecology" from the title and focus just on disparity.

This failure to recover fine-tuned ecomorphological clusters is a recurrent problem in studies of lepidosaurs, and seems to be independent of the type of dataset (geometric morphometrics in Watanabe et al., 2018 or Herrera-Flores et al., 2021a vs discrete morphological characters like in Simões et al., 2020, Martínez et al., 2021 or the present study) or differences in sampling (a broader sampling of non-lepidosaur diapsids of e.g. Simões et al., 2020 vs our broader sampling of lepidosaurs, specially Mesozoic squamates). In our opinion, the lack of resolution among ecological clusters is related to the fact that the latter are superimposed on general bauplans that sometimes have a greater weight in the general distribution in morphospace. Just as an example, snakes and other limbless taxa are plotted together, but resolution is not good enough to separate fossorial vs marine vs surface dwelling taxa inside this main cluster. The truth is that we are looking for broad-scale diversification events – e.g., we are more interested in the timing of the event that led from a limbed lizard-like squamate to the earliest limbless morphotype or from a lizard-like headed form to a snake-like headed form, than to subsequent changes once a bauplan is achieved that might have occurred multiple times, like adaptations to fossoriality or to a marine environment inside a particular group. A more detailed study of particular groups might be interesting in many ways, but it is unlikely to provide information regarding the initial diversification of squamates, which is the focus of our paper.

Although we agree with the reviewers that the term ecology would be setting perspectives not reached in the text, the term “ecology” is not used in the paper title or any subtitles.

(2) While we believe that the authors have done sufficient due diligence to rule out potential confounding factors, it would still be worthwhile to be more explicit about one limitation, namely the issue of missing data and inapplicable characters. For instance, this paper could be brought up:Gerber, S. (2019). Use and misuse of discrete character data for morphospace and disparity analyses. Palaeontology, 62(2), 305-319.We are not convinced that a broad-scale expunging of "inapplicable" characters (say) is the best way to solve the issue (one thinks of limb character in limbless forms), but a more explicit acknowledgment of this limitation would make the paper more compelling.

We agree on the importance of the limitation of missing and inapplicable data, which on the other hand is a recurrent problem identified in squamate phylogenetic data matrices (also for conducting phylogenetic analyses in particular). According to Gerber (2019), the inclusion of inapplicable characters is not advisable, but we concur with the reviewers that simply removing the inapplicable characters does not seem to be the solution; in fact, an alternative approach circumventing this problem still needs to be devised. To test the impact of missing data we performed an additional analysis where we removed all characters that presented a percentage of missing data above 40% (note that because missing and unscorable entries were both assigned a question mark in the original matrix, they cannot be treated differently). In any case, it seems that other works (e.g. Martínez et al., 2021) ended up considering both missing and inapplicable characters as ? in the morphospace analyses, so a separation of ? and – states does not seem to be necessary at this stage. The procedure described above reduced the matrix from its original 836 characters to just 184, but contained a total amount of missing data just below the 25% of missing data considered safe for interpretation. The morphospace resulting is different regarding the position of specific taxa, but still recovers the same main morphospace groups in the same positions as in the full analysis. This confirms that our main results are reliable despite the huge amount of missing data. We suspect (but have not investigated in detail yet) that those characters with a huge amount of missing data were barely contributing to the first axes, explaining the weak effect of their removal in morphospace for these main axes. We have added a comment on the limitation as discussed by Gerber (2019) where needed but we have not included this additional analysis in the results (we are open to doing so, however, if the reviewers or editors find it necessary). Note that one of the performed analyses in the original manuscript already focussed only on osteological characters, which had a similar effect of boosting completeness (mainly among fossils). Another potential problem, the express removal of autapomorphies and/or the supposed lack of collection of autapomorphies seems not to apply to the current matrix because multiple autapomorphies were identified in Conrad (2018), either in the form of autapomorphic binary characters or one or more states in multistate characters, and they were not deactivated. Regarding the true incompleteness of fossils, this cannot be avoided without deleting incomplete fossils. We are reluctant to do so because we think they still are the only source of true (not inferred) morphological information in specific points of the fossil record.

(3) One weakness is the issue of completeness, which has been raised prominently in recent years. The authors approach this by way of examining Cramér coefficients and throwing out characters that are generally missing for fossils (like those of integument). While this is not unreasonable, and results in disparity plots that are broadly comparable to analyses in which these characters are present, there remain other characters (especially those related to the limbs), this problem was less thoroughly addressed than others.

Again, we cannot disagree with the reviewer regarding the importance of completeness and unscorable characters (which effectively act as missing data). However, true incompleteness is inherent in the use of fossils, and removing those characters that concentrate the greatest amount of missing data barely affects results. The percentage of missing data across the entire matrix could be reduced by removing fossils, but even when they are incomplete, fossils provide information regarding which sets of character states were present at specific points in time. This is precisely why they were included originally, and removing them only seems advisable when they are revealed as unstable taxa that are recovered in different points of the tree (although this situation was considered by using multiple randomly selected most parsimonious trees). In the absence of fossils, or of most fossils, our only information on trait distribution in the past would be based on ancestral states estimates, and actual fossil data has been shown repeatedly to be preferable to ancestral data inferred only from modern taxa.

For each variable we want to consider, we are forced to repeat every other analysis performed in order to confirm that any difference in the results is related to this variable in particular. Considering that the time elapsed for some of the performed analyses can reach a few weeks (because the morphological matrix is very large), there is a limit to the number of options one can reasonably check. We consider the issue of incompleteness (via partially preserved taxa in the case of fossils, or unscorable characters in the case of e.g. snakes) a valid concern and we have added mentions to the problem in the text, but we feel this cannot be easily addressed at this stage. As mentioned above we performed, however, a complementary analysis where we deleted all characters that showed more than 40% of missing entries. This results in a strongly reduced matrix that, when analysed, yields very similar results of morphospace distribution. This is interesting in two ways: (1) it seems that those characters with great amounts of missing data contribute little to distributions in morpohospace along the main axes and (2) characters related to the postcranium concentrate the vast majority of characters removed because they are unknown in many fossils, and they are unscorable in limbless taxa. This result is also interesting in showing that the separation of limbless taxa (e.g. snakes and amphisbaenians) is strongly supported by skull characters too, probably due to strong skull adaptations of these taxa to fossoriality and specialised feeding, with independence of the lack of limbs.

4) One of the striking aspects of the results is that the interpreted expansion of morphospace in both the Jurassic and Cretaceous occurs before the interpreted spike in evolutionary rates (not exactly "roughly coincident" as stated). This seemingly counterintuitive result is not addressed.

Actually, the new figure 4 shows that the Jurassic peak in morphospace and evolutionary rates are coincident (note that the disparity peak is placed in the midpoint of a 24-Myr long bin). We admit that the Cretaceous peak in disparity occurs earlier than the Cretaceous peak in evolutionary rates, but the manuscript focussed on discussing the importance of the Jurassic peaks. It is worth noting that the Jurassic peak in evolutionary rates in the late Jurassic is indeed as coincident as it can be according to the disparity metric binning, which shows its peak in the time bin spanning from 168 to 144 Ma. This bin spans a very minor portion of the latest Middle Jurassic (the latest ~ 5 Myr), the entire Late Jurassic (~18 Myr), and the earliest Early Cretaceous (less than 2 Myr). The middle point of this span, which is the one used to graphically represent the value obtained, is 156 Ma, thus well inside the Late Jurassic. Morphospace occupation represents Middle and late Jurassic taxa together, which results in a representation that suggests that morphospace occupation occurred earlier than the disparity or evolutionary rates peak. Note, however, that in the case of squamates, all sampled taxa are Late Jurassic in age because Middle Jurassic taxa were not sampled due to incompleteness of the fossils. One might expect that the inclusion of Middle Jurassic taxa would move all peaks towards the Middle Jurassic but, given the limitations of the fossil record and forced incongruences in time binning, we regard the results as congruent enough to infer that they all hint to the same event.

We acknowledge that the situation is not ideal, but all these issues are common to all such deep-time studies and easy solutions are not available to remove such uncertainties. Because the requirements for each metric and plot are different, we decided to maximise resolution rather than consistency across bins (at the cost of lower resolution). Moreover, and as expressed by the reviewers in other parts of the review, there is no need for evolutionary rates and disparity to be tightly related, although if they are (and we consider that this is a real possibility), they are more likely to be related to a specific event.

5) As someone who has performed principal coordinate analyses on lepidosaur morphological datasets, one reviewer can confirm that the first few dimensions invariably separate limb-bearing from limbless forms, as well as rhynchocephalians from squamates (in the case of this analysis, anguimorphs seem to also contribute to this axis). This is exactly what is found in Figure 1b, but also previously in Watanabe et al., (2019) and Simoes et al., (2020).

Yes, and this was already acknowledged in the text. Also note that we have shown that even if the first dimensions separate limb-bearing from limbless forms, this is not necessarily related to the presence vs absence of limbs, but possibly to strong modifications of the skull, explaining why Watanabe et al., (2019) were able to get this separation with a dataset comprising only skulls (without postcranial information).

As such, separating these four groups (rhynchocephalians, snakes, anguimorphs and "generalized lizards", as defined in lines 95-104 by the authors) explains a large proportion of total variance, which loads prominently on the first axes but is likely to contribute to some extent to others. Given that the matrix being used was "designed to identify the relationships among the major squamate clades" (Conrad 2018), it is no surprise that the same pattern emerges in this study. The "Jurassic morphospace expansion" (JMA) proposed here is nothing more than a period of time when a morphospace occupied by only one of these groups (rhynchocepalinas) gives way to one occupied by all four.

Yes, and this was already acknowledged in the text. Also note that we have shown that even if the first dimensions separate limb-bearing from limbless forms, this is not necessarily related to the presence vs absence of limbs, but possibly to strong modifications of the skull, explaining why Watanabe et al., (2019) were able to get this separation with a dataset comprising only skulls (without postcranial information).

This result is not novel (i.e., does not depend on any discovery made in this study), it is just presented here in a novel and quantitative way.

Precisely, this is acknowledged when Evans (2003) is mentioned. However, going from the crude interpretation of the fossil record and phylogeny to quantitative analyses and finding support for this overlooked idea (most works tend to talk about the Cretaceous, not the Jurassic, as the key period for squamate diversification) is important.

In fact it is already stated in some of the first lines of the introduction that these morpho-groups are known to already be present in the Middle and Late Jurassic (lines 47-49), and that a similar pattern had already been mentioned in the literature (line 238). Furthermore, whether this does represent a true event of expansion or an artifact of the fossil record is also unclear given the poor fossil record of the clade during this period, as noted by the authors. Furthermore, it ultimately depends on many methodological decisions not entirely validated (such as temporal binning, choice of distance metric, etc), as well as character and taxon sampling decisions that were not optimized for the purpose of macroevolutionary inference, and might therefore supported biased patterns, not just during the Jurassic but also later on.

We understand and, to a certain degree share, the concerns of the reviewers regarding the potential impact on methodological decisions. We have tried to take all these variables into account by repeating analyses according to different topologies, temporal binnings, distance metrics, dating methods, etc. We ultimately decided to apply the temporal binnings that optimised resolution for each analysis, even if in the process we sacrificed some consistency when plotting the results. For disparity, we chose two widely used metrics: the weighted mean pairwise distance (WMPD), which is a pre-ordination matrix; and the sum of variances (SoV) which is a post-ordination matrix, in order to assess the effect of choosing one or the other type. Regarding character and taxon sampling, most morphological data matrices used in these kinds of studies were initially built for conducting phylogenetic analyses. The same observation would apply to other macroevolutionary studies, including the mentioned work of Simões et al., (2020) but also Martínez et al., (2021), and many others concerning other groups. Regarding taxon sampling, we also took this into account when we chose the Conrad (2018) matrix, which is the most complete in terms of both extant taxa and fossils. We refer the reviewers to Smith et al., (2021), who state that using morphological matrices intended for inferring phylogeny as the source data for disparity analyses is acceptable as long as the tree is balanced in terms of sampling. We think that the matrix used here presents a very good balance between fossils and extant taxa, as well as across different clades and through time. We made some initial runs of the analyses with other smaller matrices (Gauthier et al., 2012; Simões et al., 2018), but we decided not to use them because the poor sample of fossils outside the late Cretaceous apparently biased results. We would expect that increasing the number of Mesozoic taxa by choosing Conrad’s (2018) matrix would contribute to a greater resolution and a more reliable result. The same applies to the number of characters. It would be hard to justify the use of any other matrix containing a lower number of taxa and characters, unless it was implied that a given matrix had issues regarding the quality of the chosen characters.

That being said, and acknowledging that there are not many alternatives to the choices made, we recognise that some patterns might be influenced by the unevenness of the quality of the fossil record. This is already stated in the text, and is an unavoidable effect of the nature of the fossil record. However, using the alternative (ignoring the fossil record and inferring morphologies from extant taxa) introduces, in our opinion, an even greater bias, among other things because control on the exact time when a given morphological character state appears is lost.

Although all the issues reported by the reviewers are potential sources of bias, and this is acknowledged throughout the paper, only those choices consistently leading to a concentration of time divergences in the mid-late Jurassic are potentially biasing patterns towards the result we present in our manuscript. We have shown that applying Bayesian inferences (new results provided in this review, but not incorporated to the manuscript) does not yield strikingly different results in dating the trees (i.e. Bayesian results are inside the range of dates recovered by the other three methods used). A higher amount of divergence times in the Jurassic (compared to other studies) can be also explained by the proportionally larger amount of Jurassic sampled taxa (in contrast to other matrices). Other studies have used a very limited number of Jurassic and Early Cretaceous taxa, and this invariably leads to younger divergence times for key clades– no matter which method is used for dating, removing key taxa, specifically among the oldest ones, is expected to move divergence times towards the present. There is another important observation. All these datings feed upon the current knowledge of the fossil record. We have already expressed our opinion that our sample of Mesozoic squamates is better than that of other studies, but another concern seems to be the quality of the fossil record itself and associated completeness. Note, however that (1) this is unavoidable because the fossil record is invariably incomplete and, more importantly (2) improvement of knowledge of the fossil record is likely to support our results because any lineage newly found in the Jurassic has the potential to move divergence times towards older ages. The opposite would necessarily imply massive reinterpretations of the affinities of many pre-Late Cretaceous forms. Even in the latter case, the morphospace occupation would not change because it is not tied to a given phylogenetic interpretation.

Regarding the mention of the fact that not all methodological decisions have been entirely validated, we commit ourselves to fully explore alternatives to such methodological choices in an ongoing study that will complement the present manuscript (but we provide preliminary results to the reviewers in order to justify our claims regarding the fact that many of the proposedly conflictive factors have a weak influence upon results).

(6) The discovery of this peak in disparity if further linked to high rates of morphological evolution. Note however that disparity and rates are not necessarily linked, and high rates are not necessary to support a JME. However, this result is entirely dependent on the topology and divergence times supported by this study.

Although evolutionary rates as calculated depend on the topology and dating of the trees, it is important to note that differences in topology were indeed considered in two complementary ways. The main results refer to the constrained analysis, where a backbone constraint was used to force the general interrelationships among extant groups, but a second set of results feeds on a second set of trees resulting from an unconstrained analysis, so both schemes have been considered. Moreover, for each of the two schemes, several MPT’s were randomly selected, and the curve of evolutionary rates considers when evolutionary rates are consistently recovered among different topologies and dating of the trees. Regarding the latter, it has been explained above that the three methods give different divergence times for key clades, but the high evolutionary rates in the late Jurassic remain a constant result, except for the mbl method. In the latter, the concentration of nodes close to the present day results in increased evolutionary rates for these associated short branches. It is worth noting, however, that the Jurassic and Cretaceous peaks are still recovered, they just become less significant because of the massive boost reached by branches near the present. Also, we can now state that the use of alternative methods for dating the trees (Bayesian analyses) are consistent with our results in terms of dating (they are intermediate between those of the Hedman and Equal methods), and in terms of which branches have higher rates of evolution. This is in part related to the fact that the branch leading to snakes, which is invariably situated in the Jurassic, has the highest evolutionary rates of the entire tree by far (together with other branches inside this same clade). This same result regarding high evolutionary rates in the branch leading to snakes was identified in Simões et al., (2020) although in their case Jurassic taxa were not sampled, what presumably could have moved branches (and associated high evolutionary rates) towards earlier times.

The methods of inference and calibration employed here are much more simplistic than those of previous studies, and allow for a much poorer integration of uncertainty into subsequent analyses. Results differ crucially from those of all previous studies, including a much faster early diversification of squamates, and much older ages for some crown clades such as snakes, anguimorphs and iguanians.

Older ages than usually found for crown clades are only recovered in the figured tree (Figure 1), dated with the Hedman method. Note, however, that evolutionary rates were calculated for all three methods of dating, which in the other cases (e.g. equal) do not yield significantly older ages for crown clades, and still recover high evolutionary rates for the late Jurassic (see explanation above). Moreover, as explained above, we have now used Bayesian methods for dating the trees, demonstrating (see attached files) that the Bayesian trees are inside the range of ages recovered under other methods. Methods in some other recent studies include extensive genomic data as well, but such data is only available for modern taxa and can only change the overall tree topology, but not the dating of deep divergences, as here.

This leads to a concentration of branches with high levels of morphological change in the mid to late Jurassic, ultimately responsible for the peak in rates found. The methods employed and the way in which they are presented once again do not rule out the possibility that this is an artifact of the way the study was set up, and a major reason why no such increase in rates was discovered by previous studies.

This is assuming that all methods used recover older divergence ages than other studies, which is not the case. As explained above, the ages we get for the equal method are younger than those we got using Bayesian methods upon the current dataset, and they also recover the Jurassic peak. In any case, and even if we were getting slightly older ages than other methods, it is important to note that because they are usually fragmentary, most of the earliest fossils of clades are not included in the matrix, and thus divergence times recovered in all these studies are probably slightly younger than they would be if we included all available fossils. We are considering including these fossils as calibration points in the Bayesian analysis dating, what would allow to use their temporal information without decreasing the resolution of resulting trees.

In any case, we agree with the reviewers in that the recovered high evolutionary rates in the late Jurassic are not necessary to support the observed Jurassic Morphospace Expansion. These are linked in the paper because appear to be roughly contemporary (more on that later). Again, evidence of the fact that we were aware of the importance of properly dating the trees is reflected in our use of three methods (minimum branch length, equal, and Hedman) for this purpose, and comparing their results regarding evolutionary rates. Note that the divergence times vary from one method to the other, but the late Jurassic peak is consistently recovered, except for the mbl method, that we consider is stretching nodes towards the present because of the high proportion of extant taxa. We recognise that since we designed the study alternative methods for dating the trees have appeared and been used. We think that it is unlikely that using any of these new methods would dramatically change the results because the range of recovered datings is already wide and, in fact, we preliminarily demonstrate that using Bayesian inference makes little difference. In our opinion, failure of previous studies in recovering the Late Jurassic peak in evolutionary rates is more related to their poor sampling of Jurassic taxa than to their use of a more sophisticated dating method (see below, however, regarding our decision to provide a statement that our results will be further validated by an ongoing analysis using Bayesian methods).

(7) There is a lack of consistency throughout the manuscript. For example, the JME label applied to each of the four panels of Figure 1 seems to represent four different periods of time. The topology in Figure 1A was calibrated using a different method that the rates shown in Figure 1D, making it difficult to draw conclusions.

We acknowledge a lack of consistency, with two different sources: one is related to the conscious decision to favour resolution across the different measures and plots above consistency. This results in minor disarrangements in the exact time when an observation is represented. Because of this lack of consistency and the inherent error margin resulting of the use of relatively wide bins, we consider that a single event might be the cause of all these patterns. Figure 1 (former figure 1A) no longer shows the JME, but we have represented evolutionary rates and disparity (Figure 4A and B) in a way that shows that both results show a peak that is coincident in time. Regarding the methods used in figures 1A and 1D (now Figure 1 and Figure 3D), there was an error in the label, as they were both calculated under the Hedman method. Note also that results for all methods are reported in the supplementary data.

While a staggering 51 supplementary figures are presented, many of these are unedited and thus highly cryptic, others such as the many ordination plots are unlabeled and thus do not contribute much.

We think that the fact that the figure legends of the supplementary figures are placed in a separate list might make interpretation more difficult, so we are open to moving each legend below the corresponding figure, and also try to add some information that make them more readable. If the reviewers are asking us to label individual points of plots, this is very difficult for 200 points in each plot, but it was done in supplementary figure 8, which can be used as a guide to know the position in other plots (in the case of PCO1-PCO2, in figures 11,12, 14-19). Finally, it must be said that one can easily get any label plotted by slightly modifying the R code and running it again.

Some aspects of this manuscript are very interesting, such as the position of extinct lineages under topologies constrained to show different relationships among major extant clades. However, its main claim is somewhere between trivial (a change from 1 to 4 morpho-groups somewhere in the second half of the Jurassic) and potentially unsubstantiated (a coincident peak in rates).

Although we acknowledge that an early (Jurassic) diversification of squamates had been already pointed out (e.g. Evans, 2003, already mentioned in our text), this claim had never been tested using numerical methods implying disparity of evolutionary rates. Besides this, we challenge the widespread idea that not much occurred in the evolution of squamates until the apparent radiation of crown groups in the mid-Cretaceous. Moreover, an important observation is that any Jurassic fossil newly added to the dataset (because it was formerly not sampled or because it is newly discovered) is likely to increase support of the proposed diversification event by (1) moving divergence dates to older times; (2) proving a given clade was present in the Jurassic and (3) likely increasing fast evolutionary rates for that period. The opposite (this is, demonstrating that the Jurassic event is an artefact of the methodology used) would probably require that many of the Jurassic fossils were reinterpreted as rather basal forms that were not related to the diversification of the crown, and/or using a dating method that gives very young divergence times for crown clades even when Jurassic taxa are included, but divergence times cannot be younger than fossils in the clades concerned! We aim to provide additional evidence for our claims in the complementary manuscript we are preparing. The fact that other recent studies using similar methodologies have failed to recover some of the patterns presented here or have ignored them, shows that it is indeed interesting to present and discuss them. In contrast to Martínez et al., (2021), who recovered stem lepidosaurs overlapping in morphospace with rhynchocephalians for PC1-PC2, in our results the overlap is between stem-lepidosaurs and squamates for both PC1-PC2 and PC3-PC4. This seems to contradict the assumption by Martínez et al., that rhynchocephalians would present a conservative morphology. This supposed plesiomorphic morphology of rhynchocephalians has been previously claimed, but current studies suggest that at least some of the characters that were supposed to support it are controversial or their polarity was simply reversed (e.g. the presence of acrodont teeth or of a complete lower temporal bar).

The current narrative, emphasizing a completely new and unexpected view on squamate diversification might be true, but it rests on many assumptions. A more realistic interpretation should focus on the existence of four morpho-groups within squamates (which could even be tested with this data, see below), and their timing of origin. The manuscript starts by posing the question "why are some groups highly species-rich and others are not?" (line 16). This question is never answered, and in fact the manuscript does not seek to do so or use data that could. But that kind of exaggeration permeates the rest of the manuscript, and the reviewer thinks it will be benefit from a more realistic approach to the data at hand.

We agree that the manuscript does not seek to answer that particular question, which has been removed, but it investigates the timing in which squamates became morphologically diverse, and if this process was fast (fast evolutionary rates) or not. These are important questions that need to be addressed in order to attempt questions with a wider scope.

8) One reviewer thinks the paper should validate some of their methodological choices better, or even better improve upon them. For example, two disparity metrics are employed (one of which does not even show a JME), but no reason is given why these were chosen. Results from other post-ordination metrics should be presented as well, along with some validation of why certain metrics where favored. This is implemented by Guillerme et al., 2020 Eco Evo and employs some of the same programs already used.

We refer the reviewers to Gerber (2019):

“For the construction of disparity curves, I do not recommend carrying out analyses from the morphospace ordination (‘post-ordination disparity’ in Lloyd 2016) but instead obtain disparity measures directly from the distance matrix to minimize the impact of missing data”.

Regarding the two metrics employed, we chose two of the most widely used metrics, one of them is a pre-ordination metric (WMPD) and the other one (SoV) is a post-ordination metric. Gerber (2019) recommended the calculation of a pre-ordination metric to minimize the impact of missing data, so we have added a disclaimer regarding the possibility that the SoV results might be affected by missing data. We are reluctant to calculate additional postordination metrics following Gerber’s (2019) advice.

Furthermore, a large part of the findings relies on time-calibrated analyses that are poorly explained. For example, although 3 methods of post hoc scaling of topologies are explored, only 1 topology is presented as a supplementary figure. Details on the methods themselves are missing, as it is unclear what time information went into the calibration and how it was gathered and treated. Ultimately however, branch lengths scaling seems to be resulting in unlikely ages for many clades. The reviwer believes results need to be repeated using a topology calibrated using molecular data, or at the very least including secondary node constraints based on other published molecular studies.

Regarding the dating of the trees, we selected one of the dated trees (that resulting from the Hedman method) to illustrate the full results. Note, however, that we performed separate runs of evolutionary rate analyses including the trees resulting from the two other methods, and for each method, several randomly selected trees are used and dated multiple times. According to this methodology, the dating choice is actually considered when interpreting the results, and uncertainties related to specific topologies too. We have added now a figure of a randomly selected dated tree where the range of ages for each node according to the three methods is shown.

The reviewers state that there are no details for the methods used. The mbl and the equal method have been widely used (as implemented in the Paleotree package), and we have added a sentence in the text referring the readers to the package documentation. Regarding the Hedman method, we have added a sentence explaining details on the methodology. The time information was gathered from the Paleobiology database as FAD (first appearance datum) and LAD (last appearance datum) for each sampled taxon. This is the standard way to introduce time information in the methods used for time-calibration (mbl, equal and Hedman).

As expressed elsewhere in our answers to the reviewers, we are preparing a manuscript for submission as a Research Advance to *eLife* that will deal with alternative methodologies (e.g. Zhang and Wang, 2019) for dating the trees that preliminarily support our results. It is possible to use the same matrix (Conrad 2018) and run a Bayesian analysis that reports a single dated (majority rule tree) and the corresponding evolutionary rates. Again, we will calculate results for both a constrained and unconstrained phylogenetic analysis. Moreover, we will use the version of the method that deals with combined morphological and molecular datasets in order to test if the matrix of Gauthier et al., (2012) recovers some signal for the proposed event in the middle-late Jurassic. The latter approach will be interesting in that the possibility that the Jurassic peak in evolutionary rates is also detectable using smaller matrices will be tested. In the meantime, we have included a explicit statement explaining that the relevant conclusions require additional supporting data that will be published in an ongoing study.

One aspect that could also help improve this paper is an exploration of morpho-functional groups within morphospace. Currently, traditional views are simply repeated (see above regarding the four groups of lepidosaurs) and forced onto the results rather than extracted from them.

Although a more elaborate approach was considered in early stages of this study, it has some limitations. The main problem is that half the sampled taxa are fossils, and as such, their ecomorphology can only be inferred. Thus, one of the possibilities would be to assign an ecomorphological group to each of the extant taxa and use the plotted position for fossils to infer their ecomorphology. An alternative approach, to assign inferred ecomorphology to fossils according to literature, renders the argument circular. The strict approach (assigning ecomorphology for those taxa for which it is confidently known, that is, extant taxa) strongly reduces the number of truly sampled taxa (to around half of the original) and relies on the assumption that the dataset is capable of separating discrete ecomorphological groups and then correctly assign fossils to them. We have doubts regarding the viability of the methodology because, at least on some occasions, the phylogenetic and ecomorphological signals are too strongly mixed to provide a clear output at the species level. As an example, the phylogenetic signal is strong enough to hold marine rhynchocephalians and marine squamates (mosasaurs) separate. The great overlap of “generalised” lizards is another example of the limitations of the proposed exploration. Please note, however, that this same limitation has been previously encountered and is likely to derive of the way characters are distributed in lepidosaur morphological matrices. We suspect that the widely claimed high degree of homoplasy among squamates that is apparently behind the lack of correspondence between molecular and morphological phylogenetic analysis might be behind this poor resolution too. Apparently, morpho-functional signal is strong enough to force artificial groups like the limbless taxa in morphological analyses, but the phylogenetic signal is strong enough to recover many of the true clades (according to molecular data) too. Although the exploration of morphospace proposed by the reviewers is definitely interesting, we think it cannot be easily achieved, and we knowingly simplified discussion so the considered morphospace groups could be tracked through time.

What the data at hand could help establish (possibly along other datasets that are said to have been dropped due to poor sampling of Jurassic fossils) is how many morphological clusters of taxa can be found (using k-means or expectation maximization algorithms for example), and when in time do those originate based on the topologies at hand (not just the one used here but those previously published as well).

We thank the reviewers for the suggestion of the use of k-means and expectation maximization algorithms, which is very interesting, and it will be explored in our ongoing follow-up study. As an alternative approach, we have used the R function pamk of the fpc package in order to get statistically supported clusters. We have added a plot to the supplementary figures showing that when the function is set to group points into 2-4 clusters, the groups we get are almost identical to the ones we used in former figure 2B (now Figure 3b). Allowing for a greater number of groups (8 or 16) gives a much larger number of clusters (not shown) that, on the other hand, gradually increase in similarity to the groups observed in the extended data Figure 11 (this is, phylogenetic groups rather than ecomorphological groups).

(9) The earliest fossil records of the crown squamates are the Middle Jurassic. All these records (for some constituent major clades) in the Middle Jurassic should be clearly stated and/or cited in the text.

A mention to the fact that the Middle and Late Jurassic contains the earliest unambiguous squamates, and also members of the crown is stated in lines 53-56. No more detail is provided because these forms are in many cases not included in the dataset, mostly because they are too incompletely preserved and tend to decrease the resolution of the phylogenetic analyses. Middle Jurassic forms are not sampled, but if anything their inclusion would favour our interpretations.

As one reviewer thinks, this is why the authors separated Early Jurassic from Middle-Late Jurassic in their analyses. The criteria for this separation should be discussed. Otherwise, the readers would doubt why you did not separate Early-Middle Jurassic from Late Jurassic. Figure 1a, which shows the earliest record of the crown squamates later than the Middle Jurassic, is misleading in this regard.

Regarding the fossil record of squamates in Figure 1A, it is important to clarify that the represented record is not the global record of the clades, but the recorded range according to the distribution of sampled taxa in the matrix used. This is the reason why squamates are not recorded until the late Jurassic. This was (and is) explicitly stated in the caption of Figure 1a (now Figure 1).

As we acknowledged above, Middle Jurassic squamates do exist, but they are in almost every case too fragmentary to be included (and they were not in the original data matrix). In this sense, it made more sense to join two time bins that actually contained squamates (even if those from the Middle Jurassic are not sampled), than merging the Middle Jurassic with the Early Jurassic. The chosen binning also resulted in a more balanced distribution of time. Considering that our peak in evolutionary rates and disparity occur in the Late Jurassic, the time binning (early+middle and separate late Jurassic) proposed by the reviewers would fit best our interpretations. However, the inclusion of Middle Jurassic fossils would potentially move the event to the Middle Jurassic.

(10) The manuscript should be edited carefully following the author guidelines of eLife. Some of the discussions in the supplementary information text can be moved to the main text to gain a better readership. The Discussion and especially Extended Discussion would benefit from some further subdivision.

We have added some more weight to the main discussion, and have placed some subdivisions, mainly in the extended discussion.

Some of the supplementary figures (Extended Data Figures 1-10), which are not well demonstrated, can be deleted or replaced with a batch file.

We have replaced Extended Figure 1 with a figure of a randomly selected MPT dated with the Hedman method and showing the ranges of the ages according to all three methods. We are not sure what the reviewers mean with the expression “which are not well demonstrated”, so we retain all supplementary figures until the reviewers clarify which can safely be deleted.

Overall, the supplementary figures should be re-organized to have a uniform format. The authors should put more effort into the supplementary figures, many of which are unintelligible.

We have improved the supplementary figure captions where possible to introduce information that helps in interpreting them. Considering that these are just supplementary figures, and that are the result of the use of multiple packages in R, providing a uniform format is beyond something that can be easily done. However, we hope that the fact that each figure will appear together with its own caption in the published version will help in its interpretation.

Other claims that need to be revisited:(1) Line 132 – "These evolutionary rates are not dependent on phylogeny": They are very dependent on phylogeny. This analysis shows they are not dependent on topology (molecular vs. morphological) but both of those are likely to share very similar branch lengths for all branches in common, making the result still strongly dependent on phylogeny.

We revise this. We probably should have used the term topology here (we have changed the text now). What we were trying to express was that discrepancies in topology (both at the level of large groups like we see in the constrained vs unconstrained results, and at the species level representing minor changes in topology) were considered by including multiple topologies in the analyses.

(2) Lines 172-173 – "The second event, which fits well with the timing of the KTR, would be coincident with the radiation of the constituent crown groups of Squamata": Figure S1, which is the only time-calibrated topology shown in its entirety, shows crown snakes, anguimorphs, iguanians, lacertoids and skinks originating in the Jurassic, and radiating well before the KTR.

Revised. We are trying to convey a distinction here between the radiation of the crown group Squamata into their corresponding main groups (occurring in the middle-late Jurassic, when the first representatives are found) and the observed main radiation of each of these groups, which seems to be delayed until the KTR. Regarding the latter, we have argued that this is likely an artefact of the fossil record, that suddenly improves in the middle-late Cretaceous.

(3) Lines 189-191 – "The only observable changes from then on correspond to a slight expansion of the edges of the occupied morphospace, and a notable increase in the density of points filling this morphospace": This is entirely contingent on analyses using a matrix that was tuned to capture the deep phylogeny of the clade, and is something that should be acknowledged.

Revised. The problems of using matrices originally built for phylogenetic analysis have been discussed above. However, we want to note that the same main morphotypes could have appeared much later (e.g. in the Late Cretaceous), and they would not be supporting the JME, but the KTR.

(4) Line 271 – "basal position of Iguania": This and other instances of the use of the word basal should be corrected, living groups cannot be basal to other living groups.

We have replaced this expression by “sister-group relationship to the rest of squamates” in this case, and the appropriate expression in the rest of cases (in the supplementary text).

(5) The reviewer has not been able to find which dimensions were considered to estimate the disparity curve shown in Figure 1C. Please add this information to the caption.

Disparity in former Figure 1C (now Figure 4a) corresponds to the Sum of Variances and thus was calculated from the ordinated data, considering all axes data. We have added a statement in the text.

(6) The reviewer would consider striking some comments from the body of the paper that seem more like personal annotations ("We expected that….").

We have modified these comments following the reviewer advice.

(7) L.24: "the Middle and Late Jurassic (174-145 Ma)" is clearer than "the second half of the Jurassic (170-145 Ma)" to the readers, and also consistent with that in Line 48.

We use “the Middle and Late Jurassic” consistently now.

(8) L.82: Figure 1b-d as a separate figure?

Following the reviewer’s advice, and in order to improve the readability of the figures, we provide now Figures 1b and 1c,d as separate figures (new figures 2 and 4, respectively), and former Figure 1a has its own figure (new figure 1) now.

(9) L.90: For "pan-Squamata", do you mean the total group Squamata?

Yes, we have replaced the term (although pan-Squamata has been formally erected, see Gauthier et al., 2020, in Phylonyms, a companion to the PhyloCode)

(10) L.108, 171, 212: "mid-late Jurassic" Ambiguous here、Middle and Late Jurassic?

We use Middle to Late Jurassic now.

(11) L.145：'stem lepidosaurs' for 'ancestral lepidosaurs'

We have replaced ancestral by stem.

(12) L. 151: Extended Data Figure 11 as a main-text figure?

This could be easily done, and we are open to it if the reviewers and editors consider it necessary. However, we wanted to point out that this plot is redundant with former Figure 2B (now Figure 3B). The morphospace in both plots is the same, the differences are that in the main figure (former Figure 2B) the phylogenetic branches are shown and only the main clusters are labelled. In Extended data figure 11 we provided hulls, but the points are the same as in former Figure 2B (new Figure 3B)

(13) L.172, 175, 'late', upper case.

Done.

(14) Extended Data Figures 23, 24, 43-47, 'early', 'middle', 'late', upper cases.

Done.

Reviewer #2:Weaknesses:The discussion of ecology (lines 95 and following) is oversimplified (as the authors recognise), perhaps inevitably so.

We certainly recognise that a more detailed discussion on the ecology of groups appears as one of the expected outputs. We face, however, two main limitations here. One is that, even when you look at total morphospace, it is complicated to interpret ecology beyond the main four recognised groups. This is a recurrent problem in ecomorphological studies of squamates, and seems to be independent of the type of dataset (geometric morphometrics vs discrete morphological characters) or differences in sampling. In our opinion, the lack of resolution among ecological clusters is related to the fact that the latter are superimposed on general bauplans that sometimes have a greater weight in the general distribution. We plotted the ecology of taxa in the morphological plot (not shown) in order to see if we were able to make additional interpretations. The main problem is that there are local adaptations to different ecological roles inside of each clade, resulting in an overlap of ecologies that hinder interpretation. Note, however, that we do not regard this as a limitation of the methodology itself, but rather a problem derived from the complexity of the diversification patterns among squamates. This could be regarded as a result in itself and discussed further, but we decided to focus on surveying the timing of broad-scale diversification events, the main of which would have occurred during the Jurassic.

One weakness is the issue of completeness, which has been raised prominently in recent years. The authors approach this by way of examining Cramér coefficients and throwing out characters that are generally missing for fossils (like those of integument). While this is not unreasonable, and results in disparity plots that are broadly comparable to analyses in which these characters are present, there remain other characters (especially those related to the limbs), this problem was less thoroughly addressed than others.

We acknowledge that inapplicable characters are a source of uncertainty in the data, but there is not much that can be done to fix this issue. According to Gerber (2019), the inclusion of inapplicable characters is not advisable, but we concur with the reviewers in that simply removing the inapplicable characters does not seem to be the solution. Moreover, some of the performed analyses were designed to minimise the influence of incompleteness (e.g. the calculation of a pre-ordination metric like the WMPD). We have added explanations regarding the limitations of the methodology used (or of the matrix itself) where corresponding in the text, and we will explore this and other limitations in a subsequent study. We also run an additional analysis where completeness was raised by deleting characters containing more than 40% of missing entries. This resulted in a much smaller dataset, but results of morphospace are completely comparable to those of the full dataset.

One of the striking aspects of the results is that the interpreted expansion of morphospace in both the Jurassic and Cretaceous occurs before the interpreted spike in evolutionary rates (not exactly "roughly coincident" as stated). This seemingly counterintuitive result is not addressed.

We have provided additional explanations for this apparent contradiction in the text and in the detailed responses to reviewers. In the case of comparing the measure of disparity (SoV) and evolutionary rates, the peaks are actually coincident, as it is shown in the new Figure 4 (note that the layout of Figure 4A has been modified for consistency with the time binning of Figure 4B).

Reviewer #3:Bolet et al., explore the evolutionary history of lepidosaur morphology. Using a previously published dataset sampling a wide range of living and extinct lineages, they analyze both morphospace occupation and rates of evolution through time. By combining a series of phylogenetic and macroevolutionary methods, they reveal an event of morphospace expansion (coupled with increased evolutionary rates) during the second half of the Jurassic. This contrasts with previous studies, which had recovered a predominant signal of diversification and ecological innovation during the mid-Cretaceous, coinciding with the radiation of other major animal and plant lineages and a large-scale restructuring of terrestrial ecosystems. While their conclusion is novel, the manuscript is not based on any new morphological or paleontological data, and employs relatively simple methodologies that can potentially determine the outcome. My main concerns are the following:(1) As someone who has performed principal coordinate analyses on lepidosaur morphological datasets, I can confirm that the first few dimensions invariably separate limb-bearing from limbless forms, as well as rhynchocephalians from squamates (in the case of this analysis, anguimorphs seem to also contribute to this axis). This is exactly what is found in Figure 1b, but also previously in Watanabe et al., (2019) and Simoes et al., (2020). As such, separating these four groups (rhynchocephalians, snakes, anguimorphs and "generalized lizards", as defined in lines 95-104 by the authors) explains a large proportion of total variance, which loads prominently on the first axes but is likely to contribute to some extent to others. Given that the matrix being used was "designed to identify the relationships among the major squamate clades" (Conrad 2018), it is no surprise that the same pattern emerges in this study. The "Jurassic morphospace expansion" (JMA) proposed here is nothing more than a period of time when a morphospace occupied by only one of these groups (rhynchocepalinas) gives way to one occupied by all four. This result is not novel (i.e., does not depend on any discovery made in this study), it is just presented here in a novel and quantitative way. In fact it is already stated in some of the first lines of the introduction that these morpho-groups are known to already be present in the Middle and Late Jurassic (lines 47-49), and that a similar pattern had already been mentioned in the literature (line 238).

We agree that the results regarding this point are not completely novel (and this is made clear in the text), but our study confirms previous assumptions based on the crude interpretation of the fossil record and, more importantly, it contradicts recent studies using a similar approach that recovered a much later diversification of squamates. It thus contributes to an unsettled debate.

Furthermore, whether this does represent a true event of expansion or an artifact of the fossil record is also unclear given the poor fossil record of the clade during this period, as noted by the authors.

We agree with the reviewer that the fossil record is patchy and, as such, it is just providing partial information. To what degree this is biasing results it is unknown, but we want to note that our choice of morphological matrix and methods reflect our concerns regarding the fossil record issue by (1) including the largest number of fossils available (including many of the late Jurassic forms lacking in other data matrices) and (2) by limiting our conclusions to the information extracted from the data. Moreover, the poor fossil record in the Jurassic and the lack of middle Jurassic sampled squamates (due to incompleteness) are, if anything, weakening the signal for an early burst of diversification of squamates. Adding middle Jurassic taxa would move divergence times towards the past, and thus reinforce our claim of an early event in squamate diversification. It is hard to see, however, how an improved fossil record (or sampling) could move divergence times and associated evolutionary rates towards the mid Cretaceous. Moreover, if the poor fossil record is affecting results, the use of a lower amount of fossils in other studies should be even more problematic. To sum up (1) if the poor fossil record is a problem, it is even more prominent for other studies including a low number of fossils; (2) it is more likely to keep finding older and older representatives of the main groups of squamates than the opposite situation, this is, that we reinterpreted the oldest current examples of such groups in what would be the only way to move the origin of clades towards the present. In our opinion, the poor Jurassic fossil record is more likely to be underrepresenting the importance of Jurassic taxa in the establishment of lizard morphospace limits. Conversely, the enhanced fossil record of the Middle-Late Cretaceous could be artificially inflating measures of disparity, evolutionary rates or diversity for that period. According to this, we think it is worth considering the possibility that the Jurassic signal we get in our manuscript represents a true event.

Furthermore, it ultimately depends on many methodological decisions not entirely validated (such as temporal binning, choice of distance metric, etc), as well as character and taxon sampling decisions that were not optimized for the purpose of macroevolutionary inference, and might therefore supported biased patterns, not just during the Jurassic but also later on.

We agree that any methodological decision can have an impact on the results, and we have tried to provide many ways of auditing the effect of one choice as possible by trying multiple options. We have controlled the effect of topology by including multiple topologies of two different schemes: one is the result of a pure morphological phylogenetic analysis; and one is the result of a constrained analysis according to molecular studies; we have provided two measures of disparity; we have calculated measures of disparity and evolutionary rates according to different time binnings; we calculated evolutionary rates for all lepidosaurs, and for squamates and rhynchocephalians (plus stem-lepidosaurs) separately. The time spent in some of the analyses is measured in weeks, and it is thus prohibitive to keep adding variables. We aim at testing the robustness of our results for many of the points raised by the reviewers in a subsequent paper using complementary methodologies, including the use of tip dating to infer both divergence times and evolutionary rates while accounting for their uncertainties in a coherent Bayesian statistical framework.

(2) The discovery of this peak in disparity if further linked to high rates of morphological evolution. Note however that disparity and rates are not necessarily linked, and high rates are not necessary to support a JME. However, this result is entirely dependent on the topology and divergence times supported by this study. The methods of inference and calibration employed here are much more simplistic than those of previous studies, and allow for a much poorer integration of uncertainty into subsequent analyses. Results differ crucially from those of all previous studies, including a much faster early diversification of squamates, and much older ages for some crown clades such as snakes, anguimorphs and iguanians. This leads to a concentration of branches with high levels of morphological change in the mid to late Jurassic, ultimately responsible for the peak in rates found. The methods employed and the way in which they are presented once again do not rule out the possibility that this is an artifact of the way the study was set up, and a major reason why no such increase in rates was discovered by previous studies.

We are working on applying a more integrated analysis on the same matrix in order to test the robustness of results. We want to note, regarding the fact that other studies have not recovered fast rates in the Jurassic, that it is unlikely that you can get a concentration of branches with high levels in that period if most of the fossils included are late Cretaceous in age (with a minimum sampling of Early Cretaceous and Jurassic forms). No matter which approach for dating the trees you are using, results will be highly affected by a poor sampling in a key period. A Late Jurassic peak in evolutionary rates is recovered in all the three dating methods used, and only in the third (mbl) the peak appears as low in comparison to the extremely high evolutionary rates recovered in branches concentrated close to the present(we have provided details on why this method can be producing biased results in our responses to the reviewers).

(3) There is a lack of consistency throughout the manuscript. For example, the JME label applied to each of the four panels of Figure 1 seems to represent four different periods of time. The topology in Figure 1A was calibrated using a different method that the rates shown in Figure 1D, making it difficult to draw conclusions. While a staggering 51 supplementary figures are presented, many of these are unedited and thus highly cryptic, others such as the many ordination plots are unlabeled and thus do not contribute much.

Regarding the different method used in figures 1A and 1D, in fact there was an error in the caption of former figure 1D. The method used for that figure was, indeed, the Hedman method and, as such, it is congruent with that of figure 1A. Lack of consistency between Figure 1C and 1D was the result of differences in the plot layout, that have been corrected now (new Figures 4A and 4B).

Some aspects of this manuscript are very interesting, such as the position of extinct lineages under topologies constrained to show different relationships among major extant clades. However, its main claim is somewhere between trivial (a change from 1 to 4 morpho-groups somewhere in the second half of the Jurassic) and potentially unsubstantiated (a coincident peak in rates).

Although it might be true that the identification of the JME might not be surprising to some (based on a crude interpretation of the fossil record), validating this view is not a trivial point. Moreover, many of the claims of numerical approaches recently applied in other works have pointed to the KTR as the main event in the diversification of squamates. Challenging this view is important.

Regarding the initial statement that reads: [the manuscript] “employs relatively simple methodologies that can potentially determine the outcome”, before starting the process of reviewing our manuscript, we followed the reviewers advice and explored the use of other approaches like the Bayesian tip dating under relaxed morphological clocks in order to calculate divergence times and evolutionary rates while accounting for their uncertainties (as implemented by e.g. Zhang and Wang, 2019). One of the main issues we encountered is that, if we report evolutionary rates for the majority rule tree, a great polytomy at the base of Squamata precludes any interpretation of the results. This is critical because this portion of the tree is where we expect to get the high evolutionary rates, at least according to our results from other methods. There is no simple solution for this, other than changing the sampling (by adding or removing taxa) in the hope that we get a more resolved topology. This problem of poor resolution of the tree was circumvented in our methodology by using a subsample of randomly selected most parsimonious trees as the source for evolutionary rates instead of a consensus tree. The reported figures of evolutionary rates of our manuscript account for this variation by showing when high or low evolutionary rates are consistently recovered across all topologies. We are aiming at investigating how the points raised by the reviewers regarding methodology affect results in a forthcoming paper. We are in position, however, to say that this alternative methodology provides divergence times that are within those retrieved by the methodologies used in our manuscript (more specifically, between those of the Hedman and the equal methods). We have shared some of these provisional results with the reviewers.

[Editors’ note: further revisions were suggested prior to acceptance, as described below.]

The manuscript has been improved but there are some remaining issues that need to be addressed, as outlined below:See comments from Reviewer #3.Reviewer #3:I congratulate the authors on the effort put towards revising their manuscript, and after carefully reading their response and new version of the manuscript I have just a handful of minor suggestions that I would like to see implemented, or more strongly stated, in the manuscript before the final version is published.(1) Editing some of the supplementary files shown would not take that much time at all, and it would make them much more readable than they are now. Reducing the size of the label text in the trees now shown as supplementary figures to figure 1, and color coding them by clade, does not represent too much work. Same holds true for color coding by clade the dots in the many supplementary PCO plots.(2) The authors state in their response that "Methods in some other recent studies include extensive genomic data as well, but such data is only available for modern taxa and can only change the overall tree topology, but not the dating of deep divergences, as here." Genomic data can most definitely change a lot more than just topology, and in fact do modify the dating of deep divergences. The methods implemented currently for time-scaling make a number of simplified decisions regarding the calculation of branch lengths that come from the fact that they do not incorporate any quantifiable (morphological or molecular) sense of the degree of divergence occurred between successive nodes. Methods directly employing genomic data have access to this amount of divergence, and can therefore not only incorporate it into the analysis but also eschew some of the simplistic assumptions of methods such as equal or mbl (which remains true regardless of the latter being very common in the literature). It is wrong of the authors to assume that time calibration relies exclusively on the use of the right fossils (although this is certainly crucial), and there are good reasons why approaches incorporating genomic data to divergence time estimation are unambiguously better than those that don't, even when molecular data is available only for extant terminals. I strongly believe that this caveat needs to be made explicit in the main text of the manuscript, and that it is not a problem that disappears just because several different methods explored agree with each other.(3) In a similar vein, I would suggest the authors to explicitly state that some results that directly depend on the estimation of time-calibrated branch lengths (such as the calculation of evolutionary rates) necessitate a confirmation with more robust methods of time calibration, such as those that incorporate molecular data alongside the fossil record, as well as employ more realistic models of diversification such as the fossilized birth-death prior.

We thank again the expert reviewer who made additional comments on our reviewed version of the manuscript. We found them highly valuable, and we have followed them by making the following modifications on the manuscript:

1. We have modified the morphospace plots where all tips are labeled and points are now colored according to clades. We have also replaced the complete taxon names by abbreviations, to avoid (or at least minimise) label overlapping.

2. We have modified the consensus trees, now showing colors according to clade

3. We have replaced the corresponding code for morphospace by an updated version including the code for the new morphospace plots.

4. We have added a new source file for the code for plotting the new (coloured) consensus trees.

5. Following the reviewer’s advice, we have added an explicit statement highlighting that some results that directly depend on the estimation of time-calibrated branch lengths (such as the calculation of evolutionary rates) need a confirmation with more robust methods of time calibration, such as those that incorporate molecular data alongside the fossil record, as well as employ more realistic models of diversification such as the fossilized birth-death prior. This way we address the comments 2 and 3 of the reviewer.

6. We have updated the manuscript text by removing the previous track changes, and now contains only the few modifications made according to the changes described above (mainly related to the captions corresponding to modified figures)